# RPG: A Repository Planning Graph for Unified and Scalable Codebase Generation

**Jane Luo**[1,*,‡] **Xin Zhang**[1,*,†] **Steven Liu**[1,‡] **Jie Wu**[2,‡] **Jianfeng Liu**[1] **Yiming Huang**[3]
**Yangyu Huang**[1] **Chengyu Yin**[4,‡] **Ying Xin**[1] **Yuefeng Zhan**[1] **Hao Sun**[1] **Qi Chen**[1]
**Scarlett Li**[1] **Mao Yang**[1]

[1]Microsoft, [2]Tsinghua University
[3]University of California, San Diego
[4]Beijing Jiaotong University

## Abstract

Large language models excel at generating individual functions or single files of code, yet generating complete repositories from scratch remains a fundamental challenge. This capability is key to building coherent software systems from high-level specifications and realizing the full potential of automated code generation. The process requires planning at two levels: deciding what features and modules to build (proposal stage) and defining their implementation details (implementation stage). Current approaches rely on natural language planning, which often produces unclear specifications, misaligned components, and brittle designs due to its inherent ambiguity and lack of structure. To address these limitations, we introduce the Repository Planning Graph (RPG), a structured representation that encodes capabilities, file structures, data flows, and functions in a unified graph. By replacing free-form natural language with an explicit blueprint, RPG enables consistent long-horizon planning for repository generation. Building on RPG, we develop ZeroRepo, a graph-driven framework that operates in three stages: proposal-level planning, implementation-level construction, and graph-guided code generation with test validation To evaluate, we construct RepoCraft, a benchmark of six real-world projects with 1,052 tasks. On RepoCraft, ZeroRepo produces nearly 36K Code Lines and 445K Code Tokens, on average 3.9× larger than the strongest baseline (Claude Code), and 68× larger than other baselines. It achieves 81.5% coverage and 69.7% test accuracy, improving over Claude Code by 27.3 and 35.8 points. Further analysis shows that RPG models complex dependencies, enables more sophisticated planning through near-linear scaling, and improves agent understanding of repositories, thus accelerating localization. Our data and code are available at `https://github.com/microsoft/RPG-ZeroRepo`.

## 1 Introduction

Recent large language models (LLMs) have shown strong performance on function-level and file-level code generation, reliably producing functions and files from natural language descriptions (Zhu et al., 2024; Wang et al., 2025; Liu et al., 2025; Zeng et al., 2025). However, scaling this capability from functions and files to generate large-scale software repositories from scratch remains a fundamental challenge. The core difficulty is bridging the gap between high-level user intent and the repository's intricate network of files, classes, and dependenciesTao et al. (2025); Li (2025). Successfully navigating this gap necessitates a process of progressive planning, which naturally decomposes into two complementary phases: **proposal-level planning**, which determines *what to build* by defining the functional scope and key capabilities, and **implementation-level planning**, which determines *how to build* it by specifying the file structure, interfaces, dependencies, and data flows.

---

[*]Equal contribution.
[†]Corresponding author.
[‡]This work is done during their internships at Microsoft.

Prior work has explored this challenge through three paradigms. Distributed planning frameworks (e.g., MetaGPT (Hong et al.), ChatDev (Qian et al., 2024)) assign specialized roles such as manager, architect, and engineer to negotiate between requirements and implementations. Workflow-based systems (e.g., Paper2Code (Seo et al., 2025), AutoP2C (Lin et al., 2025)) follow fixed pipelines that first build architectural skeletons before filling in details. Iterative terminal agents (e.g., Open-Hands (Wang et al.), Claude Code (Anthropic, 2025b), Gemini CLI (Google, 2025)) externalize intermediate plans, often in markdown, and refine them step by step. Despite their differences, these approaches share a dependency: natural language as the intermediate medium for planning.

While natural language remains a flexible and human-readable medium, it can often be less efficient for large-scale repository generation. Its inherent ambiguity may blur distinctions between intent and constraints (Wang et al., 2024), its lack of explicit hierarchy makes dependency tracking particularly difficult (Besta et al., 2024), and static plans may gradually degrade over long horizons without adaptive adjustment (Sun et al., 2023). When extended to automatic repository generation, these limitations can more easily lead to unstable proposal-level planning, where functionalities are sometimes incomplete, overlapping, or unevenly scoped (Zhu et al., 2025), and fragmented implementation-level planning, where plans drift across iterations, introducing inconsistencies in dependencies, data flows, and modular boundaries (Almorsi et al., 2024; Ashrafi et al., 2025).

To address these limitations, we introduce the Repository Planning Graph (RPG), a persistent and evolvable representation that unifies proposal and implementation planning for repository generation. RPG encodes functional goals and designs in a single graph: nodes capture hierarchical capabilities with files, classes, and functions, while edges specify semantic relations and data flows. By replacing free-form language with a structured medium, RPG provides a compact, interpretable basis for consistent long-horizon planning. Building on this representation, we develop ZeroRepo, a graph-driven framework for controllable repository generation. Given a user specification, ZeroRepo proceeds in three stages: (1) **Proposal-Level Construction**, which organizes and refines requirements into a functional graph via a large-scale feature tree; (2) **Implementation-Level Construction**, which expands this graph into the full RPG by encoding file skeletons, interfaces, and flows; and (3) **Graph-Guided Code Generation**, which traverses the RPG in topological order with test-driven development, guided localization, and iterative editing.

To evaluate agents' planning ability in repository generation, we construct **RepoCraft**, a benchmark of six projects with 1,052 tasks. On RepoCraft, ZeroRepo attains 81.5% functional coverage and a 69.7% pass rate, exceeding the strongest baseline (Claude Code) by 27.3 and 35.8 points, while producing repositories with 36K Lines of Code and 445K Code Tokens, about 3.9× larger than Claude Code and 68× larger than other baselines. Further analysis shows that **Repository Planning Graph** (RPG) captures complex dependencies, including inter-module data flows and function-level relations. It enables near-linear scaling of functionality and code size, supporting complex planning and providing a foundation for large-scale repositories and long-horizon development. As a global representation, RPG enhances agents' repository understanding and accelerates localization.

Our Contributions are list below:

1. We introduce the Repository Planning Graph (RPG), a unified representation integrating proposal- and implementation-level planning, encoding functionality, file structures, data flows, and function designs.
2. We develop ZeroRepo, a graph-driven framework that constructs RPG through proposal- and implementation-level planning, and generates code with test validation.
3. To evaluate agent planning ability in repository generation, we build **RepoCraft**, a benchmark of 6 projects with 1,052 tasks assessing coverage, accuracy, and code scale.
4. On RepoCraft, ZeroRepo achieves strong improvements over baselines, reaching 81.5% functional coverage and nearly 69.7% test accuracy, while producing repositories 3.9× larger than the strongest baseline. Further analysis shows that RPG captures complex dependencies, enables more sophisticated planning through near-linear scaling, and enhances agents' repository understanding, thereby accelerating localization.

## 2 RELATED WORK

**LLM-based Code Generation** SOTA models (e.g., GPT-4o (OpenAI, 2024), Claude 4 (Anthropic, 2025a), DeepSeek-R1 (Guo et al., 2025)) excel at SWE tasks such as code completion, test

generation (Zafar et al., 2022; Dakhel et al., 2024), refactoring (Gautam et al., 2025), and program repair (Jimenez et al.). Instruction-tuned variants (e.g., Qwen-Coder (Hui et al., 2024), EpiCoder (Wang et al., 2025)) further improve reliability. These advances deliver strong function-level performance, forming the basis for broader Software Engineering(SWE) progress.

**Agents for Repository-Level Generation**   Agent frameworks embed LLMs and can automate SWE tasks: multi-agent systems (Qian et al., 2024; Hong et al.) assign roles; workflows (Seo et al., 2025; Lin et al., 2025) run pipelines; industrial tools (OpenAI, 2025; Anthropic, 2025b) enable more advanced automation. Yet most rely on ephemeral language plans that lack persistent structure, producing fragmented implementations; ZeroRepo instead uses a graph-guided abstraction for structured planning and execution.

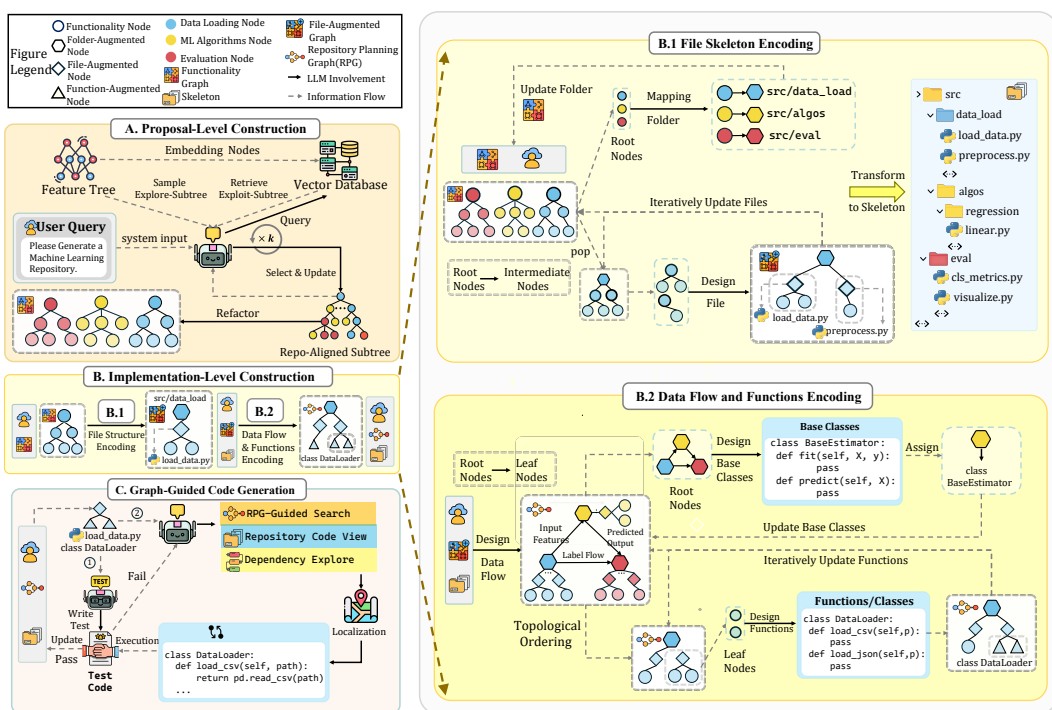

Figure 1: The ZeroRepo pipeline for repository generation. (A) Proposal-Level Construction maps query to a functionality graph. (B) Implementation-Level Construction refines via (B1) File Structure Encoding into a file-augmented graph and (B2) Data-Flow/Function Encoding into the Repository Planning Graph (RPG). (C) Graph-Guided Code Generation traverses RPG to generate the repository.

## 3   REPOSITORY PLANNING GRAPH CONSTRUCTION

To address the ambiguity of natural language plans, we propose the **Repository Planning Graph** (RPG), a structured representation that encodes repository functionality and implementation logic as nodes and edges. Building on RPG, we develop ZeroRepo, a framework for repository generation from scratch. This section first introduces the structure of RPG (§3.1), and then explains how ZeroRepo constructs it through proposal-level planning (§3.2) and implementation-level refinement (§3.3). The overall pipeline is shown in Figure 1(A-B).

### 3.1   REPOSITORY PLANNING GRAPH STRUCTURE

As shown in Figure 2, RPG provides a unified representation for repository planning by encoding functionality and implementation in a structured graph that is both explicit and machine-interpretable, rather than unstable or ambiguous natural language. Its nodes carry dual semantics and serve complementary roles. At the functional level, they represent progressively refined capabilities: high-level modules (e.g., `algorithms`) decompose into mid-level components and ultimately into leaf nodes corresponding to concrete algorithms. At the structural level, this hierarchy closely mirrors repository organization: root nodes align with file regions or directories,

intermediate nodes with files, and leaf nodes with specific functions or classes, thereby unifying functional decomposition with the code structure.

Beyond hierarchical nodes, edges in RPG capture explicit, grounded, and execution dependencies across levels. Inter-module edges (black arrows in Figure 2) encode data flows, e.g., outputs from `Data Loading` feeding into `ML Algorithms` and then `Evaluation`. Intra-module edges (gray dashed arrows) capture file-level orderings; for instance, `load_data.py` precedes `preprocess.py`, with outputs passed to preprocessing. Collectively, these edges impose a topological order aligning functional decomposi-

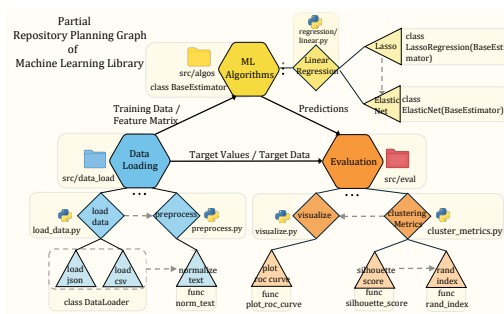

Figure 2: Repository Planning Graph: nodes encode repository capabilities, edges capture hierarchy and flows

tion with code organization, ensuring coherence between global semantics and local implementation.

## 3.2 PROPOSAL-LEVEL CONSTRUCTION

At the proposal level, the goal is to translate high-level specifications into a functionality graph. This involves two steps: selecting a repository-aligned subtree via an explore–exploit search over ZeroRepo's knowledge base, and refactoring it into the graph (algorithms detailed in Appendix B.1).

**A Global Tree as Knowledge Base** LLMs alone provide unstable and biased capability enumeration, often with incomplete coverage (Valmeekam et al., 2023; Armony et al., 2025).To stabilize planning, we use the EpiCoder Feature Tree (Wang et al., 2025), an ontology of 1.5M software capabilities, as ZeroRepo's knowledge base. Its breadth and hierarchy act as a structured prior, mitigating randomness and bias and improving coverage. For efficient retrieval, each node is embedded in a vector space, with its hierarchical path stored as metadata in a vector index. This preserves semantic similarity and structural context, enabling precise, scalable grounding. Statistics are in Appendix B.2.

**Explore–Exploit Subtree Selection** Using the Feature Tree as a structured knowledge base, the first step is to construct a **repo-aligned subtree** tailored to the user's goal. Exhaustive enumeration is infeasible at the 1.5M scale, so ZeroRepo incrementally expands the subtree via an explore–exploit strategy. (1) **Exploitation** ensures precision: we retrieve top-$k$ feature paths most aligned with the user goal and augment them with keywords suggested by LLM queries. (2) **Exploration** ensures diversity: we deliberately expand into unvisited regions of the ontology to capture less obvious but relevant functionalities. Candidates from both strategies are filtered by the LLM and integrated into the evolving subtree, yielding a balanced and comprehensive foundation for downstream planning.

**Refactoring by Goal Alignment** The repo-aligned subtree, though capturing relevant functionalities, still inherits the generic organization of the global ontology. To align it with the user's repository goal, we refactor feature placements into a modular **functionality graph**. The LLM partitions functionalities into cohesive modules following software engineering principles of cohesion and coupling. For instance, in a machine learning library, metrics such as `silhouette_score` are reorganized under an evaluation module rather than within clustering algorithms. The resulting graph establishes clear functional boundaries, encoding proposal-level planning directly into the representation.

## 3.3 IMPLEMENTATION-LEVEL CONSTRUCTION

After proposal-level construction establishes the multi-level functional plan, the graph is further enriched with implementation details, culminating in the complete Repository Planning Graph (RPG) at this stage. The process includes encoding the repository's file structure, modeling inter-module data flows and intra-module orderings, and specifying concrete functions and interfaces.

### 3.3.1 FILE STRUCTURE ENCODING

While proposal-level planning defines modular functionalities, it remains abstract and detached from implementation. To bridge this gap, the graph is extended with folder and file layouts, instantiating a repository skeleton that maps modules into executable structures, yielding a file-augmented graph.

**Folder-Level Encoding**    Proposal-level planning partitions functionalities into modular subgraphs, yet this abstraction does not define the repository's structure. We enrich root nodes with folder-level specifications, assigning each subgraph a directory namespace (e.g., `algos/`, `eval/`). This encoding couples semantic modularity with structural separation, ensuring descendants inherit a consistent namespace and the repository skeleton aligns with high-level capability decomposition.

**File-Level Encoding**    Once folder regions are encoded at root nodes, the graph is enriched by assigning files to intermediate nodes, specifying how module functionalities map to executable files. For example, preprocessing utilities consolidate into `preprocess.py`, while models like linear regression and variants group into `linear_models.py`. Embedding file structure preserves semantic cohesion, reduces cross-file coupling, and yields a file-augmented graph anchoring design.

### 3.3.2    Data Flow and Functions Encoding

After obtaining the file-augmented graph, this stage finalizes the full Repository Planning Graph (RPG) by assigning executable roles to leaf nodes. To ensure coherence across modules and functions, we first incorporate inter- and intra-module data flows as input–output constraints, then abstract shared structures as design anchors, and finally refine leaf nodes into concrete functions or classes.

**Data-Flow Encoding**    To ground interface design in execution, the graph is augmented with data-flow edges capturing inter- and intra-module relations. Globally (Figure 2), typed input–output flows connect subgraph roots; for example, a data-loading module may provide an `array` of training data to algorithms. Locally, files within a module are ordered to ensure coherent, dependency-aware implementation. These flows impose a hierarchical order constraining and organizing interface design.

**Abstracting Global Interfaces**    To improve scalability and maintainability, recurring input–output patterns across modules are abstracted into common data structures or base classes, serving as anchors that enforce consistency and reduce redundancy. For example, algorithms can be unified under a `BaseEstimator` class to standardize interaction with preprocessing and evaluation modules.

**Adaptive Interface Design**    Within each file-level subgraph, leaf features cluster into executable interfaces by semantic relatedness. Independent features become standalone functions, while interdependent ones form shared classes with methods. For example, in Figure 2, `load_json` and `load_csv` are grouped into a `DataLoader` class, while `elastic_net` is implemented as an `ElasticNet` class. This adaptive mapping balances granularity and cohesion, yielding a Repository Planning Graph (RPG) that preserves modularity and semantic consistency at repository scale.

## 4    Graph-Guided Code Generation

As shown in Figure 1(C), given a user query and the completed RPG, ZeroRepo generates repositories by traversing the graph in topological order, ensuring dependencies precede dependents. At each leaf node, test-driven development (TDD) is applied: tests are derived from the specification, then the corresponding functions or classes are implemented and validated; failing cases trigger revisions until passing or the iteration limit is reached. Only functions that pass all tests are committed, enabling incremental expansion while preserving stability. Further details are in Appendix D.

**Graph-Guided Localization and Editing**    To handle implementation and debugging requests, we adopt a two-stage workflow: first localizing the target in the RPG, then editing the associated code. Localization leverages the graph's global structure and three tools: (1) **RPG-Guided search**, which uses functionality-based fuzzy matching to identify candidate functions; (2) **repository code view**, retrieving full interface bodies for inspection or modification; and (3) **dependency exploration**, tracing edges to reveal related modules and interactions. Once localized, the agent revises or generates the corresponding code to complete the requested implementation or repair.

**Graph-Guided Test Validation**    To ensure correctness and catch errors early, validation proceeds in stages aligned with the graph. Each function or class is first checked in isolation using unit tests from its docstring. Validated components trigger regression tests when modified, while subgraphs undergo integration tests to verify data flows and contracts across modules. A lightweight majority-vote diagnosis separates genuine implementation errors from environment or test issues, automatically handling the latter and returning the former for repair via the localization–editing workflow.

Table 1: Overview of the six reference repositories and their paraphrased counterparts (Para. Name) in RepoCraft. #F. Cate. denotes functional categories, #Files the total source files, LOC the effective lines of code, and Task Counts the evaluation tasks for measuring code accuracy.

| Real Repo | Para. Name | #F. Cate. | #Files | LOC | Code Tokens | Task Counts |
|---|---|---|---|---|---|---|
| scikit-learn | MLKit-Py | 47 | 185 | 65,972 | 592,187 | 236 |
| pandas | TableKit | 81 | 217 | 106,447 | 943,873 | 175 |
| sympy | SymbolicMath | 40 | 699 | 218,924 | 2,339,881 | 192 |
| statsmodels | StatModeler | 88 | 271 | 83,325 | 893,824 | 234 |
| requests | HttpEasy | 22 | 17 | 2,793 | 22,297 | 50 |
| django | PyWebEngine | 42 | 681 | 109,457 | 917,622 | 165 |

## 5 EXPERIMENT SETUP

### 5.1 REPOCRAFT BENCHMARK

A key challenge in repository-level generation is the lack of benchmarks for end-to-end reasoning and planning. Prior work mainly targets incremental development (editing, refactoring, or bug fixing in existing codebases (Jimenez et al.; Gautam et al., 2025; Zhang et al., 2025; Li et al., 2025; Huang et al., 2024)) or repository generation with detailed skeletons and specifications that limit autonomous planning (Zhao et al., 2025; Starace et al., 2025). RepoCraft fills this gap by requiring agents to build complete repositories from high-level natural language descriptions and evaluating them against real-world projects on scale, functionality, and correctness, with final statistics in Table 1.

#### 5.1.1 REFERENCE REPOSITORY SELECTION

RepoCraft grounds evaluation in six widely used Python projects: *scikit-learn*, *pandas*, *sympy*, *statsmodels*, *requests*, and *django*. They are chosen as strong references for embodying high-quality engineering with active community development, modular design, and comprehensive tests. Covering domains from scientific computing to web frameworks, they ensure breadth and realism. To mitigate pretraining leakage, we paraphrase their names and descriptions before providing them to agents.

#### 5.1.2 METRICS

RepoCraft evaluates generated repositories along three dimensions (detailed in Appendix E.3.1):

**Functionality**   We evaluate both **Coverage** and **Novelty**. Coverage measures the proportion of functional categories defined in official documentation that are represented in the generated repository; a category is counted as covered if at least one generated functionality corresponds to it (Appendix E.4). Novelty measures the proportion of generated functionalities not in the reference taxonomy, capturing the system's ability to propose coherent and meaningful new capabilities beyond the specification.

**Accuracy**   Correctness at task level via (1) **Pass Rate**: fraction of tests passed, and (2) **Voting Rate**: fraction confirmed by majority-vote checks.

**Code-Level Statistics**   We sreport repository scale indicators, including file count, normalized Lines of Code (LOC), and token count, measured after excluding non-core code such as tests and examples.

#### 5.1.3 FUNCTIONAL TASK CONSTRUCTION AND EVALUATION

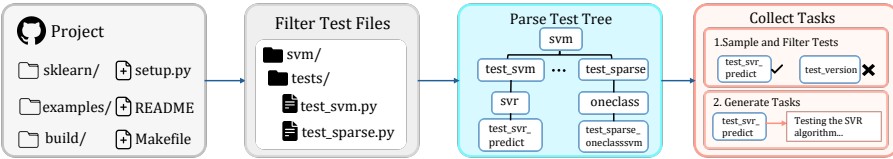

Figure 3: Pipeline for Evaluation Task Collection. It comprises test file filtering, hierarchical parsing into test trees, sampling and filtering, and final task generation.

To assess models' planning ability on constructed repositories, we evaluate whether they (i) implement the intended algorithms and (ii) realize them correctly. Simple measures of repository size or coverage

are insufficient for this purpose, so RepoCraft introduces task-level evaluations that capture both functional fidelity and implementation accuracy (see Appendix E.2 for details).

To enable such fine-grained evaluation, RepoCraft derives tasks from reference repositories. As shown in Figure 3, we collect all available test functions and classes, organize them hierarchically following each project's modular structure, and apply stratified sampling to ensure representative coverage. Trivial or non-algorithmic tests are filtered out, resulting in a diverse and computationally meaningful set of 1,052 tasks that closely mirror practical software evaluation.

Each task includes a natural language description of the target algorithm, a ground-truth test, and auxiliary materials. Evaluation proceeds in three steps: (1) **Localization**, mapping requirements to candidate functions or classes in the generated repository; (2) **Semantic Validation**, applying majority-vote checks over two rounds to confirm fidelity to the specification; and (3) **Execution Testing**, adapting and running the ground-truth test to verify interface correctness under realistic inputs and outputs. This design mirrors real-world development while reducing sensitivity to spurious model errors. We use `o3-mini` as the base model for automated evaluation.

### 5.2 BASELINES

We compare three paradigms: (1) **Multi-agent** frameworks (MetaGPT (Hong et al.), ChatDev (Qian et al., 2024)) assigning specialized roles for end-to-end development; (2) **Workflow-based** system (Paper2Code (Seo et al., 2025)) with a fixed three-stage pipeline; (3) **Terminal agents** (Codex CLI (OpenAI, 2025), Claude Code CLI (Anthropic, 2025b), Gemini CLI (Google, 2025), Open-Hands (Wang et al.)) performing natural language editing, debugging, and multi-file reasoning. For comparability, MetaGPT, ChatDev, and Paper2Code are run with two backbones: o3-mini (OpenAI, 2025) and Qwen3-Coder-480B-A35B-Instruct (Qwen3-Coder) (Team, 2025). Codex CLI, Claude Code CLI, and Gemini CLI are evaluated with their official strongest model. **We enable Terminal Agents to retrieve real-world knowledge via web search.** To ensure fairness, all runs extend to 30 iterations, with agents prompted at each step to propose or implement functionality.

### 5.3 IMPLEMENTATION DETAILS

We run 30 iterations for feature selection in Proposal-Level Graph Construction and use infly/inf-retriever-v1 (Yang et al., 2025) for node embeddings. In the Code Generation Stage, each function allows up to 8 debugging iterations with 20 localization attempts. For test failures, we use 5-round majority voting for attribution and allow up to 20 remediation attempts for test or environment errors.

## 6 MAIN RESULTS

Table 2: Performance of agent frameworks and model backbones on RepoCraft. "Nov." denotes novelty rate; the number in parentheses is Novel/Total, where Novel is the count of novel functionalities and Total the number of planned ones. Gold Projects serve as a confidence ablation for the automatic evaluation pipeline, and per-repository results are reported in Appendix F.2.

| Agent | Model | Cov. (%) ↑ | Nov. (%) (Novel/Total) ↑ | Pass. / Vot. (%) ↑ | Files ↑ | LOC ↑ | Tokens ↑ |
|---|---|---|---|---|---|---|---|
| MetaGPT | o3-mini | 16.6 | 0.0 (0.0/24.8) | 4.5 / 10.2 | 2.3 | 225.3 | 2180.3 |
|  | Qwen3-Coder | 17.1 | 0.0 (0.0/32.7) | 3.2 / 9.4 | 8.5 | 326.5 | 3369.3 |
| ChatDev | o3-mini | 18.3 | 9.2 (3.0/32.8) | 2.6 / 10.5 | 5.8 | 410.3 | 4458 |
|  | Qwen3-Coder | 22.1 | 3.9 (1.5/38.3) | 6.9 / 11.6 | 6.3 | 540.7 | 5422.2 |
| OpenHands | o3-mini | 22.0 | 0.3 (0.1/36.5) | 5.1 / 16.9 | 9.8 | 292.2 | 2712.8 |
|  | Qwen3-Coder | 21.7 | 0.0 (0.0/33.7) | 5.8 / 11.2 | 8.3 | 458.0 | 4778.3 |
| Paper2Code | o3-mini | 21.7 | 5.2 (2.1/40.0) | 6.0 / 15.8 | 7.2 | 547.7 | 5920.8 |
|  | Qwen3-Coder | 30.2 | 5.5 (4.0/73.8) | 4.9 / 15.9 | 8.8 | 1365.2 | 14,555.0 |
| Codex CLI | o3 pro | 28.4 | 0.0 (0.0/48.5) | 11.0 / 20.0 | 5.3 | 611.5 | 6248.5 |
| Gemini CLI | gemini 2.5 pro | 42.0 | 0.6 (0.8/132.7) | 14.5 / 37.9 | 15.2 | 1484.8 | 14,922.2 |
| Claude Code CLI | claude 4 sonnet | 54.2 | 6.7 (41.6/621.0) | 33.9 / 52.5 | 33.3 | 10,586.7 | 105,236.2 |
| Gold Projects | Human Developers | - | - | 81.0 / 92.0 | 345 | 97,819.7 | 951,614 |
| **ZeroRepo** | o3-mini | **81.5** | **13.6 (151.5/1114.2)** | **69.7 / 75.0** | **271.5** | **23,977.3** | **260,761.2** |
|  | Qwen3-Coder | **75.1** | **9.2 (108.3/1173.3)** | **57.3 / 68.0** | **389.0** | **36,941.0** | **445,511.8** |

**RPG enables richer functionality and larger repositories.** ZeroRepo shows that RPG–guided planning yields repositories of greater scale, diversity, and novelty than existing approaches. On RepoCraft, it achieves 81.5% coverage with `o3-mini`, a 27.3% absolute improvement over the

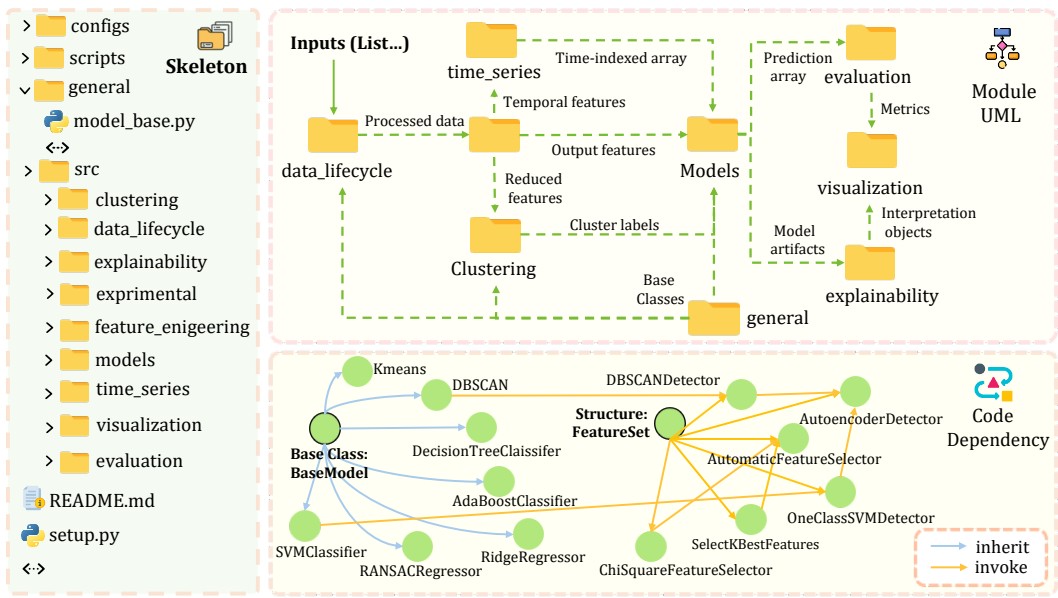

Figure 4: Dependencies in the repository generated by Qwen3-Coder on `MLKit-Py`, including (1) the repository skeleton at folder/file level, (2) inter-module data flows, and (3) interfaces dependencies.

strongest baseline (Claude Code). Beyond covering required functionality, ZeroRepo demonstrates innovation, reaching novelty rates of 11–13% with over 100 new functionalities, whereas most baselines add fewer than 10. In size, ZeroRepo with `Qwen3-Coder` generates 36K LOC and 445K tokens, corresponding to 3.9× Claude Code and about 68× other baselines. Among methods, ZeroRepo is closest to human-developed Gold Projects, underscoring that RPG is a key structured representation for building repositories that are larger, more diverse, and closer to real-world complexity.

**RPG enhances reasoning consistency and structural fidelity.** Beyond scale, ZeroRepo delivers higher correctness and stability. To ensure reliability, we first validate the automatic localization and validation pipeline on human-developed Gold Projects, where it achieves 81.0% pass rate and 92.0% voting agreement, establishing the ceiling under our test harness. Under the same protocol, ZeroRepo attains a 69.7% pass rate with `o3-mini`, an absolute improvement of 35.8% compared to the Claude Code. These results indicate that RPG serves as a structured reasoning representation that enforces modular boundaries and functional contracts, thereby supporting coherent planning and yielding repositories that more faithfully realize intended specifications.

**RPG induces complex data flows and dependencies.** To illustrate the capacity of RPG–guided planning for generating complex repositories, we visualize ZeroRepo with `Qwen3-Coder` on the `MLKit-Py` task. At the file level, RPG organizes a coherent folder hierarchy; at the module level, inter-module flows define execution pipelines from `data_lifecycle` through `clustering` and `models` to `evaluation`; and at the function level, inheritance and invocation edges capture intricate class interactions. These results show that RPG induces layered dependencies and coordinated execution, enabling repositories with both structural complexity and internal coherence.

# 7 ANALYSIS

## 7.1 ANALYSIS OF THE RPG'S SCALABILITY

**RPG enables near-linear growth of repository functionalities.** A key question in repository-level generation is whether functionalities can continue to expand with iterative planning over time and scale in practice, or whether growth instead stagnates. To examine this, we compute the number of planned features at each iteration on RepoCraft, averaging results across 30 rounds for strong baselines (Claude Code, Gemini CLI, Codex CLI) and for ZeroRepo.

As shown in Figure 5, ZeroRepo exhibits near-linear growth, surpassing 1,100 leaf features with `o3-mini`, while natural-language–based baselines show only limited scalability: Claude Code grows steadily but with diminishing rates, Gemini CLI increases slowly before converging by round 30, and Codex ceases features after 4–5 iterations. These results demonstrate that RPG provides a persistent, extensible substrate that refines high-level goals into richer functionalities while sustaining structural consistency, ultimately offering a stronger and more reliable foundation for modeling complex repositories.

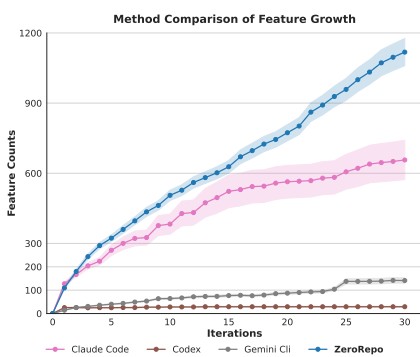

Figure 5: Feature comparison of ZeroRepo (`o3-mini`) against strong baselines across iterations.

**RPG ensures near-linear growth in repository size.** Functional scalability matters only when realized in code. We measure this by tracking repository growth in lines of code (LOC) across iterations. As shown in Figure 6, ZeroRepo sustains near-linear expansion, surpassing 30K LOC within 30 iterations. In contrast, natural-language baselines stagnate: Claude Code and Gemini CLI plateau at 3–4K LOC, while Codex stays below 1K. This gap reflects a core weakness: natural language planning accumulates inconsistencies, producing fragmented specifications that fail to converge into coherent code. By contrast, RPG provides a structured, extensible representation where new functionalities are anchored in explicit modules, interfaces, and data flows. This grounding ensures expansions materialize as code, enabling repositories to grow in size and integrity. The results highlight RPG's capacity to sustain scaling in codebase size and structure, establishing it as a foundation for long-horizon code generation.

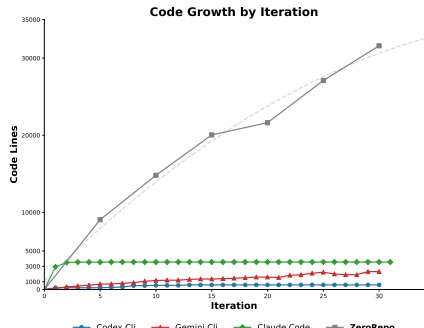

Figure 6: Scaling behavior of LOC across iterations on MLKit-Py.

## 7.2 ANALYSIS OF RPG'S STABILITY AND INNOVATION POTENTIAL

**RPG supports comprehensive and extensible functionality.** A key challenge in repository generation is satisfying user requirements while coherently extending beyond them. As shown in Table 3, ZeroRepo scales coverage from 70.2% (Iter 5) to nearly 96% (Iter 30), significantly surpassing baselines that plateau below 60% (Table 2). Simultaneously, it maintains 8% novelty with over 100 additional features, whereas baselines contribute fewer than 50. These results suggest RPG functions as a persistent planning substrate, enabling broad coverage and principled growth beyond reference implementations (validated in Appendix F.3).

Table 3: Coverage and Novelty of the Constructed RPG over Iterations on MLKit-Py

| Iteration | Cov. (%) ↑ | Nov. (%) ↑ |
|---|---|---|
| 5 | 70.2 | 4.6 (15.3/336.1) |
| 10 | 80.9 | 5.4 (29.01/542.0) |
| 15 | 83.0 | 4.9 (39.0/796.0) |
| 20 | 85.1 | 5.2 (51.0/981.0) |
| 25 | 87.2 | 7.0 (73.5/1043.0) |
| 30 (**ours**) | **95.7** | **7.9 (99.4/1258.0)** |

## 7.3 ANALYSIS OF GRAPH-GUIDED LOCALIZATION

**Graph guidance accelerates agent localization.** We evaluate the impact of RPG guidance by comparing localization steps with and without graph support (Table 4). Across Integration Testing (Int. Test.), Incremental Development (Incr. Dev.), and Debugging (Debug.), graph-guided search reduces effort by 30–50%. This shows that RPG equips agents with a principled navigation mechanism, enabling faster dependency tracing, more accurate bug localization, and smoother module integration, thereby improving efficiency. Compared to natural language, RPG offers a global structural representation of the repository, enabling agents to localize targets from a functionality-wide perspective and accelerating the development cycle.

Table 4: Ablation results for Graph-Guided Localization on MLKit-Py using `o3-mini`. Steps (mean ± SD). "- wo/- Graph" denotes ZeroRepo without Graph.

| Category | IntTest | IncDev | Debug |
|---|---|---|---|
| ZeroRepo | **6.2 ± 2.1** | **6.8 ± 1.8** | **5.8 ± 2.8** |
| - w/o Graph | 13.3 ± 11.1 | 10.8 ± 2.6 | 8.5 ± 2.9 |

## 7.4 ANALYSIS OF AUTOMATED EVALUATION

**Automated evaluation is reliable and aligned with human judgments.** Table 5 shows that automated evaluators preserve the human ranking and relative gaps between ZeroRepo and Claude-Code across all metrics. In particular, o3-mini attains strong Pearson correlations with human scores (0.893 for coverage and 0.958 for novelty), indicating close agreement at the instance level. These results suggest that the automated judge captures the same semantic distinctions as human experts, supporting its use as a robust and scalable proxy for manual assessment in our experiments. Detailed results are provided in Appendix E.5.

Table 5: Comprehensive evaluation. Top: performance metrics. Bottom: Pearson agreement with humans.

| Method | Metric | DeepSeek | GPT-5 | o3-mini | Human |
|---|---|---|---|---|---|
| | | *Performance Evaluation* | | | |
| Claude-Code | Cov. (%) | $53.4 \pm 1.1$ | $58.7 \pm 1.2$ | $52.9 \pm 0.6$ | **50.5** |
| | Nov. (%) | $4.5 \pm 0.6$ | $3.9 \pm 0.9$ | $6.3 \pm 0.2$ | **4.0** |
| | Acc (P/V) | $38.7/52.1$ | $35.1/49.9$ | $36.2/52.0$ | **33.0/50.0** |
| ZeroRepo | Cov. (%) | $76.0 \pm 0.8$ | $81.7 \pm 0.1$ | $82.8 \pm 2.2$ | **80.4** |
| | Nov. (%) | $9.1 \pm 0.4$ | $7.0 \pm 0.3$ | $13.5 \pm 0.8$ | **10.2** |
| | Acc (P/V) | $61.9/79.2$ | $65.1/70.3$ | $69.8/75.0$ | **62.0/73.0** |
| | | *Pearson Consistency w/ Human* | | | |
| Cov. Pearson | | 0.7801 | 0.8149 | **0.8932** | – |
| Nov. Pearson | | 0.8901 | 0.8690 | **0.9587** | – |

## 8 ABLATION ON EPICODER FEATURE TREE

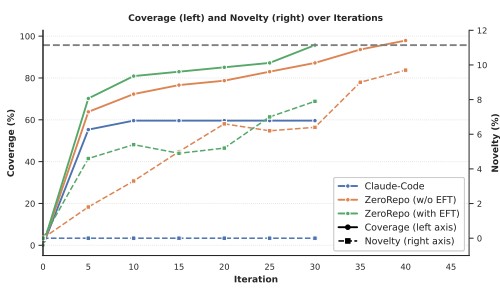

Figure 7: Ablation on the EpiCoder Feature Tree (EFT): coverage (left axis) and novelty (right axis) over iterations.

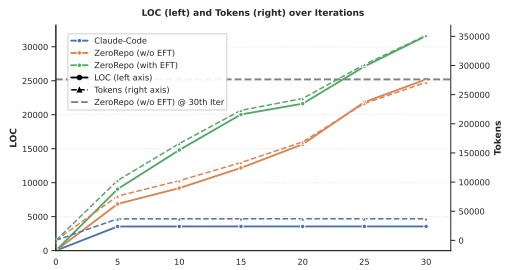

Figure 8: Ablation on the EpiCoder Feature Tree (EFT): LOC (left axis) and generated tokens (right axis) over iterations.

To isolate the RPG structure, we ablate the EpiCoder Feature Tree (EFT) on MLKit-Py (o3-mini). We compare *Claude-Code* (unstructured), *ZeroRepo (w/o EFT)*, and *ZeroRepo (with EFT)*, extending the EFT-free run to 40 iterations to assess convergence (Table 6, Figures 7–8).

**RPG structure drives scalability.** Comparing *ZeroRepo (w/o EFT)* against *Claude-Code* isolates the structural impact. While the unstructured baseline saturates at Iteration 10 (59.6% coverage), the RPG agent sustains growth, reaching 25,202 LOC and 87.2% coverage by Iteration 30. This $7\times$ volume increase suggests repository-scale planning is intrinsic to the graph structure, mitigating context limitations that stall unstructured models.

Table 6: Iteration-30 ablation on MLKit-Py (o3-mini).

| Setting | Cov. (%) | Nov. (%) | Files | LOC | Tokens |
|---|---|---|---|---|---|
| Claude-Code | 59.6 | 0.0 (0/163) | 31 | 3,559 | 37,056 |
| ZeroRepo (w/o KB) | 87.2 | 6.4(62/974) | 191 | 25,202 | 271,039 |
| ZeroRepo (with KB) | **95.7** | 7.9 ( 99/1258) | **266** | **31,596** | **351,554** |

**EpiCoder Feature Tree as an Accelerator.** Ablation confirms that while EFT enhances efficiency, it is not a prerequisite for capability. Removing EFT introduces a temporal delay rather than a performance ceiling: ZeroRepo (w/o EFT) reaches 97.9% coverage by Iteration 40, matching the EFT-enhanced version at Iteration 30. Both configurations exhibit robust linear growth in LOC ($y$) over iterations ($x$). Strong linear fits for EFT-enhanced ($y \approx 983x + 2992$, $R^2 = 0.97$) and EFT-free ($y \approx 800x + 989$, $R^2 = 0.98$) settings indicate the RPG structure ensures stable planning, with EFT optimizing the initial context (intercept) and generation velocity (slope).

## 9 CONCLUSION

In this paper, we presented the Repository Planning Graph (RPG), a structured representation unifying proposal- and implementation-level planning for repository generation. Building on RPG, we developed ZeroRepo, a graph-driven framework that achieves state-of-the-art coverage, correctness, and scalability on the RepoCraft benchmark. Our analyses show that RPG captures complex dependencies, supports increasingly sophisticated planning through near-linear scaling, and enhances agents' repository understanding, thereby improving localization. These findings highlight the promise of graph-based representations as a foundation for long-horizon, large-scale repository generation.

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

## A   THE USE OF LARGE LANGUAGE MODEL

In this paper, we employ a large language model (LLM) for proofreading and icon creation.

## B   APPENDIX OF PROPOSAL-LEVEL GRAPH CONSTRUCTION

The construction of the RPG is central to our framework, as it transforms high-level repository goals into a structured and persistent representation. The process starts with carefully designed prompting strategies for selecting repository-relevant functionalities from the global feature ontology, followed by iterative refinement to ensure both semantic coverage and modular coherence.

### B.1   ALGORITHMS OF FUNCTIONALITY GRAPH CONSTRUCTION

---

**Algorithm 1** Feature Sampling with Diversity-Aware Rejection Sampling

---

**Require:** Root node $R$; frequency library $F$; temperature $t$; per-tree sample size $S$; overlap threshold $\rho$; maximum number of retries $T_{\max}$

 1: **function** BASESAMPLE($R, F, t, S$)
 2:     selected_set $\leftarrow \emptyset$
 3:     **for** $s = 1$ to $S$ **do**
 4:         $C \leftarrow$ GET_CHILDREN($R$)
 5:         **if** $C = \emptyset$ **then**
 6:             **break**
 7:         **end if**
 8:         $f_i \leftarrow F[i]$ for all $i \in C$
 9:         $p_i \leftarrow f_i / \sum_{j \in C} f_j$ for all $i \in C$
10:         $q_i \leftarrow$ TEMPTRANSFORM($p_i, t$) for all $i \in C$
11:         cur_node $\leftarrow$ SAMPLE_NODE($C, [q_1, q_2, \ldots]$)
12:         selected_set.ADD(cur_node)
13:         $R \leftarrow$ cur_node                                    ▷ move root downward for next step
14:     **end for**
15:     **return** selected_set
16: **end function**

17: **function** REJECTSAMPLE($R, F, t, S, \rho, T_{\max}$)
18:     best_T $\leftarrow \emptyset$;    best_ovl $\leftarrow +\infty$
19:     $T^\star \leftarrow \emptyset$
20:     **for** $\tau = 1$ to $T_{\max}$ **do**                              ▷ retry up to $T_{\max}$ times
21:         $T_{\text{cand}} \leftarrow$ BASESAMPLE($R, F, t, S$)                    ▷ sample a candidate tree
22:         ovl $\leftarrow$ OVERLAP($T_{\text{cand}}, \mathcal{S}_{\text{seen}}$)                   ▷ compute overlap with seen nodes
23:         **if** ovl $<$ best_ovl **then**
24:             best_ovl $\leftarrow$ ovl;    best_T $\leftarrow T_{\text{cand}}$                ▷ update best candidate so far
25:         **end if**
26:         **if** ovl $\leq \rho$ **then**
27:             $T^\star \leftarrow T_{\text{cand}}$                     ▷ accept immediately if overlap $\leq$ threshold
28:             **break**
29:         **end if**
30:     **end for**
31:     **if** $T^\star = \emptyset$ **then**
32:         $T^\star \leftarrow$ best_T                        ▷ fallback: choose least-overlap candidate
33:     **end if**
34:     **return** $T^\star$                                    ▷ return the final selected tree
35: **end function**

---

**Rejection Sampling Algorithm**   We extend the base sampling strategy introduced in EPI-CODER (Wang et al., 2025) by incorporating a diversity-aware rejection mechanism, as shown

in Algorithm 1. At each step, a candidate tree is accepted only if its overlap with previously sampled nodes is below a specified threshold; otherwise, the tree with the minimal overlap is returned. This encourages broader feature space exploration under a limited number of sampling iterations.

**Repository-Aligned Subtree Selection** Algorithm 2 outlines the procedure for constructing a repository-specific feature subtree from a global feature tree. The algorithm iteratively selects candidate features based on a combination of exploitation (retrieving top-scoring nodes) and exploration (sampling unvisited nodes). At each iteration, an LLM agent filters and ranks candidates, proposes missing but relevant features, and performs batch-level self-checks to ensure internal consistency. Accepted candidates are incorporated into the current subtree, and the process continues until a fixed iteration budget is reached. The resulting subtree captures features most relevant to the target repository while balancing coverage and quality.

---

**Algorithm 2** Repository-Specific Subtree Selection

---

**Require:** Global Feature Tree $\mathcal{T}$; Repo description $\mathcal{R}$; iteration budget $K$; top-$k$; termination threshold $\tau$; batch size $B$; LLM
**Ensure:** Repository-specific subtree $\mathcal{T}'$
 1: Initialize current repo tree $\mathcal{T}' \leftarrow \emptyset$; missing features $\mathcal{C}_{\text{missing}} \leftarrow \emptyset$; visited set $\mathcal{V} \leftarrow \emptyset$
 2: **for** $k = 1 \ldots K$ **do** ▷ iterate with given budget
 3: $\quad \mathcal{E}_{\text{exploit}} \leftarrow \text{RETRIEVETOPK}(\mathcal{T}, \mathcal{R}, k = \text{top-}k)$ ▷ select promising nodes (exploit)
 4: $\quad \mathcal{E}_{\text{explore}} \leftarrow \text{SAMPLEUNVISITED}(\mathcal{T}, \mathcal{V})$ ▷ sample unexplored nodes (explore)
 5: $\quad$ **// Candidate selection via LLM**
 6: $\quad \mathcal{C}_{\text{exploit}} \leftarrow \text{LLM.SELECTEXPLOITCANDIDATES}(\mathcal{E}_{\text{exploit}}, \mathcal{T}', \mathcal{R})$ ▷ filter exploit candidates
 7: $\quad \mathcal{C}_{\text{explore}} \leftarrow \text{LLM.SELECTEXPLORECANDIDATES}(\mathcal{E}_{\text{explore}}, \mathcal{T}', \mathcal{R})$ ▷ filter explore candidates
 8: $\quad \mathcal{C}_{\text{missing}} \leftarrow \mathcal{C}_{\text{missing}} \cup \text{LLM.PROPOSEMISSING}(\mathcal{T}', \mathcal{R})$ ▷ generate novel candidates not in tree
 9: $\quad \mathcal{C}_{\text{raw}} \leftarrow \mathcal{C}_{\text{exploit}} \cup \mathcal{C}_{\text{explore}} \cup \mathcal{C}_{\text{missing}}$ ▷ merge all candidate sources
10: $\quad$ **// Batch self-check (filter useful paths within each batch)**
11: $\quad$ **for all** batch $\mathcal{B} \subseteq \mathcal{C}_{\text{raw}}$ with $|\mathcal{B}| \leq B$ **do** ▷ process in small batches
12: $\quad\quad \mathcal{B}^{\star} \leftarrow \text{LLM.SELFCHECK}(\mathcal{T}', \mathcal{B})$ ▷ accept only consistent/relevant paths
13: $\quad\quad \mathcal{T}' \leftarrow \text{INSERTPATHS}(\mathcal{T}', \mathcal{B}^{\star})$ ▷ expand repo-specific tree
14: $\quad\quad \mathcal{V} \leftarrow \mathcal{V} \cup \mathcal{B}$ ▷ mark all evaluated paths as visited
15: $\quad$ **end for**
16: **end for**
17: **return** $\mathcal{T}'$ ▷ return final subtree

---

**Repository Subtree Reorganization into the functionality graph** The algorithm operates in three stages to refactor subtree. In the first stage, an LLM agent iteratively extracts meaningful features from the input, organizing them into subgraphs until sufficient coverage of leaf nodes is reached. In the second stage, the agent reorganizes subgraphs by merging semantically related components or moving branches across groups to improve structure. Finally, each subgraph is refined to ensure naming consistency and hierarchical coherence, yielding a clean, interpretable functionality graph.

## B.2 DETAILED CONSTRUCTION PROCESS

| Level | #Elements | Examples |
|---|---|---|
| 1 | 17 | functionality, data structures, data processing |
| 2 | 1,527 | text processing, process monitoring, create flowchart |
| 3 | 21,739 | heap allocation, dayjs, affine transformation |
| 4 | 113,348 | update record by ID, automated versioning, M5 Model Tree |
| 5 | 613,311 | add vertices, angular velocity, find minimum by group, mark outlier data |
| 6 | 33,801 | min with inclusion, multiple with keyword |
| 7 | 781 | validate against thesaurus, swipe event detection |

Table 7: Statistics of the global feature tree across hierarchical levels, with representative examples from each level.

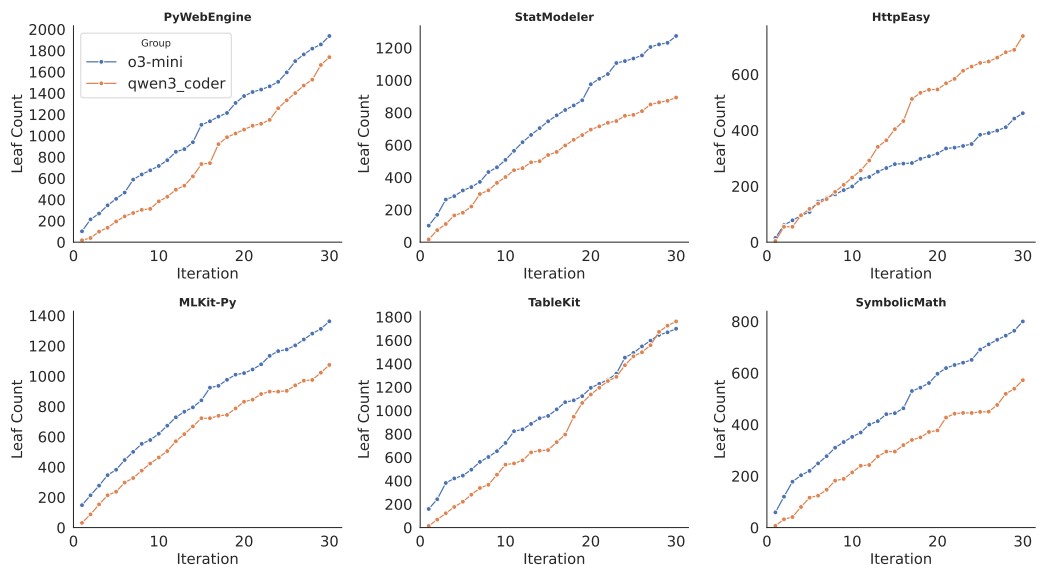

Figure 9: Evolution of Feature Tree Leaf Counts over Iterations Across Repositories, Highlighting the Differences Between qwen3 coder and o3-mini

**Global Feature Tree**   The global feature tree consists of more than one million nodes across seven hierarchical levels (Table 7), reflecting a broad and diverse functional knowledge base. Nevertheless, the distribution of features across Level-1 categories is highly skewed (Figure 10). In particular, the *data processing* branch dominates the tree, while many other categories contain only a small number of nodes, resulting in a pronounced long-tail distribution. Such bias is inherent to real-world software ecosystems, where data processing utilities are disproportionately prevalent compared to specialized functionalities. As a consequence, constructing a repository-specific RPG requires large-scale filtering and reorganiza-

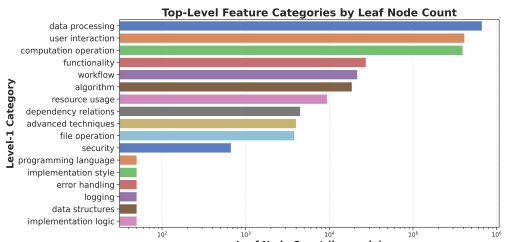

Figure 10: Distribution of feature counts under Level-1 categories in the global feature tree.

tion in order to extract the most relevant features and mitigate the imbalance of the global distribution.

**Model-Specific Growth Patterns**   Beyond the two traces in Fig. 9. Concretely, *qwen3-coder* exhibits the most open expansion, with an approximately linear increase in leaf counts per iteration—maximizing coverage early but with a higher risk of admitting loosely related features. *o3-mini* follows with a moderately aggressive trajectory, striking a balance between breadth and relevance. Together, these curves delineate different points on the recall–precision spectrum of subtree selection strategies that can be matched to repository needs.

**From Global to Repository-Specific Distributions**   The comparison between the global feature tree (Fig.10) and the final repository-specific profiles (Figs.12) highlights the transformative effect of model-guided reorganization. While the global tree is dominated by generic categories such as `data processing` and `user interaction`, the restructured graphs consistently downweight these high-frequency but less discriminative categories and elevate domain-relevant branches to the foreground. This shift effectively counteracts the inherent long-tail bias of the global ontology, redistributing feature density toward categories that better capture repository semantics. As a result, the constructed graphs are not only semantically sharper but also more functionally coherent with respect to the target domain. Between models, *qwen3-coder* favors broad coverage with slower convergence and higher variance, whereas *o3-mini* achieves a more balanced trade-off between

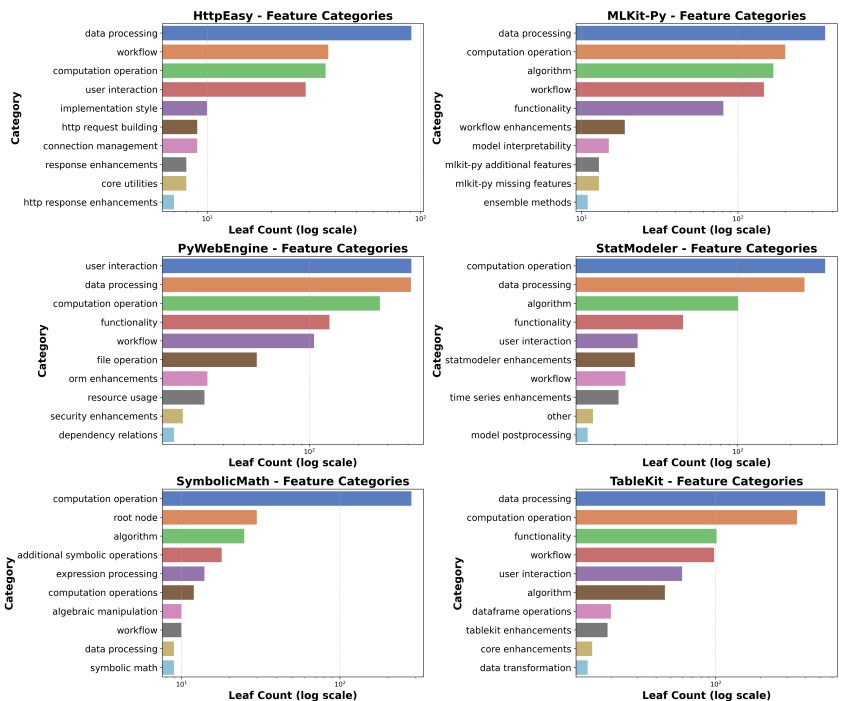

Figure 11: Final distribution of feature counts across subtrees for all repositories under *o3-mini*. The figure shows how features are reorganized after the iterative construction process, reflecting the model's preference in balancing breadth and precision.

generality and specificity. Together, these contrasting tendencies illustrate complementary strategies along the recall–precision spectrum, offering flexibility in matching feature selection to downstream repository needs.

**Final Graph Structures**    The final RPGs (Figure 13a, 13b) reveal how repository-specific functionalities are consolidated into coherent modular organizations. Compared to the more diffuse subtree distributions, the resulting graphs exhibit a markedly skewed allocation of functionalities across subgraphs: a small number of core subgraphs absorb the majority of features, while peripheral subgraphs remain lightweight. This reflects a natural modularization process, where dominant clusters correspond to central repository capabilities and minor clusters capture auxiliary or specialized functions. Between models, the partitioning strategies diverge: *qwen3-coder* produces a larger number of medium-sized subgraphs, favoring breadth and parallel coverage; whereas *o3-mini* yields a more balanced distribution, with several subgraphs of comparable size anchoring distinct semantic roles. These differences indicate that model-driven reorganization not only mitigates the global ontology's long-tail bias but also shapes the granularity of modular decomposition, thereby influencing how functional responsibilities are distributed within the generated graph.

### B.3    PROMPT TEMPLATE

---

**Parts of Prompt Templates for Exploit–Explore Strategy in Subtree Selection**

**Prompt for Exploitation Paths**
You are a GitHub project assistant responsible for expanding a repository's feature tree through path-based modifications to ensure alignment with the project's goals.
In each response, you will be given:
- An Exploit Feature Tree: A curated subset of high-relevance feature paths.
- The Current Repository Feature Tree.
When returning selected paths, always use '"path/to/feature"' format with '/' as the separator.

---

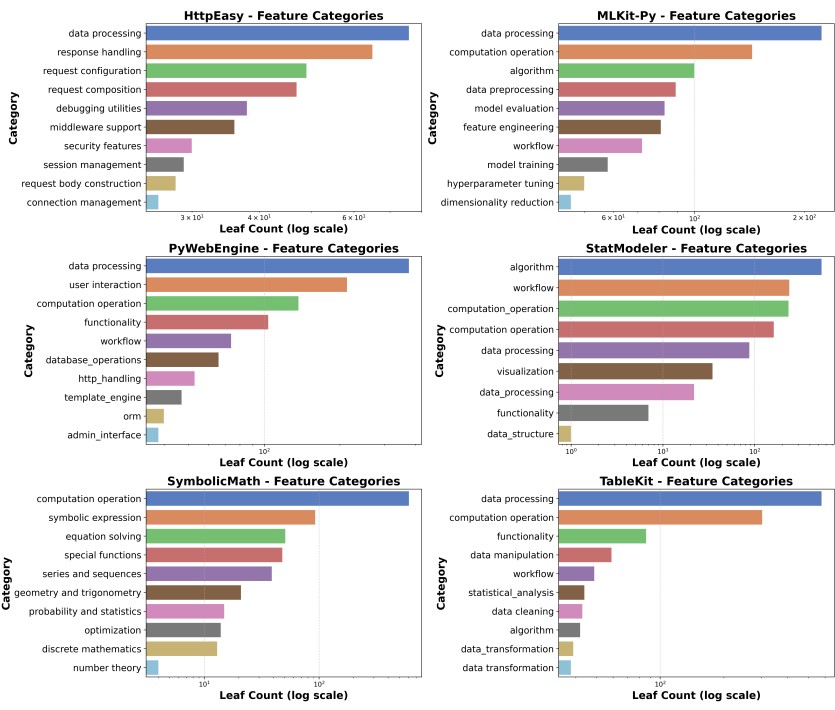

Figure 12: Final distribution of feature counts across subtrees for all repositories under *qwen3-coder*. The figure shows how features are reorganized after the iterative construction process, reflecting the model's preference in balancing breadth and precision.

**Objective (excerpt)**
Expand the Repository Feature Tree so it: 1. Aligns with the repository's purpose and scope.
2. Achieves broad coverage across functional areas.
3. Ensures essential capabilities are represented.
4. Identify and fill critical gaps. . . .
**Selection Principles (excerpt)**
- Select exclusively from the Exploit Feature Tree.
- Include all non-duplicated, useful paths.
- Maintain structural balance by covering underrepresented modules.
. . .
**Exclusions (excerpt)**
Skip generic infra (e.g., logging, configuration) and abstract goals (e.g., "optimize CPU usage").
**Response Format**
Respond only with a Thought and an Action.
<think>
Reason about relevance and gaps in the Exploit Tree. </think>
<action>
{
"all_selected_feature_paths": [
"path/to/feature", ...
]
}
</action>

**Prompt for Exploration Paths**
You are a GitHub project assistant responsible for expanding a repository's feature tree through path-based modifications to ensure alignment with the project's goals.

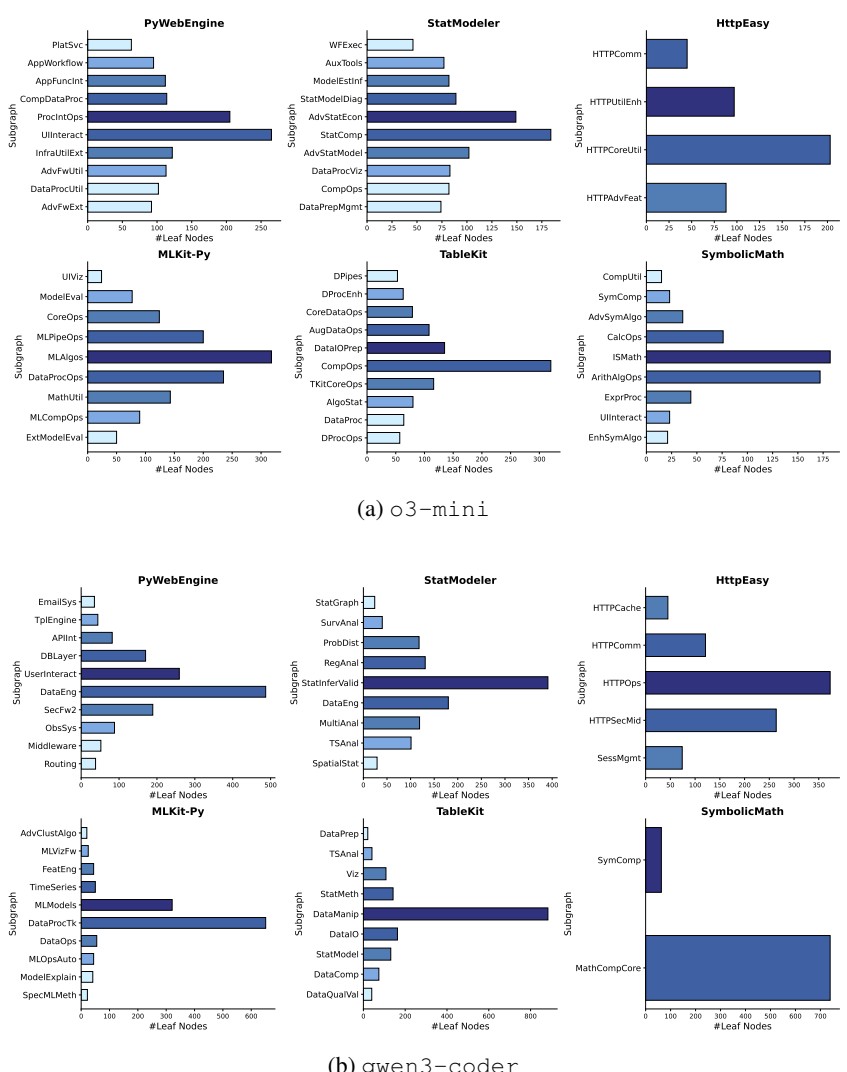

Figure 13: Leaf node counts distribution across feature subgraphs in each repository RPG, reorganized by different models.

In each response, you will be given:
- A Sampled Feature Tree (Exploration Tree).
- The Current Repository Feature Tree.
When returning selected paths, always use '"path/to/feature"' format with '/' as the separator.
**Objective (excerpt)**
Improve and expand the Repository Feature Tree so that it: - Aligns with the repository's purpose and usage scenarios.
- Achieves comprehensive coverage of core and supporting areas.
. . .
**Selection Principles (excerpt)**
- Select only from the Exploration Tree.
- Include actionable, domain-relevant features.
- Skip paths already present in the current Repository Tree.
- Slight over-inclusion is acceptable.
. . .
**Exclusions (excerpt)**

Do not select generic infra (e.g., logging, config) or large-scale features (e.g., cloud integrations).
**Response Format**
Respond only with a single `<think>` and `<action>` block.
`<think>`
Explain how each Exploration Tree path was evaluated and why it was included or excluded.
`</think>`
`<action>`
{ "all_selected_feature_paths": [ "path/to/feature", ... ] }
`</action>`

---

**Parts for Prompt Template for Retrieving Missing Features**

**Instruction** You are a GitHub project assistant tasked with designing a functionally complete, production-grade repository.
Your goal is to identify and recommend **missing functional capabilities or algorithms** that the project should include, based on its real-world purpose, scope, and domain expectations.
Focus on intended functionality — not the existing Feature Tree, which may be incomplete.
**Objective (excerpt)**
Identify groups of functionally concrete features that: 1. Align with the repository's domain and purpose.
2. Are entirely missing or only superficially represented.
3. Are specific and implementable (e.g., functions, classes, modules, algorithms).

**Inclusion Criteria (excerpt)**
- Must be code-level operations (computation, transformation, algorithm, evaluation).
- Realistically implementable within the repository's scope.
- Both standard and advanced algorithms are allowed.

**Exclusion Criteria (excerpt)**
Do not include abstract intentions (e.g., "improve accuracy"), generic infra (e.g., logging, connectors), placeholders, or duplicates.
**Naming Rules (excerpt)**
- Use 3–5 lowercase words, separated by spaces.
- Each leaf node must describe a concrete algorithm or behavior.
- Avoid vague terms, camelCase, or snake_case.

**Structure Guidelines (excerpt)**
- Organize into logical hierarchies (up to 4–5 levels).
- Reflect computational architecture, not documentation taxonomy.
**Response Format** Respond with ONLY a `<think>` and `<action>` block:
`<think>`
Reason about functional domains, workflows, and algorithms that are missing from the current Feature Tree but expected in real-world use.
`</think>`
`<action>`
{ "missing_features": { "root node": { "child node 1": [ "leaf feature 1", "leaf feature 2" ], "child node 2": [ "leaf feature 3", "leaf feature 4" ] } } }
`</action>`

---

## C  APPENDIX OF IMPLEMENTATION-LEVEL GRAPH CONSTRUCTION

This section illustrates how the RPG is enriched with file organization and function design to form concrete code structures.

## C.1 Prompt Template for Implementation-Level Graph Construction

We provide the prompt templates that guide the transformation from graph subtrees into modular code skeletons.

---

**Prompt**

You are a system architect tasked with designing the inter-subtree data flow for a Python software repository. Your goal is to define how data moves between functional modules (subtrees) — including who produces it, who consumes it, and how it is transformed — and express this as a structured, directed graph.

## Data Flow
### Format
[ { "from": "<source subtree name>", "to": "<target subtree name>", "data_id": "<unique name or description of the data>", "data_type": "<type or structure of the data>", "transformation": "<summary of what happens to the data, if anything>" }, ... ]
### Validity & Structural Constraints
2. Full Connectivity Required
- Every subtree listed in {trees_names} must appear in at least one edge.
- No subtree should be isolated or unused.
3. Acyclic Structure
- The data flow must form a Directed Acyclic Graph (DAG):
4. Field Guidelines
- 'data_id': Use unique, descriptive names to identify each data exchange.
- 'data_type': Use precise and interpretable types to describe the structure, format, or abstract role of the data being passed.
- 'transformation': Describe how the data is modified, filtered, enriched, or combined. If unchanged, say '"none"'.
...

## Output Format
<solution>
[ { "from": "...", "to": "...", "data_id": "...", "data_type": "...", "transformation": "..." }, ... ]
</solution>

---

**Parts of Prompt Templates for Raw Skeleton Mapping**

You are a repository architect responsible for designing the initial project structure of a software repository in its early development stage. Your task is to design a clean, modular file system skeleton that organizes the repository into appropriate top-level folders based on these subtrees.

## Requirements
1. The folder structure must clearly separate each functional subtree and reflect logical domain boundaries.
2. Folder names must be concise, meaningful, and follow Python conventions (e.g., 'snake_case'). Names should feel natural and developer-friendly.
3. Folder names do not need to match subtree names exactly.
- Treat subtree names as functional labels.
- Rename folders as needed for clarity and convention, while preserving the correct mapping.
- When assigning a subtree to a folder, include the exact subtree name in the mapping (e.g., '"ml_models": ["Machine Learning"]').
4. You may choose a flat structure (all folders at root level) or a nested structure (e.g., under 'src'), depending on what best supports clarity, organization, and practical use.
5. Include commonly used auxiliary folders as appropriate.
6. The proposed structure should balance clarity, scalability, and maintainability. Avoid

---

unnecessary complexity or excessive nesting.

...

## Output Format
Return a single JSON-style nested object representing the repository folder structure:
- '"folder_name": ["Subtree Name"]' means this folder is assigned to a specific subtree. The name in the list must match exactly what appears in the given list of subtrees.
- '"folder_name": []' means the folder exists but does not correspond to a specific subtree (e.g., utility or support folders).
- '"file_name.ext": null' indicates the presence of a file. File content is not required.

---

**Prompt for Mapping Feature Paths to Python Skeleton Files**

You are a repository architect tasked with incrementally assigning all remaining leaf-level features from a functional subtree into the repository's file structure. This is an iterative process, You are not expected to assign all features at once — each round should make clear, meaningful progress. Your ultimate goal is a production-grade file organization that evolves cleanly and logically over time.
## Context
In each iteration, you will receive:
- A list of unassigned leaf features (each is a full path like '"a/b/c"').
- A designated functional folder under which all new paths must begin.
- A partial skeleton showing the current structure (existing assignments are hidden).
Assign the remaining features to '.py' file paths that:
- Begin with the designated folder.
- Group semantically related features together.
- Reflect how real developers would modularize logic in a production Python codebase. -
Prefer organizing major functional categories into subfolders when appropriate.

### File & Folder Structure
- Group features by functionality into logically meaningful modules that reflect real-world development practice.
- Avoid bundling many unrelated features into a single file
- If a folder contains 10 or more files, introduce subfolders based on semantic structure (e.g., 'format/', 'client/', 'csv/') to keep directories manageable.

### Naming & Organization Guidelines
...
### Examples
...
## Output Format
You must structure your response in two clearly separated blocks, each wrapped with the appropriate tags:
<think>
Explain how you grouped the features into logically coherent modules with clean file and folder structure.
Describe how your choices improve clarity, minimize clutter, and reflect good design principles.
</think>
<solution>
{ "<path to file1.py>": [ "feature1", "feature2" ], "<path to file2.py>": [ "feature3" ] }
</solution>

**Prompt for Converting Subgraphs into Base Classes**

You are an expert software engineer tasked with designing reusable abstractions and shared data structures for a Python codebase.
...

## Base Class Output Format
You must return your design as a set of code blocks grouped by target subtree and file:
## General
### path/to/file.py
```python
...
```

...

## <Subtree Name>
### path/to/file.py
```python
...
```

</solution>

## Design Strategy
Abstractions must follow system structure and dataflow analysis, not mechanical repetition.
- Shared Data Structures: define for nodes with high out-degree (outputs consumed widely). Good candidates are feature batches, inference results, or training containers. Create a global type only when field names, data types, and usage context are stable and consistent.
- Functional Base Classes: define for nodes with high in-degree (consuming many inputs). Use when multiple modules share roles (e.g., cleaning, predicting), follow common lifecycles ('run()', 'build()', 'validate()'), or rely on similar hooks.
- Principles:
- Avoid speculative abstractions.
- Prefer fewer, well-justified classes (typically 1–3 globally).
- Capture structural commonality that aids extensibility and coordination.
...
## Output Formate
Wrap your entire output in two blocks:
<think>
...
</think>
<solution>
## SubtreeA
### path/to/file.py
```python
...
```

...
</solution>

**Prompt for Mapping Feature Paths to Interfaces**

You are designing modular interfaces for a large-scale Python system. You are given repository context: overview, tree structure, skeleton, data flow, base classes, upstream interfaces, target subtree, and target file.
### Objective
- For each feature, define exactly one interface (function or class).
- Provide imports, signature, and detailed docstring (purpose, args, returns, assumptions).

- No implementation: use 'pass'.
- One interface per block.
 ### Design Guidelines
- Function: simple, atomic, stateless.
- Class: stateful, multiple methods, inherits base class, or extensible.
- Prefer fewer, well-justified abstractions.
- Group only tightly related features.
- Use global abstractions sparingly.
### Output Format
Use two blocks:
<think>
reasoning
</think>
<solution>
design_itfs_for_feature(features=["feature/path", ...]):
```python
# One interface (function or class) with docstring and pass
``
</solution>

## C.2 CASE OF BUILT SKELETON AND DESIGNED INTERFACES

We present the skeleton generated by `o3-mini`, together with the mapping between the generated skeleton and the nodes of machine learning algorithms. In addition, we illustrate one or two designed base classes as well as concrete functions or classes.

**Repository Skeleton**

```
src/
  algorithms/
    advanced/
      enhancements/
        general_enhancements/
          __init__.py
          active_learning_strategies.py
          classification_clustering_enhancements.py
          misc_general_enhancements.py
          optimization_and_meta_learning.py
          regression_enhancements.py
        __init__.py
      extended_techniques/
        extended_methods/
          __init__.py
          interpolation_and_model_learning.py
          validation_and_clustering.py
        new_models/
          __init__.py
          new_model_techniques.py
          new_model_techniques_additional.py
        __init__.py
        baselines.py
      supplemental_algorithms/
        __init__.py
        advanced_clustering_and_dimensionality_methods.py
        advanced_tokenization_and_perceptron.py
        classification_and_feature_importance_methods.py
        diverse_algorithmic_methods.py
        ensemble_evaluation_and_anomaly_detection.py
        meta_optimization_methods.py
        model_optimization_methods.py
        numerical_interpolation_methods.py
        regression_and_svm_optimization_methods.py
        spline_interpolation_and_adjusted_classifiers.py
        svm_ensemble_and_optimization_methods.py
        tokenization_methods.py
      __init__.py
    ensemble_tree/
```

```
    boosting_bagging/
      boosting/
        __init__.py
        boosting_advanced_features.py
        boosting_algorithms.py
        boosting_parameter_tuning.py
      stacking_voting/
        __init__.py
        primary.py
        secondary.py
      __init__.py
      bagging.py
      gradient_boosting.py
    decision_trees/
      __init__.py
      gradient_boosting_tree.py
      id3.py
      post_pruning.py
      random_forest.py
      regression_tree.py
    __init__.py
  regression/
    linear_models/
      __init__.py
      lasso.py
      multiple_linear.py
      polynomial.py
      simple_linear.py
    __init__.py
    elastic_net_regression.py
    ridge_classifier.py
  supervised/
    classification/
      logistic/
        __init__.py
        cost.py
        optimization.py
        sigmoid.py
      __init__.py
      decision_tree.py
      knearest.py
      naive_bayes.py
      support_vector.py
    __init__.py
  unsupervised/
    clustering/
      __init__.py
      advanced_clustering.py
      kmeans.py
      supplemental_clustering.py
    dimensionality_reduction/
      __init__.py
      extended_dr.py
      kernel_pca.py
      pca.py
    __init__.py
  __init__.py
core/
  data_conversion/
    __init__.py
    api_requests.py
    feature_encoding.py
    feature_extraction.py
    format_conversion.py
    sql_queries.py
  data_transform/
    __init__.py
    filter_advanced.py
    filter_basic.py
    join_operations.py
    scaling_advanced.py
    scaling_basic.py
    sorting.py
    splitting.py
  numerics/
    __init__.py
    basic_statistics.py
    block_multiplication.py
    decompositions.py
```

```
        dot_products.py
        integration_and_distances.py
        inversions.py
        matrix_factorization.py
        matrix_rearrangements.py
        regression_statistics.py
        sparse_lu.py
      preprocessing/
        __init__.py
        csv_io.py
        data_cleaning.py
        dimensionality_analysis.py
        inverse_transformations.py
        json_io.py
        log_transformations.py
        noise_augmentation.py
      __init__.py
  data_processing/
    analysis_pipeline/
      analytical/
        __init__.py
        aggregation_algorithms.py
        data_perturbation.py
        data_query.py
        join_operations.py
        list_manipulation.py
        sample_partition.py
        seasonal_analysis.py
      pipeline_utilities/
        __init__.py
        data_streaming.py
        learning_setup.py
        model_validation.py
        performance_metrics.py
      __init__.py
    cleaning_preparation/
      advanced/
        __init__.py
        duplicate_handling.py
        imputation_methods.py
        outlier_detection.py
      preparation/
        __init__.py
        data_splitting.py
        imputation_labeling.py
        validation.py
      __init__.py
      type_conversion.py
    integration_merge/
      __init__.py
      aggregation_retrieval.py
      merge_operations.py
      merge_search.py
    integration_storage/
      __init__.py
      api_operations.py
      conversion_auth.py
      dictionary_config.py
      export_integration.py
      io_logging.py
    manipulation/
      __init__.py
      data_manipulation.py
      shuffling.py
    string_pivot/
      __init__.py
      pivoting.py
      string_operations.py
    transformation_feature_eng/
      __init__.py
      feature_extraction_encoding.py
      file_io.py
      text_enhancements.py
      transformation_normalization.py
    utilities/
      __init__.py
      metadata.py
      metrics.py
      parallel.py
```

```
      sparse_storage.py
      text_processing.py
    validation/
      inspection/
        __init__.py
        overview.py
        sorting.py
        statistics.py
      __init__.py
      data_integrity.py
    __init__.py
extended_eval/
  diagnostics/
    __init__.py
    model_quality.py
    statistical_diagnostics.py
    temporal_analysis.py
  __init__.py
  explainability.py
  predictive_assessment.py
  robustness.py
math_utils/
  algorithms/
    core_techniques/
      __init__.py
      clustering_and_detection.py
      fairness_and_feature_analysis.py
      matrix_operations.py
      optimization_and_selection.py
      statistical_methods.py
    __init__.py
  auxiliary/
    __init__.py
    data_manipulation.py
    geometric_operations.py
    gradient_and_imaging.py
    math_computations.py
    ml_utilities.py
    optimization_methods.py
    outlier_validation.py
    random_operations.py
    tensor_and_likelihood.py
    text_processing.py
    time_processing.py
  data_preprocessing/
    __init__.py
    model_persistence.py
    sampling.py
    text_tools.py
  performance/
    __init__.py
    drift_detection.py
    statistical_tests.py
    system_monitoring.py
    time_series_analysis.py
  pipeline_evaluation/
    __init__.py
    advanced_analysis.py
    data_resampling.py
    evaluation_metrics.py
    hyperparameter_tuning.py
    model_export_and_cv.py
    online_learning_support.py
    performance_benchmarking.py
    pipeline_creator.py
  simulation/
    __init__.py
    hyperparameter_tuning.py
    nearest_neighbor_search.py
    random_sampling.py
    simulation.py
    time_interpolation.py
  statistical_analysis/
    __init__.py
    descriptive_stats.py
    inferential_methods.py
    multivariate_analysis.py
    probabilistic_models.py
    survival_analysis.py
```

```
        time_series_models.py
        variance_metrics.py
      __init__.py
ml_compute/
  ml_methods/
      __init__.py
      clustering_methods.py
      dimensionality_reduction.py
      svm_validation.py
  optimization/
      __init__.py
      clustering_graph_techniques.py
      optimization_algorithms.py
      training_control.py
  stat_inference/
      __init__.py
      hypothesis_tests_advanced.py
      hypothesis_tests_basic.py
      model_evaluation_metrics.py
      model_selection_metrics.py
      parameter_estimation.py
      statistical_tests_metrics.py
  time_series/
      distribution/
          __init__.py
          bayesian_likelihood.py
          distribution_estimation.py
          inferential_hypothesis.py
          inferential_variance.py
      __init__.py
      forecast_plots.py
      forecasting_algorithms.py
  __init__.py
model_eval/
  deployment/
      __init__.py
      deployment_ops.py
      testing_resources.py
  evaluation/
      diagnostics/
          __init__.py
          advanced_diagnostics.py
          basic_diagnostics.py
          performance_metrics.py
      __init__.py
      additional_analysis.py
      error_display.py
      regression_diagnostics.py
      training_ops.py
  management/
      __init__.py
      counterfactual.py
      feature_explanation.py
      hyperparameter.py
      persistence_ops.py
      pipeline_integration.py
      uncertainty.py
  visualization/
      __init__.py
      dashboard.py
      visual_reports.py
  __init__.py
pipeline/
  data_cleaning/
      __init__.py
      dimensionality.py
      encoding.py
      filtering.py
      imputation.py
      knn_methods.py
      merging.py
  deployment/
      __init__.py
      export.py
      integration_testing.py
      interactive.py
      monitoring.py
      online.py
      reporting_formats.py
```

```
        standard_plots.py
        trend_analysis.py
      evaluation/
        __init__.py
        cross_validation.py
        gd_optimization.py
        metrics_plots.py
        misc_evaluation.py
        monitoring.py
      feature_engineering/
        __init__.py
        basic_transformation.py
        extraction.py
        interaction.py
        selection.py
        synthesis.py
      orchestration/
        __init__.py
        configuration.py
        startup.py
        transformers.py
        workflow.py
      preprocessing/
        __init__.py
        advanced_parsing.py
        imputation.py
        input_validation.py
      training/
        __init__.py
        training_adjustments.py
        training_strategies.py
      tuning/
        __init__.py
        calibration.py
        evolution.py
        gaussian.py
        meta.py
        parzen.py
      __init__.py
  ui/
    interactivity/
      __init__.py
      domain_commands.py
      general_commands.py
      help_support.py
      navigation_actions.py
    visualization/
      __init__.py
      dashboard.py
      standard_charts.py
    __init__.py
  __init__.py
main.py
setup.py
```

## SubGraph-to-Skeleton

```
Machine Learning Algorithms
    AdvancedExtensions [-> dir: src/algorithms/advanced]
        Miscellaneous [-> dir: src/algorithms/advanced/supplemental_algorithms]
            OtherAlgorithms [-> file: src/algorithms/advanced/supplemental_algorithms/
                classification_and_feature_importance_methods.py, file: src/algorithms/
                advanced/supplemental_algorithms/
                regression_and_svm_optimization_methods.py, file: src/algorithms/
                advanced/supplemental_algorithms/
                advanced_clustering_and_dimensionality_methods.py, file: src/algorithms
                /advanced/supplemental_algorithms/meta_optimization_methods.py, file:
                src/algorithms/advanced/supplemental_algorithms/
                svm_ensemble_and_optimization_methods.py, file: src/algorithms/advanced
                /supplemental_algorithms/spline_interpolation_and_adjusted_classifiers.
                py, file: src/algorithms/advanced/supplemental_algorithms/
                numerical_interpolation_methods.py, file: src/algorithms/advanced/
                supplemental_algorithms/diverse_algorithmic_methods.py, file: src/
                algorithms/advanced/supplemental_algorithms/
                advanced_tokenization_and_perceptron.py, file: src/algorithms/advanced/
                supplemental_algorithms/ensemble_evaluation_and_anomaly_detection.py,
```

```
                    file: src/algorithms/advanced/supplemental_algorithms/
                    tokenization_methods.py, file: src/algorithms/advanced/
                    supplemental_algorithms/model_optimization_methods.py]
        ExtendedTechniques [-> dir: src/algorithms/advanced/extended_techniques]
                ExtendedMethods [-> file: src/algorithms/advanced/extended_techniques/
                    extended_methods/interpolation_and_model_learning.py, file: src/
                    algorithms/advanced/extended_techniques/extended_methods/
                    validation_and_clustering.py]
                NewModels [-> file: src/algorithms/advanced/extended_techniques/new_models/
                    new_model_techniques.py, file: src/algorithms/advanced/
                    extended_techniques/new_models/new_model_techniques_additional.py]
                Baselines [-> file: src/algorithms/advanced/extended_techniques/baselines.py
                    ]
        EnhancementsAndGeneral [-> dir: src/algorithms/advanced/enhancements/
             general_enhancements]
                GeneralEnhancements [-> file: src/algorithms/advanced/enhancements/
                    general_enhancements/optimization_and_meta_learning.py, file: src/
                    algorithms/advanced/enhancements/general_enhancements/
                    regression_enhancements.py, file: src/algorithms/advanced/enhancements/
                    general_enhancements/active_learning_strategies.py, file: src/
                    algorithms/advanced/enhancements/general_enhancements/
                    misc_general_enhancements.py, file: src/algorithms/advanced/
                    enhancements/general_enhancements/
                    classification_clustering_enhancements.py]
    Regression [-> dir: src/algorithms/regression/linear_models]
        LinearModels [-> dir: src/algorithms/regression/linear_models]
                MultipleLinear [-> file: src/algorithms/regression/linear_models/
                    multiple_linear.py, file: src/algorithms/regression/linear_models/lasso
                    .py]
                PolynomialRegression [-> file: src/algorithms/regression/linear_models/
                    polynomial.py]
                Lasso [-> file: src/algorithms/regression/linear_models/multiple_linear.py,
                    file: src/algorithms/regression/linear_models/lasso.py]
                SimpleLinear [-> file: src/algorithms/regression/linear_models/simple_linear
                    .py]
        OtherRegression [-> dir: src/algorithms/regression]
                RidgeRegressionClassification [-> file: src/algorithms/regression/
                    ridge_classifier.py]
                ElasticNet [-> file: src/algorithms/regression/elastic_net_regression.py]
    UnsupervisedLearning [-> dir: src/algorithms/unsupervised]
        DimensionalityReduction [-> dir: src/algorithms/unsupervised/
             dimensionality_reduction]
                KernelMethods [-> file: src/algorithms/unsupervised/dimensionality_reduction
                    /kernel_pca.py]
                OtherDR [-> file: src/algorithms/unsupervised/dimensionality_reduction/
                    extended_dr.py]
                PCA [-> file: src/algorithms/unsupervised/dimensionality_reduction/pca.py]
        Clustering [-> dir: src/algorithms/unsupervised/clustering]
                KMeans [-> file: src/algorithms/unsupervised/clustering/kmeans.py]
                OtherClustering [-> file: src/algorithms/unsupervised/clustering/
                    supplemental_clustering.py]
                AdvancedClustering [-> file: src/algorithms/unsupervised/clustering/
                    advanced_clustering.py]
    EnsembleAndTreeMethods [-> dir: src/algorithms/ensemble_tree/boosting_bagging]
        BoostingBagging [-> dir: src/algorithms/ensemble_tree/boosting_bagging]
                StackingVoting [-> file: src/algorithms/ensemble_tree/boosting_bagging/
                    stacking_voting/secondary.py, file: src/algorithms/ensemble_tree/
                    boosting_bagging/stacking_voting/primary.py]
                Bagging [-> file: src/algorithms/ensemble_tree/boosting_bagging/bagging.py]
                GradientBoosting [-> file: src/algorithms/ensemble_tree/boosting_bagging/
                    gradient_boosting.py]
                Boosting [-> file: src/algorithms/ensemble_tree/boosting_bagging/boosting/
                    boosting_algorithms.py, file: src/algorithms/ensemble_tree/
                    boosting_bagging/boosting/boosting_advanced_features.py, file: src/
                    algorithms/ensemble_tree/decision_trees/gradient_boosting_tree.py, file
                    : src/algorithms/ensemble_tree/boosting_bagging/boosting/
                    boosting_parameter_tuning.py]
        DecisionTrees [-> dir: src/algorithms/ensemble_tree/decision_trees]
                ID3 [-> file: src/algorithms/ensemble_tree/decision_trees/id3.py]
                RegressionTree [-> file: src/algorithms/ensemble_tree/decision_trees/
                    regression_tree.py]
                GradientBoostingTree [-> file: src/algorithms/ensemble_tree/boosting_bagging
                    /boosting/boosting_advanced_features.py, file: src/algorithms/
                    ensemble_tree/decision_trees/gradient_boosting_tree.py]
                PostPruning [-> file: src/algorithms/ensemble_tree/decision_trees/
                    post_pruning.py]
                RandomForest [-> file: src/algorithms/ensemble_tree/decision_trees/
                    random_forest.py]
    SupervisedLearning [-> dir: src/algorithms/supervised/classification/logistic]
```

```
        Classification [-> dir: src/algorithms/supervised/classification/logistic]
            LogisticRegression [-> file: src/algorithms/supervised/classification/
                logistic/sigmoid.py, file: src/algorithms/supervised/classification/
                logistic/optimization.py, file: src/algorithms/supervised/
                classification/logistic/cost.py]
            SupportAndTree [-> file: src/algorithms/supervised/classification/
                decision_tree.py, file: src/algorithms/supervised/classification/
                support_vector.py]
            NaiveBayes [-> file: src/algorithms/supervised/classification/naive_bayes.py
                ]
            KNearest [-> file: src/algorithms/supervised/classification/knearest.py]
```

## Base Classes Example

```python
## General
### src/general/base_components.py
```python
from abc import ABC, abstractmethod

class BaseComponent(ABC):
    """
    Abstract base class for core pipeline components.

    Provides a standard lifecycle:
      - initialize(): Prepare the component.
      - process(data): Process input data.
      - finalize(): Clean up resources.

    Concrete subclasses must implement process().
    """

    def __init__(self) -> None:
        self.initialized: bool = False

    def initialize(self) -> None:
        """
        Set up the component before processing starts.
        """
        self.initialized = True

    @abstractmethod
    def process(self, data):
        """
        Process the input data.

        Args:
            data: Input data in a predefined schema.

        Returns:
            Output data after processing.
        """
        pass

    def finalize(self) -> None:
        """
        Tear down or clean up the component after processing.
        """
        self.initialized = False

class EstimatorComponent(BaseComponent):
    """
    Abstract base class for estimator components (e.g., models).

    Defines the contract for model training and prediction.
    """

    @abstractmethod
    def fit(self, training_data) -> None:
        """
        Train the estimator using the provided training data.

        Args:
            training_data: Data batch conforming to a shared TrainingBatch schema.
        """
        pass
```

```
    @abstractmethod
    def predict(self, input_data):
        """
        Generate predictions based on the input data.

        Args:
            input_data: Data in a format specified by the pipeline.

        Returns:
            Predictions corresponding to the input features.
        """
        pass

    def process(self, data):
        """
        For an estimator, process() defaults to prediction.
        """
        return self.predict(data)
```

**Designed Interfaces**

```
## Feature Paths: "Regression/LinearModels/PolynomialRegression/Cubic Regression with
    regularization", "Regression/LinearModels/PolynomialRegression/polynomial model
    fitting", "Regression/LinearModels/PolynomialRegression/cubic regression", "
    Regression/LinearModels/PolynomialRegression/quadratic regression", "Regression/
    LinearModels/PolynomialRegression/Quadratic Regression with regularization"
### src/algorithms/regression/linear_models/polynomial.py
from src.general.base_components import EstimatorComponent
class PolynomialRegressor(EstimatorComponent):

    def __init__(self, degree: int, regularization_lambda: float=0.0) -> None:
        pass

    def fit(self, X: list[float], y: list[float]) -> None:
        """
        Fit the polynomial regression model to the provided data.

        Constructs the polynomial features based on the specified degree and applies
        optional regularization if regularization_lambda is provided (> 0).

        Args:
            X (list[float]): A list of feature values.
            y (list[float]): A list of target values corresponding to the features.

        Returns:
            None

        Raises:
            ValueError: If the degree is not supported or if input lists are empty or
                mismatched.
        """
        pass

    def predict(self, X: list[float]) -> list[float]:
        """
        Generate predictions using the fitted polynomial regression model.

        Transforms the input features into polynomial features and computes
        the output via the fitted model coefficients. Applies regularization adjustments
        if the model was fitted with a regularization term.

        Args:
            X (list[float]): A list of feature values for prediction.

        Returns:
            list[float]: A list of predicted values.
        """
        pass
```

### C.3 PATTERNS IN IMPLEMENTATION-LEVEL GRAPH CONSTRUCTION

The mapping from RPGs to code structures exhibits a strong isomorphic relationship: each subgraph corresponds to a coherent code region, with files, classes, and functions serving as structural anchors. Table 9 illustrates this correspondence for the case of *o3-mini* during `sklearn` generation, where algorithmic subgraphs (e.g., ML Algorithms, Data Processing, ML Pipeline) map to a larger number of files and functions, while auxiliary subgraphs (e.g., Diagnostics, Visualization) remain compact yet feature-dense. This pattern reflects the semantic granularity of different subgraphs: core computational domains require broader structural scaffolding, whereas specialized domains concentrate more features per unit. Extending to the cross-repository view in Table 8, we observe that both models preserve this structural isomorphism but with distinct emphases: *o3-mini* tends to distribute features more evenly across units, while *qwen3-coder* consistently produces the highest feature densities, especially at the class level. Together, these results demonstrate that the graph-to-code translation process not only preserves the hierarchical semantics of the RPG but also manifests in distinct structural footprints that vary with model choice.

Table 8: Per-repository structural statistics across o3-mini and qwen-coder. "Count" = number of entities (Files/Classes/Functions) per repository; "Avg Feat." = mean number of features per entity (features per file/class/function).

| Repo | o3-mini | | | | | | qwen-coder | | | | | |
|---|---|---|---|---|---|---|---|---|---|---|---|---|
| | Files | | Classes | | Functions | | Files | | Classes | | Functions | |
| | Count | Avg Feat. | Count | Avg Feat. | Count | Avg Feat. | Count | Avg Feat. | Count | Avg Feat. | Count | Avg Feat. |
| TableKit | 475 | 3.64 | 252 | 2.28 | 1092 | 1.05 | 271 | 6.69 | 496 | 1.62 | 587 | 1.50 |
| MLKit-Py | 266 | 4.74 | 321 | 1.64 | 708 | 1.04 | 566 | 2.44 | 815 | 1.30 | 281 | 1.13 |
| StatModeler | 219 | 4.71 | 117 | 2.47 | 726 | 1.02 | 330 | 3.48 | 573 | 1.22 | 411 | 1.07 |
| SymbolicMath | 126 | 4.70 | 95 | 2.17 | 370 | 1.04 | 89 | 8.98 | 71 | 1.73 | 786 | 0.86 |
| PyWebEngine | 440 | 3.89 | 576 | 1.74 | 689 | 1.02 | 482 | 3.52 | 890 | 1.26 | 501 | 1.13 |
| HttpEasy | 104 | 4.17 | 79 | 2.43 | 235 | 1.03 | 178 | 4.28 | 239 | 1.52 | 366 | 1.06 |
| **Average** | 271.7 | 4.31 | 240 | 2.12 | 636.7 | 1.03 | 319.3 | 4.90 | 514 | 1.44 | 488.7 | 1.12 |

Table 9: Structural distribution of files, classes, functions, and feature densities corresponding to each **subgraph in the feature graph of o3-mini during sklearn synthesis**. Here, "Files/Classes/Functions" denote the number of code units mapped from each subgraph; "File/Class/Function Features" are the total extracted features; and "Avg Features/..." indicates the average number of features per unit type.

| Subgraph | Files | Classes | Functions | File Features | Class Features | Function Features | Avg Feat./File | Avg Feat./Class | Avg Feat./Func |
|---|---|---|---|---|---|---|---|---|---|
| ML Algorithms | 58 | 171 | 67 | 323 | 256 | 67 | 5.57 | 1.50 | 1.00 |
| Math Utilities | 47 | 26 | 102 | 143 | 40 | 103 | 3.04 | 1.54 | 1.01 |
| Data Processing | 45 | 30 | 169 | 231 | 59 | 172 | 5.13 | 1.97 | 1.02 |
| ML Pipeline | 38 | 23 | 149 | 202 | 42 | 160 | 5.32 | 1.83 | 1.07 |
| Core Operations | 30 | 15 | 76 | 124 | 33 | 91 | 4.13 | 2.20 | 1.20 |
| ML Computation | 19 | 39 | 34 | 88 | 54 | 34 | 4.63 | 1.38 | 1.00 |
| ML Evaluation | 17 | 12 | 51 | 77 | 25 | 52 | 4.53 | 2.08 | 1.02 |
| ML Diagnostics | 6 | 3 | 44 | 50 | 6 | 44 | 8.33 | 2.00 | 1.00 |
| Visualization | 6 | 2 | 16 | 24 | 8 | 16 | 4.00 | 4.00 | 1.00 |

## D APPENDIX OF GRAPH-GUIDED REPOSITORY GENERATION

### D.1 DETAILS ON LOCALIZATION

To facilitate the localization stage in graph-guided repository generation, we designed a graph-guided toolset that allows agents to systematically explore and map design-level features onto concrete code artifacts. The tools support both fine-grained inspection of files and interfaces, as well as feature-driven exploration across the repository. Specifically, `view_file_interface_feature_map` and `get_interface_content` enable inspection of code structures and retrieval of their implementations, while `expand_leaf_node_info` and `search_interface_by_functionality` allow navigation of the RPG and fuzzy semantic search. Finally, the `Terminate` command ensures that the localization process produces a ranked and standardized output. Together, these tools provide a structured workflow that balances automation with flexibility, ensuring both accuracy and interpretability in the localization process.

---

**Localization Tools**

```
### Interface Inspection Tools
- `view_file_interface_feature_map(file_path)`
  Inspects a single Python file to list the interface structures (functions, classes,
      methods) it contains, along with the feature mappings they support.
  *Usage*: Useful for quickly understanding which interfaces exist in a given file and
      the feature tags associated with them.
  *Example*:
  \begin{verbatim}
  view_file_interface_feature_map('src/algorithms/classifier.py')
  \end{verbatim}

- `get_interface_content(target_specs)`
  Retrieves the full implementation code of a specific function, class, or method, given
       its fully qualified name (file path + entity name).
  *Usage*: Applied when a particular interface has been located and its source code
      needs to be examined in detail.
  *Example*:
  \begin{verbatim}
  get_interface_content(['src/core/data_loader.py:DataLoader.load_data'])
  get_interface_content(['src/core/utils.py:clean_text'])
  \end{verbatim}

### Feature-Driven Exploration Tools
- `expand_leaf_node_info(feature_path)`
  Given a feature path from the implemented feature tree, this tool expands and lists
      all associated interfaces (functions or classes) in a structural summary.
  *Usage*: Applied when analyzing how a specific functional leaf node in the design tree
       maps to repository interfaces.
  *Example*:
  \begin{verbatim}
  expand_leaf_node_info('Algorithm/Supervised Learning/Classification Algorithms/Naive
      Bayes')
  \end{verbatim}

- `search_interface_by_functionality(keywords)`
  Performs a fuzzy semantic search for interfaces based on given keywords and returns
      the top-5 most relevant interface implementations.
  *Usage*: Useful when the exact file or interface name is unknown, but functionality-
      related keywords are available.
  *Example*:
  \begin{verbatim}
  search_interface_by_functionality(['optimize', 'initialize'])
  \end{verbatim}

### Termination Tool
- `Terminate(result)`
  Terminates the localization exploration and returns the final ranked list of located
      interfaces. The result must follow the specified JSON-style format, including
      file path and interface type (function, class, or method).
  *Usage*: Invoked after completing exploration to deliver the final interface
      localization results.
  *Example*:
  \begin{verbatim}
  Terminate(result=[
    {"file_path": "top1_file_fullpath.py", "interface": "method: Class1.function1"},
    {"file_path": "top2_file_fullpath.py", "interface": "function: function2"},
    {"file_path": "top3_file_fullpath.py", "interface": "class: Class3"},
  ])
  \end{verbatim}
```

## D.2 TOOLS FOR CODING

To support systematic repository modification, we introduce a suite of editing tools that operate at multiple levels of code granularity.

**Editing Coding Tools**

```
### edit_whole_class_in_file(file_path, class_name)
```

```
Use when: An entire class (including all methods and its docstring) requires editing or
     replacement.
Output must: Provide the full class definition, with all methods and docstring.

### edit_method_of_class_in_file(file_path, class_name, method_name)
Use when: A single method inside a class needs to be edited or replaced.
Output must:
- Return the full `class ClassName:` block containing only the target method.
- Exclude all unrelated methods.
- Do not output the method alone; it must appear within its class block.

### edit_function_in_file(file_path, function_name)
Use when: A top-level function requires editing or replacement.
Output must: Provide the full function, including signature, logic, and docstring.

### edit_imports_and_assignments_in_file(file_path)
Use when: Import statements or top-level assignments in the file need to be added or
     corrected.
Output must:
- Contain only import statements and top-level assignments (no functions or classes).
- Follow a strict import order: (1) standard library, (2) third-party packages, (3)
     local modules.
Important:
- Do not remove existing imports unless they are demonstrably incorrect (e.g., invalid
     path, typo, or non-existent module).
- Retain imports even if they appear unused, to preserve runtime dependencies.

### Terminate()
Use when: All required edits are complete and no further tool invocations are necessary.
Important: Do not call prematurely; ensure the issue is fully resolved before
     termination.
```

## D.3   GRAPH-GUIDED LOCALIZATION AND EDITING LOGS

To illustrate how the proposed system operates in practice, we include representative logs that demonstrate the end-to-end process: (i) locating relevant interfaces in the repository through RPG–guided exploration, and (ii) applying code editing operations to implement the required functionality. These logs capture the step-by-step decision process, tool invocations, and final termination results.

---

**Task Example Trajectory**

```
Task:  This issue is to implement the AdvancedDifferentialTransformer class in src/
     symbolic/enhanced/calculus_summation/differential.py. Contributors should aim to
     encapsulate the necessary transformation logic and ensure the methods integrate
     well with the rest of the symbolic algebra framework. Clear error handling and
     input validation are expected in order to gracefully manage undefined behaviors.

               ============Localization=================
Agent: <think>
I will begin by exploring the file "src/symbolic/enhanced/calculus_summation/
     differential.py" since the AdvancedDifferentialTransformer class should be
     implemented here. I will use the view_file_interface_feature_map tool on that file
     to inspect its structure and get an idea of what is already provided, and any hints
      or TODOs might be present.
</think>
<solution>
```
view_file_interface_feature_map('src/symbolic/enhanced/calculus_summation/differential.
     py')
```
</solution>

Env: ```python
## src/symbolic/enhanced/calculus_summation/differential.py
from typing import Any, Dict
### Features: Calculus & Advanced Summation/Differential & Asymptotic Methods/Advanced
     Transformation Methods/asymptotic series expansion, Calculus & Advanced Summation/
     Differential & Asymptotic Methods/Advanced Transformation Methods/differentiate
     special functions, Calculus & Advanced Summation/Differential & Asymptotic Methods/
     Advanced Transformation Methods/simplify inequality expressions
class AdvancedDifferentialTransformer:
```

```
        def asymptotic_series_expansion(self, expression: Any, expansion_order: int) -> Any:
            ...

        def differentiate_special_function(self, expression: Any, function_name: str,
            additional_params: Dict[str, Any]=None) -> Any:
            ...

        def simplify_inequality(self, inequality_expr: Any, tolerance: float=1e-05) -> Any:
            ...
```

Agent: <think>
I'll now check the file "src/symbolic/enhanced/simplification/algebraic.py" since it is
    part of the Enhanced Symbolic Algorithms simplification area, and it might contain
    functions or methods relevant to simplifying inequalities.
</think>
<solution>
```
view_file_interface_feature_map('src/symbolic/enhanced/simplification/algebraic.py')
```
</solution>

Env: ```python
## src/symbolic/enhanced/simplification/algebraic.py
from typing import Any
from src.expr_transform.simplification.techniques import
    apply_commutative_associative_rewriting, order_terms
from src.symbolic.core.expression_manipulation.tree_utils import collect_like_terms
from src.symbolic.core.expression_manipulation.rewriting import
    canonicalize_expression_tree
### Features: Simplification/Algebraic Simplification/Basic Techniques/simplify
    expressions
def simplify_expression(expr: Any) -> Any:
    ...
```

Agent: <think>
I will now search for interfaces related to "differentiate special function" or advanced
     symbolic differentiation since our method differentiate_special_function might
    rely on those routines. This might reveal supporting functions or logic that can be
     reused.
</think>
<solution>
```
search_interface_by_functionality(['differentiate special function', 'advanced symbolic
    differentiation'])
```
</solution>

Env:  Keyword "differentiate special function" matched interfaces: class
    AdvancedExpressionAnalyzer, class AutomaticDifferentiator, class
    MultivariableDifferentiator.
Keyword "advanced symbolic differentiation" matched interfaces: class
    AutomaticDifferentiator, class IntegerAdditionOperations, class
    MultivariableDifferentiator.

## File: src/calculus/differentiation/partial_derivatives.py

from typing import Any, List, Callable

### Features: Differentiation/Specialized/Techniques/validate partial derivatives,
    Differentiation/Specialized/Techniques/compute partial derivatives, Differentiation
    /Specialized/Techniques/symbolic gradient computation, Differentiation/Specialized/
    Techniques/Jacobian matrix, Differentiation/Specialized/Techniques/symbolic hessian
     computation, Differentiation/Specialized/Techniques/multivariable differentiation
class MultivariableDifferentiator:

    def validate_partial_derivatives(self, function: Callable[..., Any], variables: List
        [str], computed: Any) -> bool:
        ...

    def compute_partial_derivative(self, function: Any, variable: str) -> Any:
        """
     ...

    def compute_symbolic_gradient(self, function: Any, variables: List[str]) -> Any:
        ...
```

```
    def compute_jacobian_matrix(self, functions: List[Any], variables: List[str]) -> Any
        :
        ...

    def compute_symbolic_hessian(self, function: Any, variables: List[str]) -> Any:
        ...

    def differentiate_multivariable(self, function: Any, variables: List[str]) -> Any:
        ...
...

Agent: <think>
Next, I'll search for interfaces that mention "asymptotic series expansion" because that
    functionality is required in AdvancedDifferentialTransformer.
    asymptotic_series_expansion. This might reveal candidate functions that can be
    reused to generate series expansions.

I'll also search for interfaces related to "simplify inequality" to see if there's any
    existing logic for inequality simplification that we can leverage.

Let's perform both searches.
</think>
<solution>
```
search_interface_by_functionality(['asymptotic series expansion'])
search_interface_by_functionality(['simplify inequality'])
```
</solution>

Env: Keyword "asymptotic series expansion" matched interfaces: class
    AdvancedExpressionAnalyzer, class MultivariableDifferentiator, class
    NumericalIntegrator.

## File: src/calculus/differentiation/partial_derivatives.py

from typing import Any, List, Callable

### Features: Differentiation/Specialized/Techniques/validate partial derivatives,
    Differentiation/Specialized/Techniques/compute partial derivatives, Differentiation
    /Specialized/Techniques/symbolic gradient computation, Differentiation/Specialized/
    Techniques/Jacobian matrix, Differentiation/Specialized/Techniques/symbolic hessian
     computation, Differentiation/Specialized/Techniques/multivariable differentiation
class MultivariableDifferentiator:
    ...

Agent: <think>
We have gathered several candidate supporting interfaces needed for our implementation
    of AdvancedDifferentialTransformer. Our analysis indicates that the following
    components are very likely to be relevant dependencies:
...

These are our top-5 interface-level code units that are most likely to serve as
    essential dependencies for implementing the class AdvancedDifferentialTransformer
    with its three key methods.

</think>
<solution>
```
Terminate(result=[
  {"file_path": "src/symbolic/core/calculus/differentiation.py", "interface": "function:
      differentiate_expression_tree"},
  {"file_path": "src/symbolic/enhanced/simplification/algebraic.py", "interface": "
      function: simplify_expression"},
  {"file_path": "src/expr_transform/transformation/analysis_expression.py", "interface":
      "class: AdvancedExpressionAnalyzer"},
  {"file_path": "src/arith_algebra/differentiation.py", "interface": "class:
      AutomaticDifferentiator"},
  {"file_path": "src/arith_algebra/equation_solving/nonlinear_symbolic.py", "interface":
      "class: SymbolicEquationSolver"}
])
```
              ===========Coding============
Agent: <think>
We need to implement the AdvancedDifferentialTransformer class in the file src/symbolic/
    enhanced/calculus_summation/differential.py. The class should provide three methods
    . For asymptotic_series_expansion, we will check that expansion_order is a positive
     integer and that the expression is provided. Our implementation will simulate an
    asymptotic expansion by returning a new symbolic expression with a type marker (e.g
```

```
                ., "asymptotic_series") including the original expression and the specified order.
                Similarly, differentiate_special_function should validate that the provided
                function name is among a set of allowed special functions (for example, erf, gamma,
                 beta, bessel, hypergeometric) and then simulate a specialized differentiation by
                returning a new symbolic expression that indicates that the transformation has been
                 applied. Finally, simplify_inequality will check that the input is structured (i.e
                . a dictionary) and that tolerance is a positive float reasonably bounded (we can
                enforce tolerance > 0 and maybe below 1 to avoid very high tolerance) and then
                simulate a simplification by deep-copying the input expression with additional
                metadata fields. We also include proper error handling and input validation.
</think>
<solution>
edit_whole_class_in_file("src/symbolic/enhanced/calculus_summation/differential.py", "
        AdvancedDifferentialTransformer")
```python
from typing import Any, Dict
import copy

class AdvancedDifferentialTransformer:
    """
    Provides advanced differential transformation methods for asymptotic series
            expansion,
    differentiating special functions, and simplifying inequality expressions within
            symbolic mathematics.

    This class encapsulates transformation techniques that are applied to symbolic
            expressions
    in the context of differential and asymptotic methods, enabling enhanced symbolic
            analyses and computations.

    Methods:
        asymptotic_series_expansion(expression: Any, expansion_order: int) -> Any:
            Expand the provided expression into its asymptotic series representation up
                    to the given order.

        differentiate_special_function(expression: Any, function_name: str,
            additional_params: Dict[str, Any] = None) -> Any:
            Differentiate a symbolic expression that contains a special function using
                    advanced rules.

        simplify_inequality(inequality_expr: Any, tolerance: float = 1e-5) -> Any:
            Simplify a symbolic inequality expression using transformation techniques to
                    reduce complexity.
    """
...
```
</solution>

Env:
# Code Patch Diff:
--- src/symbolic/enhanced/calculus_summation/differential.py
+++ src/symbolic/enhanced/calculus_summation/differential.py
@@ -1,4 +1,5 @@
 from typing import Any, Dict
+import copy

 class AdvancedDifferentialTransformer:
     """
@@ -18,6 +19,7 @@
         simplify_inequality(inequality_expr: Any, tolerance: float = 1e-5) -> Any:
             Simplify a symbolic inequality expression using transformation techniques
                    to reduce complexity.
     """
+    ALLOWED_SPECIAL_FUNCTIONS = {'erf', 'gamma', 'beta', 'bessel', 'hypergeometric'}

     def asymptotic_series_expansion(self, expression: Any, expansion_order: int) -> Any
            :
            """
@@ -26,15 +28,23 @@
        Args:
            expression (Any): The symbolic expression to be expanded.
            expansion_order (int): The order up to which the series expansion should be
                    computed.
-
+
        Returns:
            Any: A symbolic expression representing the asymptotic series expansion.

-        Edge Cases:
```

```
-                 - If expansion_order is non-positive, the behavior is undefined.
-                 - The function expects a valid symbolic expression input.
+         Raises:
+             ValueError: If expansion_order is not a positive integer.
+             TypeError: If the expression is not in a valid symbolic format (e.g., a
      dictionary).
          """
-         pass
+         if not isinstance(expansion_order, int) or expansion_order <= 0:
...

Agent: <solution>
Terminate()
</solution>
```

### D.4 TESTING FRAMEWORK

As illustrated in Algorithm 3, the testing framework combines **unit tests**, **regression tests**, and **integration tests** into a unified process. Unit tests validate each new patch at a fine-grained level, regression tests re-execute existing nodes to ensure backward compatibility, and integration tests verify that multiple patched components interact correctly when combined.

---

**Algorithm 3** Patch-Oriented Testing with Unit, Regression, and Integration Stages

---

**Require:** Patch set $\mathcal{P}$; repo skeleton $\mathcal{R}$; dependency code $D$; existing unit nodes $\mathcal{N}_u$; existing integration nodes $\mathcal{N}_i$; task description $\Theta$

1: **function** TESTPATCHES($\mathcal{P}, \mathcal{R}, D, \Theta$)
2:     $\mathcal{T}_{unit} \leftarrow []; \quad \mathcal{T}_{inte} \leftarrow []$
3:     $\mathcal{T}_{traj} \leftarrow \{\text{unit} : \{\}, \text{inte} : \{\}\}$
4:     $\mathcal{P}' \leftarrow \mathcal{P} \cup \text{FINDDEPPATCHES}(\mathcal{P})$         ▷ Extend patch set with dependency patches
5:     **for** patch $p \in \mathcal{P}'$ **do**
6:         $n_{old} \leftarrow \text{FINDEXISTINGUNITNODE}(\mathcal{N}_u, p)$
7:         **if** $n_{old} \neq \emptyset$ **and** SAMESIGNATUREORLOGIC($n_{old}, p$) **then**
8:             $n_{new} \leftarrow n_{old}$    ▷ Regression test: reuse existing node if signature/logic unchanged
9:         **else**
10:             $n_{new}, traj \leftarrow \text{CREATEORUPDATEUNITNODE}(p, D, \Theta, n_{old})$
11:             $\mathcal{T}_{traj}[\text{unit}][p.\text{key}] \leftarrow traj$
12:         **end if**
13:         $\mathcal{R}.\text{INSERTFILE}(n_{new}.\text{test\_file}, n_{new}.\text{test\_code})$
14:         $res \leftarrow n_{new}.\text{EXECUTETEST}()$
15:         $\mathcal{T}_{unit}.\text{APPEND}(res)$
16:     **end for**
17:     **for** patch group $\mathcal{G}$ clustered by integration-node **do**
18:         $n_{old} \leftarrow \text{FINDEXISTINGINTEGRATIONNODE}(\mathcal{N}_i, \mathcal{G})$
19:         **if** $n_{old} \neq \emptyset$ **and** ALLEQUAL($n_{old}, \mathcal{G}$) **then**
20:             $n_{new} \leftarrow n_{old}$         ▷ Regression integration test: reuse existing node
21:         **else**
22:             $n_{new}, traj \leftarrow \text{CREATEINTEGRATIONNODE}(\mathcal{G}, \Theta)$
23:             $\mathcal{T}_{traj}[\text{inte}][\mathcal{G}] \leftarrow traj$
24:             $\mathcal{R}.\text{INSERTFILE}(n_{new}.\text{test\_file}, n_{new}.\text{test\_code})$
25:         **end if**
26:         $res \leftarrow n_{new}.\text{EXECUTETEST}()$
27:         $\mathcal{T}_{inte}.\text{APPEND}(res)$
28:     **end for**
29:     **return** $\mathcal{T}_{unit} \cup \mathcal{T}_{inte}, \mathcal{T}_{traj}$
30: **end function**

---

As illustrated in Algorithm 4, the testing pipeline proceeds in a sequence of stages: branch planning, test generation, execution, judgment, and repair. First, a candidate test branch is created for the given code unit(s). Then, test code is generated and wrapped into a `TestNode` or `IntegrationTestNode`, which is executed inside a controlled Docker environment. The execu-

tion results are judged by an LLM; if failures are detected, the framework automatically generates fix queries and iteratively repairs the test until a validated version is obtained.

---

**Algorithm 4** End-to-End Test Generation, Execution, and Repair

---

**Require:** Repo skeleton $\mathcal{R}$; tested unit(s) $U$; source code $C$; optional prior test node $n_{old}$; maximum retries $T_{\max}$

1: **function** RUNTESTINGPIPELINE($\mathcal{R}, U, C, n_{old}$)      ▷ Main entry point for testing workflow
2:      // — Step 1: Plan test branches —
3:      $branch \leftarrow$ GENERATECODEBRANCH($C, n_{old}, T_{\max}$)
4:      // — Step 2: Generate candidate test code —
5:      $test\_code \leftarrow$ GENERATETEST($branch, C, U, n_{old}$)
6:      // — Step 3: Build a TestNode —
7:      **if** $U$ represents integration of multiple units **then**
8:          $n \leftarrow$ INTEGRATIONTESTNODE($U, test\_code$)
9:      **else**
10:          $n \leftarrow$ UNITTESTNODE($U, C, test\_code$)
11:      **end if**
12:      // — Step 4: Execute test code in Docker —
13:      $result \leftarrow n$.EXECUTETEST()
14:      $output \leftarrow result.stdout \parallel result.stderr$
15:      // — Step 5: LLM judge outcome —
16:      **if** $result$ contains errors **then**
17:          $(err\_type, reviews) \leftarrow$ LLMJUDGE($C, test\_code, output, branch$)
18:          **if** $err\_type \in \{\texttt{test\_code}, \texttt{environment}\}$ **then**
19:              $query \leftarrow$ GENERATEFIXQUERY($C, test\_code, output, branch, reviews$)
20:              $n \leftarrow$ FIXTESTANDENV($query, U, C, output, n$)
21:          **end if**
22:      **end if**
23:      // — Step 7: Return final validated test node —
24:      **return** $n$
25: **end function**

---

## D.5 STATISTICS OF THREE STAGE

Table 10 demonstrates that graph-guided localization provides reasonable efficiency across repositories, with *Incremental Development* generally easier to localize than *Integration Testing* or *Debugging*. In terms of models, o3-mini achieves higher localization efficiency but with larger variance, whereas qwen3-coder shows more stable yet overall lower efficiency. These results suggest that while graph guidance is effective, model capacity and stability jointly influence localization performance.

Table 10: Localization results across six open-source repositories under three task categories: *Integration Testing*, *Incremental Development*, and *Debugging*. Each entry reports the mean performance with standard deviation (mean±std) of the corresponding model–task pair.

| Model | Task | TableKit | MLKit-Py | HttpEasy | PyWebEngine | StatModeler | SymbolicMath |
|---|---|---|---|---|---|---|---|
| o3-mini | Integration Testing | 13.33±2.92 | 8.75±4.32 | 10.94±3.44 | 6.65±1.98 | 9.24±3.65 | 7.88±3.30 |
| | Incremental Development | 12.30±5.19 | 9.83±4.00 | 11.60±5.09 | 12.51±6.67 | 12.62±6.15 | 9.93±5.13 |
| | Debugging | 11.59±5.74 | 8.24±4.40 | 9.15±5.55 | 10.28±8.50 | 13.02±7.01 | 8.90±6.21 |
| qwen3-coder | Integration Testing | 6.16±2.37 | 6.62±2.12 | 7.89±2.42 | 5.93±2.06 | 9.24±3.65 | 7.88±3.30 |
| | Incremental Development | 6.81±1.87 | 7.10±1.98 | 7.48±1.85 | 6.98±1.92 | 6.49±1.79 | 7.12±1.77 |
| | Debugging | 6.75±2.21 | 6.01±2.16 | 6.25±1.82 | 6.62±2.47 | 5.94±2.19 | 6.42±1.94 |

As shown in Table 11, **o3-mini** achieves relatively high code success rates across repositories, often exceeding 75% and in some cases approaching 90%, whereas **qwen3-coder** lags behind with rates around 50–55%. In contrast, the corresponding test coverage remains moderate, typically within the 60–70% range. Figure 14 further illustrates that coverage fluctuates and tends to decline as code length increases: shorter implementations reach high class-level coverage, but both function-level

and overall coverage drop significantly with greater complexity. These results suggest that while current models are increasingly effective at generating functional code, their ability to produce comprehensive and high-quality test cases remains limited, highlighting test generation as a key bottleneck for practical deployment.

Table 11: Average success rate and test coverage (%) for six repositories across two models.

| Model | TableKit | | MLKit-Py | | HttpEasy | | PyWebEngine | | StatModeler | | SymbolicMath | |
|---|---|---|---|---|---|---|---|---|---|---|---|---|
| | Success | Coverage | Success | Coverage | Success | Coverage | Success | Coverage | Success | Coverage | Success | Coverage |
| o3-mini | 81.8% | 65.0% | 82.8% | 61.0% | 88.9% | 64.0% | 74.7% | 60.0% | 71.0% | 62.0% | 84.8% | 59.0% |
| qwen3-coder | 55.0% | 48.0% | 52.0% | 46.0% | 50.0% | 45.0% | 53.0% | 47.0% | 54.0% | 48.0% | 51.0% | 46.0% |

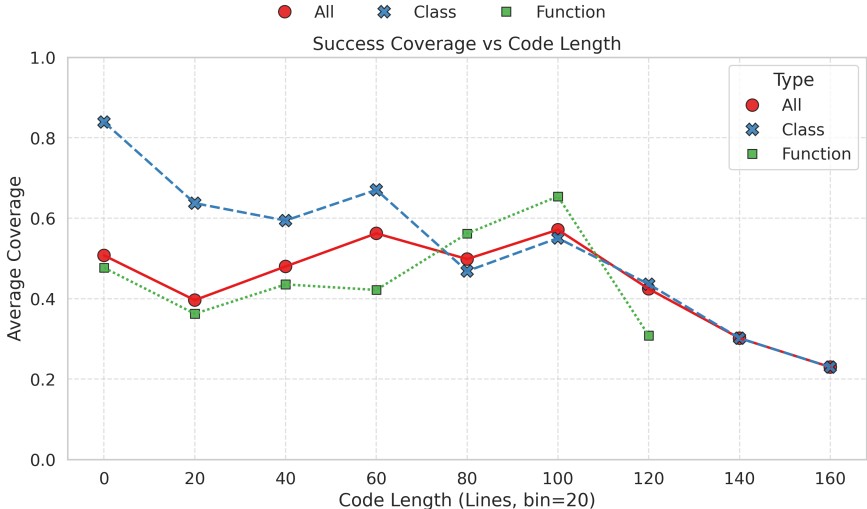

Figure 14: Test coverage of **o3-mini** on **MLKit-Py** during generation. The figure shows how the coverage of generated test functions varies as code length increases.

# E   DETAILS ABOUT REPOCRAFT BENCHMARK

In this section, we describe the construction of the REPOCRAFT benchmark, covering four key aspects: the choice of repositories, the preparation of test data, the evaluation methodology, and the configuration of agent systems.

## E.1   REPOSITORIES SELECTION

For the benchmark, we curated six representative open-source repositories: *scikit-learn*, *pandas*, *Django*, *statsmodels*, *SymPy*, and *requests*. These projects span diverse functional domains including machine learning, data analysis, web frameworks, statistical modeling, symbolic computation, and HTTP communication, thereby ensuring broad coverage of typical software development tasks. To prevent models from simply memorizing or retrieving solutions from training data, we deliberately anonymized the repositories by modifying their names and descriptions. Furthermore, the task instructions prohibit directly reusing the original implementations, requiring models to generate solutions guided only by feature specifications. This setup enforces a fairer evaluation, focusing on the models' capacity for feature-grounded reasoning and code generation rather than exploitation of prior exposure.

## E.2   EVALUATION TASKS COLLECTION

To construct a diverse and reliable evaluation set, we developed an automated pipeline that extends and systematizes the collection of test functions from the official repositories. Our design leverages

Table 12: Overview of the six benchmark repositories in REPOCRAFT. Each repository is anonymized by renaming to prevent direct memorization or retrieval by models. We list both the anonymized names and their original counterparts, together with category, purpose, and scope.

| Original Name | Anonymized Name | Category | Purpose | Scope |
|---|---|---|---|---|
| **scikit-learn** | **MLKit-Py** | Machine Learning Framework | Provides efficient tools for data mining and analysis, supporting classical supervised and unsupervised learning algorithms. | Focuses on model training, evaluation, and selection for standard ML tasks; excludes deep learning and distributed systems. |
| **sympy** | **SymbolicMath** | Symbolic Computation Library | Enables symbolic mathematics including algebraic manipulation, calculus, simplification, and equation solving. | Pure Python implementation of symbolic and algebraic computation, lightweight and extensible, without external dependencies. |
| **pandas** | **TableKit** | Data Analysis Library | Provides flexible data structures (e.g., DataFrame, Series) for manipulating and analyzing tabular data. | Supports efficient single-machine structured data processing; excludes distributed data frameworks. |
| **django** | **PyWebEngine** | Web Framework | High-level framework for rapid development with ORM, routing, templating, and admin support. | Offers an all-in-one toolkit for building web applications on small-scale/single-server deployments. |
| **requests** | **HttpEasy** | HTTP Client Library | Simple, human-friendly library for sending HTTP requests and handling responses. | Covers API for requests, responses, cookies, sessions, headers; excludes advanced networking and async features. |
| **statsmodels** | **StatModeler** | Statistical Modeling Library | Provides econometric and statistical modeling tools, including regression, time series, and hypothesis testing. | Focuses on classical statistical analysis and diagnostics; excludes modern machine learning and deep learning. |

the fact that mature open-source projects typically include comprehensive test suites with robust inputs and ground-truth outputs, ranging from unit-level checks to integration-level workflows. These tests provide a principled source of evaluation data, ensuring that generated repositories are assessed on both algorithmic diversity and functional correctness.

**Test Function Harvesting.** For each repository, we first gathered all available test functions and classes. These serve as the raw pool of evaluation candidates, capturing the behaviors developers themselves deemed important to verify.

**Hierarchical Categorization.** Next, we organized the collected tests into a hierarchical taxonomy. At the top level, categories follow the natural modular structure used by human developers (e.g., `metrics`, `linear_model`, `decomposition`). Within each category, we grouped related test classes and functions by algorithmic target. For example:

```
{
  "metrics": {
    "test_regression": {
      "functions": {
        "reg_targets": [
          "test__check_reg_targets",
          "test__check_reg_targets_exception"
        ],
        "regression_metrics": [
          "test_regression_metrics",
          "test_root_mean_squared_error_multioutput_raw_value",
          ...
        ],
        "pinball_loss": [
          "test_mean_pinball_loss_on_constant_predictions",
```

```
            "test_dummy_quantile_parameter_tuning",
            "test_pinball_loss_relation_with_mae"
        ]
      }
    }
  }
}
```

This taxonomy mirrors repository semantics: higher levels correspond to broad functional modules, while deeper levels capture fine-grained algorithmic tests.

**Sampling and Filtering.** To ensure balanced coverage, we applied the sampling algorithm(Alg 1) to draw representative subsets of test categories. Each sampled test was then refined into a task description that models could follow during generation. Finally, we filtered out cases irrelevant to core algorithmic behavior (e.g., string formatting checks, version consistency tests), retaining only tests that probe substantive computational functionality.

**Example Task Instance.** To illustrate the outcome of the pipeline, consider the following task specification extracted from the *Django* repository:

```
{
    "category": "gis_migrations",
    "file": "tests/gis_tests/gis_migrations/test_operations.py",
    "module": "class OperationTests",
    "cap": "spatial_index",
    "functions": [
        "test_create_model_spatial_index",
        "test_alter_field_add_spatial_index",
        "test_alter_field_remove_spatial_index",
        "test_alter_field_nullable_with_spatial_index",
        "test_alter_field_with_spatial_index"
    ],
    "task_query": "You are testing an algorithm that applies migration operations to GIS
        models, ensuring that spatial indexes on spatial fields are properly created, enabled
        , disabled, or removed as dictated by the migration specifications.",
    "id": "django-0109"
}
```

Each task is represented by (i) its repository category and file location, (ii) the associated test class and functions, and (iii) a natural-language query summarizing the algorithm under test.

Given such a task, the benchmark provides the **algorithm description**, its corresponding **input–output ground truth**, and the **test method**. Evaluation is then conducted along two dimensions: (1) *Algorithm Presence* — whether the generated repository contains an implementation that matches the target algorithm, and (2) *Algorithm Correctness* — whether the adapted tests pass against the generated implementation, reflecting functional accuracy. This dual perspective allows us to measure both coverage of algorithmic functionality and the reliability of generated implementations.

### E.3 AGENT PIPELINE

The evaluation employs a three-stage agent pipeline to connect task descriptions with generated repositories and derive executable judgments of success.

**Stage 1: Localization.** Given a task and its algorithmic description, the agent first explores the generated repository to locate candidate functions or classes that may implement the target algorithm. This step uses the exploration tools detailed in Appendix D.1, and produces a set of potentially relevant code anchors.

**Stage 2: Majority-Vote Validation.** To verify whether the localized candidates truly correspond to the target algorithm, we employ a majority-voting mechanism with a large language model (LLM). Each candidate is evaluated five times; the majority outcome is taken as the decision. If the validation fails, the pipeline triggers a re-localization attempt. The localization–validation loop is retried up to three times; if all attempts fail, the repository is judged to lack an implementation of the algorithm.

**Stage 3: Test Adaptation and Execution.** For validated candidates, the agent then adapts the task's reference test code. Concretely, the provided ground-truth test (including inputs, outputs, and checking methods) is rewritten to match the naming and structural conventions of the localized function or class. The adapted test is executed, and its outcome determines whether the generated implementation is functionally correct.

This pipeline ensures that evaluation captures both *coverage* (whether an algorithm is present in the generated repository) and *correctness* (whether its implementation passes the adapted tests).

### E.3.1 METRICS

To comprehensively evaluate the generated repositories, we adopt a multi-dimensional set of metrics that capture four complementary aspects: *functionality alignment*, *novelty*, *execution accuracy*, and *code scale*. The motivation is to move beyond a single success/failure judgment and instead characterize (i) whether the right algorithms are generated, (ii) whether new functionalities are introduced, (iii) whether these implementations actually work, and (iv) at what level of scale and complexity they are realized. Together, these metrics provide a holistic view of the strengths and limitations of different models.

**Functionality Coverage.** The first question is whether a model can reproduce the expected range of functionalities in a target repository. We extract feature descriptions from both ground-truth repositories and generated repositories, and define a reference set of categories $\mathcal{C} = \{c_1, \ldots, c_K\}$ based on official documentation and developer guidelines. Generated functionalities $\mathcal{G} = \{g_1, \ldots, g_N\}$ are obtained either from structured intermediate outputs (for agent-based methods) or directly from raw code (for baseline models). To align generated features with reference categories, we perform K-Means clustering with $\mathcal{C}$ as fixed centroids, plus an additional centroid $c_{\text{OOD}}$ for out-of-distribution features. Each generated feature $g_i$ is mapped to $f(g_i) \in \mathcal{C} \cup \{c_{\text{OOD}}\}$, with assignments further refined by an LLM-as-Judge to reduce semantic drift. Coverage is then defined as the fraction of reference categories that are "hit" by at least one generated feature:

$$\text{Coverage} = \frac{1}{|\mathcal{C}|} \sum_{j=1}^{K} \mathbb{1}\left[\exists g_i \in \mathcal{G}, \ f(g_i) = c_j\right]. \tag{1}$$

This metric quantifies how well the generated repository aligns with the intended functionality footprint.

**Functionality Novelty.** Coverage alone cannot distinguish between a model that simply memorizes existing categories and one that proposes extensions. To capture creativity and diversity, we measure the proportion of generated functionalities that fall outside the reference taxonomy. Specifically, novelty is the fraction of generated nodes assigned to the out-of-distribution centroid $c_{\text{OOD}}$:

$$\text{Novelty} = \frac{1}{|\mathcal{G}|} \sum_{i=1}^{N} \mathbb{1}\left[f(g_i) = c_{\text{OOD}}\right]. \tag{2}$$

High novelty indicates a tendency to introduce new capabilities, though such capabilities may or may not be useful. This metric is therefore best interpreted jointly with accuracy (below).

**Functionality Accuracy.** Even if a repository covers the right categories, the implementations must be correct. We therefore evaluate repository-specific tasks by checking whether generated code passes adapted test cases. Two complementary statistics are reported:

- **Voting Rate** — the fraction of tasks where the localization–validation pipeline confirms that an implementation of the target algorithm is present. This measures algorithm *presence*.

- **Success Rate** — the fraction of tasks where the adapted tests execute successfully. This measures algorithm *correctness*.

Together, these metrics disentangle whether errors stem from missing functionality versus incorrect implementation.

**Code-Level Statistics.** Finally, we report statistics on the scale and complexity of generated codebases. This helps distinguish minimal solutions from more realistic, full-fledged repositories. We compute these metrics over filtered Python source files, excluding directories unrelated to core functionality (e.g., `tests`, `examples`, `benchmarks`). The reported quantities are:

- **File Count**: number of valid source files, reflecting modular spread;
- **Normalized LOC**: effective lines of code after removing comments, docstrings, and blank lines, capturing implementation size;
- **Code Token Count**: number of tokens in normalized code, measured with a standard tokenizer, reflecting lexical complexity.

By jointly considering these four dimensions (coverage, novelty, accuracy in terms of presence and correctness, and scale), we obtain a nuanced evaluation of generated repositories. This design ensures that models are rewarded not only for producing functional code, but also for producing diverse, accurate, and realistically sized repositories.

### E.4 GROUND-TRUTH TAXONOMY FOR COVERAGE AND NOVELTY CALCULATION

```
{
  "Supervised learning": {
    "Linear Models": [],
    "Kernel ridge regression": [],
    "Support Vector Machines": {
      "SVC": [],
      "SVR": [],
      "OneClassSVM": []
    },
    "Nearest Neighbors": [],
    "Gaussian Processes": [],
    "Cross decomposition": [],
    "Naive Bayes": [],
    "Decision Trees": [],
    "Ensembles": [],
    "Multiclass and multioutput algorithms": [],
    "Feature selection": [],
    "Semi-supervised learning": [],
    "Isotonic regression": [],
    "Probability calibration": [],
    "Neural network models (supervised)": []
  },
  "Unsupervised learning": {
    "Gaussian mixture models": [],
    "Manifold learning": [],
    "Clustering": [],
    "Biclustering": [],
    "matrix factorization problems": [],
    "Covariance estimation": [],
    "Novelty and Outlier Detection": [],
    "Density Estimation": [],
    "Neural network models (unsupervised)": []
  },
  "Model selection and evaluation": {
    "Cross-validation": [],
    "Tuning the hyper-parameters of an estimator": [],
    "Validation curves": {
      "Classification Metrics": [],
      "Regression Metrics": [],
      "Clustering Metrics": []
    }
  },
  "Inspection": {
```

```
    "Permutation feature importance": []
  },
  "Dataset transformations": {
    "Feature extraction": {
      "Feature hashing": [],
      "Text feature extraction": [],
      "Image feature extraction": []
    },
    "Preprocessing data": {
      "Scaling and Normalization": [],
      "Discretization and Binarization": [],
      "Polynomial and Non-linear Feature Engineering": [],
      "Categorical Feature Encoding": [],
      "Missing Value Imputation": [],
      "Kernel and Matrix Centering": []
    },
    "Unsupervised dimensionality reduction": [],
    "Random Projection": [],
    "Kernel Approximation": [],
    "Pairwise metrics, Affinities and Kernels": []
  },
  "Dataset loading utilities": {},
  "Model persistence": []
}
```

```
{
  "requests": {
    "Core Request Features": {
      "HTTP Method Support": [],
      "URL and Query Handling": [],
      "Request Body Construction": [],
      "Custom Headers": []
    },
    "Response Handling": {
      "Response Body Access": [],
      "JSON Processing": [],
      "Response Metadata": [],
      "Cookie Handling": []
    },
    "Session Management": {
      "Session Persistence": [],
      "Session Customization": []
    },
    "Advanced Configuration": {
      "Timeouts and Retries": [],
      "Redirect Control": [],
      "Streaming and Chunking": [],
      "Authentication Support": [],
      "Event Hooks": []
    },
    "Security and Transport": {
      "SSL Verification": [],
      "Client Certificates": [],
      "Transport Control": [],
      "Proxy Support": []
    },
    "Compliance and Encoding": {
      "Encoding Handling": [],
      "Standards Compliance": [],
      "Blocking Behavior": []
    }
```

```
    }
  }
```

## E.5 ANALYSIS OF AUTOMATED EVALUATION

Our main results rely on an automated, model-based evaluation pipeline rather than purely manual annotation. To justify this design, we examine whether these automatic judges produce assessments that are consistent with human raters and stable across different backbone models.

### E.5.1 FUNCTIONALITY EVALUATION

Table 13: Per-repo Coverage (Cov) and Novelty (Nov) mean $\pm$ std across responses for each model and method, with human annotation as reference (all values in %).

| Repo | DeepSeek-V3.1 Cov | Nov | o3-mini Cov | Nov | GPT-5 Cov | Nov | Human Cov | Nov |
|---|---|---|---|---|---|---|---|---|
| **Method: Claude-Code** | | | | | | | | |
| TableKit | $56.1 \pm 3.3$ | $1.6 \pm 0.5$ | $55.0 \pm 2.2$ | $1.7 \pm 1.6$ | $57.9 \pm 3.7$ | $1.7 \pm 0.9$ | **54.5** | **2.1** |
| StatModeler | $30.7 \pm 2.0$ | $1.6 \pm 0.5$ | $37.4 \pm 2.9$ | $1.5 \pm 1.3$ | $41.5 \pm 0.8$ | $2.2 \pm 2.6$ | **40.1** | **2.5** |
| SymbolicMath | $64.6 \pm 4.2$ | $0.1 \pm 0.1$ | $42.2 \pm 5.7$ | $2.8 \pm 0.4$ | $68.8 \pm 0.0$ | $1.7 \pm 0.4$ | **40.1** | **2.1** |
| HttpEasy | $51.5 \pm 5.2$ | $0.0 \pm 0.0$ | $50.0 \pm 0.0$ | $0.0 \pm 0.0$ | $54.5 \pm 0.0$ | $0.0 \pm 0.0$ | **52.5** | **0.0** |
| MLKit-Py | $51.8 \pm 1.2$ | $7.9 \pm 0.3$ | $57.5 \pm 2.1$ | $12.9 \pm 11.3$ | $55.3 \pm 3.0$ | $7.6 \pm 1.5$ | **51.2** | **8.8** |
| PyWebEngine | $65.4 \pm 3.4$ | $16.3 \pm 3.5$ | $72.0 \pm 7.1$ | $18.1 \pm 11.0$ | $74.4 \pm 1.5$ | $10.0 \pm 0.6$ | **64.4** | **8.4** |
| **Method: ZeroRepo-o3-mini** | | | | | | | | |
| TableKit | $73.2 \pm 3.8$ | $0.5 \pm 0.4$ | $79.6 \pm 6.3$ | $9.9 \pm 8.3$ | $74.2 \pm 0.0$ | $1.3 \pm 0.4$ | **72.8** | **0.8** |
| StatModeler | $71.2 \pm 2.6$ | $22.6 \pm 2.0$ | $82.0 \pm 4.1$ | $28.8 \pm 12.8$ | $80.8 \pm 0.2$ | $15.8 \pm 0.4$ | **76.9** | **29.3** |
| SymbolicMath | $61.1 \pm 4.8$ | $0.0 \pm 0.1$ | $62.8 \pm 0.2$ | $0.5 \pm 0.1$ | $64.6 \pm 2.9$ | $0.2 \pm 0.1$ | **64.5** | **0.8** |
| HttpEasy | $93.9 \pm 2.6$ | $1.1 \pm 0.2$ | $96.3 \pm 3.2$ | $3.2 \pm 1.4$ | $97.7 \pm 3.2$ | $2.6 \pm 0.5$ | **100.0** | **4.4** |
| MLKit-Py | $88.7 \pm 2.5$ | $17.3 \pm 0.5$ | $88.0 \pm 8.6$ | $11.1 \pm 5.9$ | $94.8 \pm 3.4$ | $13.9 \pm 0.4$ | **94.2** | **14.9** |
| PyWebEngine | $68.1 \pm 1.2$ | $13.2 \pm 3.2$ | $80.5 \pm 1.1$ | $22.7 \pm 9.7$ | $78.0 \pm 2.1$ | $8.4 \pm 1.8$ | **74.2** | **11.1** |

We first ask whether automatic annotators capture functionality-level behavior in a way that is consistent with human judgments. Table 13 reports per-repo coverage (Cov) and novelty (Nov) means and standard deviations for each method under different automatic annotators, with human judgments as reference. Across repositories and methods, the automatic annotators recover the same qualitative patterns as humans: ZeroRepo variants attain consistently higher coverage and novelty than Claude-Code, and the magnitude of these gaps is broadly similar to that observed under human annotation. Table 14 further summarizes Pearson correlations between model-based and human scores aggregated over all repo–method pairs. Coverage correlations range from 0.71 to 0.88, and novelty correlations from 0.44 to 0.78, indicating moderate to strong alignment with human assessments. Taken together, these results suggest that our automated evaluation is reasonably robust, agrees well with human annotators at both per-repo and aggregate levels, and remains stable across different backbone models used as judges.

Table 14: Pearson Consistency Between Models and Humans on Coverage and Novelty Metrics

| Model | Coverage Pearson | Novelty Pearson |
|---|---|---|
| DeepSeek-V3.1 | 0.780121 | 0.890096 |
| gpt-4o | 0.893207 | 0.958715 |
| gpt-5 | 0.814894 | 0.869036 |

### E.5.2 LOCALIZATION AND MAJORITY-VOTE STABILITY

Beyond end-to-end coverage and novelty, we further examine the reliability of the key intermediate stages in our automated pipeline: repository-level localization and majority-vote (MV) validation. To this end, we randomly sample 200 evaluation tasks for each of two systems (ZeroRepo and the Claude-Code baseline) and have human annotators re-label the pipeline's decisions (Table 15). *Existence Rate* indicates how often the generated repository actually contains a correct implementation of

Table 15: Manual validation of localization and majority voting (MV) in our automated pipeline on 200 sampled tasks for each model. *Existence Rate* measures whether the required algorithm is present in the generated repository; *Localization Accuracy* measures correctness of the identified implementation given existence; MV metrics evaluate the reliability of the majority-vote step.

| Model | Exist. Rate | Loc. Acc. | MV Acc. | MV Prec. | MV Rec. | MV F1 |
|---|---|---|---|---|---|---|
| Claude-Code | 50% | 89.00% | 85.50% | 77.95% | 99.00% | 87.22% |
| ZeroRepo | 73% | 84.00% | 79.00% | 78.26% | 98.63% | 87.27% |

the target algorithm, *Localization Accuracy* measures whether the pipeline correctly identifies that implementation when it exists, and the MV metrics (accuracy, precision, recall, F1) quantify the reliability of the majority-vote step. We observe high localization accuracy for both systems (84–89%) and consistently strong MV F1 (about 87% for both ZeroRepo and Claude-Code), with MV recall above 98%. These results suggest that, conditioned on a solution being present, our pipeline can reliably locate the corresponding code and that the MV stage is both stable across models and unlikely to discard correct implementations, providing additional evidence for the robustness of our automated evaluation.

### E.5.3 ACCURACY EVALUATION STABILITY

Table 16: Accuracy (Pass / Vote Rate) for Claude Code and ZeroRepo(o3-mini) on 100 sampled tasks, evaluated by different model and human judges.

| Method | DeepSeek-V3.1 | GPT-5 | o3-mini (Ours) | **Human** |
|---|---|---|---|---|
| Claude Code | 38.7 / 52.1 | 35.1 / 49.9 | $36.2 \pm 3.25$ / $52.0 \pm 0.71$ | **33.0 / 50.0** |
| ZeroRepo (o3-mini) | 61.9 / 79.2 | 65.1 / 70.3 | $68.95 \pm 0.75$ / $79.9 \pm 4.90$ | **62.0 / 73.0** |

We further assess the stability of our accuracy (Pass / Vote Rate) evaluation pipeline under different judges. We re-run the accuracy evaluation on 100 randomly sampled tasks using DeepSeek-V3.1, GPT-5, our o3-mini-based evaluator, and human annotators (Table 16). Across all judges, ZeroRepo(o3-mini) attains substantially higher pass and vote rates than Claude Code, and the gaps are of similar magnitude for model-based and human evaluations (e.g., humans report 62.0% / 73.0% for ZeroRepo versus 33.0% / 50.0% for Claude Code). This consistent ranking and stable difference across judges indicate that the accuracy evaluation component of our pipeline is robust to the choice of judging model.

### E.5.4 TEST CODE QUALITY

Table 17: Manual audit of LLM-adapted test code for Claude Code and ZeroRepo. We randomly sample 100 tests per method from the main experiments and check their correctness and error types.

| Model | #Tests | Correct (%) | Import/Name | Sig. Mismatch | Data Mismatch | Hardcoded |
|---|---|---|---|---|---|---|
| Claude Code | 100 | 91 | 6 | 1 | 1 | 1 |
| ZeroRepo | 100 | 90 | 5 | 4 | 1 | 0 |

Finally, we examine the quality of the LLM-adapted test code used in our experiments. We manually audit 100 tests for Claude Code and 100 tests for ZeroRepo, checking whether each test correctly encodes the intended behavior and categorizing any errors (Table 17). We find that 91% of tests for Claude Code and 90% for ZeroRepo are correct. The remaining cases (9 for Claude Code, 10 for ZeroRepo) are dominated by minor, localized issues such as import or name errors, signature mismatches, and occasional data-structure mismatches; we observe only a single "hardcoded" test that trivially forces a pass. None of these errors indicate systematic flaws in the test-adaptation procedure. This analysis suggests that the vast majority of adapted tests are reliable, and that the pass/fail rates reported in the main paper are driven by genuine code quality rather than artifacts of incorrect tests.

# F EXPERIMENT RESULTS

## F.1 BASELINE CONFIGURATIONS

To ensure fair comparison, we evaluate three representative systems for repository synthesis: **MetaGPT**, **ChatDev**, and **Paper2Code**, together with several single-agent LLM baselines. All methods are run with their official or default configurations.

**MetaGPT.** MetaGPT is a multi-agent framework that simulates a software company by assigning roles such as Product Manager, Architect, Project Manager, Engineer, and Tester. The agents collaborate following predefined Standard Operating Procedures to complete planning, design, implementation, and debugging.

**ChatDev.** ChatDev also follows a company-style organization, where agents take charge of requirement analysis, coding, testing, and review. It uses a chat-based interaction mechanism to coordinate between stages. We run ChatDev with its default settings.

**Paper2Code.** Paper2Code is a fixed workflow system designed to convert machine learning papers into executable repositories. It follows a three-stage pipeline of planning, analysis, and generation, which we execute sequentially using the default setup.

**Vibe-Coding Agent (OpenHands, Codex, Claude Code, Gemini CLI).** For comparison with standalone LLM systems, we configure each model with a maximum of 30 iterations. The first round is initialized with the repository description. In each subsequent round, the model receives a fixed self-reflection prompt:

```
Please check whether the current repository still has any features that could be enhanced or
    any missing functionality that needs to be added. If there are no further improvements,
    or if you consider the task complete, please reply with "yes" only. If there are still
    potential enhancements or improvements to be made, please continue working on them, and
    do not reply with "yes" just because you are concerned about complexity.
```

## F.2 DETAILED EXPERIMENT RESULTS

We report the results of different methods on six repositories. For each repository, the methods are evaluated under the same settings to enable direct comparison.

Table 18: Performance on the **MLKit-Py** "Nov." denotes the novelty rate; the number in parentheses is Novel/Total, where Novel is the number of novel functionalities and Total is the total number of planned functionalities.

| Agent | Model | Cov. (%) ↑ | Nov. (%) (Novel/Total) ↑ | Pass. / Vot. (%) ↑ | Files ↑ | LOC ↑ | Tokens ↑ |
|---|---|---|---|---|---|---|---|
| MetaGPT | o3-mini | 14.9 | 0.0 (0.0/13.0) | 6.3 / 7.3 | 3.0 | 95.0 | 928.0 |
| | Qwen3-Coder | 19.2 | 0.0 (0.0/23.0) | 9.9 / 12.0 | 8.0 | 170.0 | 1718 |
| ChatDev | o3-mini | 8.5 | 14.3 (2/14) | 6.3 / 7.3 | 6 | 163 | 2064 |
| | Qwen3-Coder | 12.8 | 0.0 (0/49) | 10.5 / 11.5 | 7 | 280 | 3100 |
| OpenHands | o3-mini | 31.9 | 0.0 (0/39) | 11.5 / 13.6 | 14 | 272 | 2499 |
| | Qwen3-Coder | 34.0 | 0.0 (0/48) | 11.0 / 14.0 | 26 | 1020 | 10213 |
| Paper2Code | o3-mini | 25.5 | 0.0 (0/41) | 17.8 / 19.9 | 5 | 564 | 6346 |
| | Qwen3-Coder | 31.9 | 0.0 (0/118) | 18.8 / 24.6 | 12 | 1710 | 20821 |
| Codex CLI | o3 pro | 31.9 | 0.0 (0/59) | 11.0 / 16.9 | 14 | 829 | 8344 |
| Gemini CLI | gemini 2.5 pro | 59.6 | 0.0 (0/141) | 0.0 / 33.5 | 19 | 2316 | 24782 |
| Claude Code CLI | claude 4 sonnet | 59.6 | 0.0 (0/163) | 27.5 / 42.4 | 31 | 3559 | 37056 |
| Gold Projects | Human Developers | - | - | 85.1 / 98.3 | 185 | 65972 | 592187 |
| **ZeroRepo** | o3-mini | **97.9** | 4.7 (54/1258) | 73.5 / 78.7 | 266 | 31596 | 351554 |
| | Qwen3-Coder | 85.1 | 15.0 (176/1233) | 63.6 / 74.6 | 642 | 60553 | 741634 |

Table 19: Performance on the **HttpEasy** repo. "Nov." denotes the novelty rate; the number in parentheses is Novel/Total, where Novel is the number of novel functionalities and Total is the total number of planned functionalities.

| Agent | Model | Cov. (%) ↑ | Nov. (%) (Novel/Total) ↑ | Pass. / Vot. (%) ↑ | Files ↑ | LOC ↑ | Tokens ↑ |
|---|---|---|---|---|---|---|---|
| MetaGPT | o3-mini | 22.7 | 0.0 (0/12) | 5.0 / 15.0 | 1 | 167 | 1802 |
| | Qwen3-Coder | 31.8 | 0.0 (0/17) | 20.0 / 25.0 | 4 | 175 | 2023 |
| ChatDev | o3-mini | 36.4 | 18.2 (2/11) | 15.0 / 15.0 | 3 | 177 | 2055 |
| | Qwen3-Coder | 40.9 | 3.5 (1/31) | 20.0 / 30.0 | 2 | 323 | 3151 |
| OpenHands | o3-mini | 22.7 | 0.0 (0/5) | 20.5 / 28.2 | 3 | 72 | 669 |
| | Qwen3-Coder | 31.8 | 0.0 (0/20) | 20.0 / 30.0 | 2 | 214 | 1960 |
| Paper2Code | o3-mini | 27.3 | 0.0 (0/18) | 0.0 / 24.2 | 5 | 192 | 1856 |
| | Qwen3-Coder | 50.0 | 2.7 (1/39) | 0.0 / 45.5 | 5 | 377 | 3965 |
| Codex CLI | o3 pro | 45.5 | 0.0 (0/19) | 14.0 / 28.0 | 1 | 197 | 1879 |
| Gemini CLI | gemini 2.5 pro | 59.1 | 3.1 (1/33) | 40.0 / 56.0 | 1 | 420 | 5407 |
| Claude Code CLI | claude 4 sonnet | 50.0 | 0.0 (0/21) | 36.0 / 42.0 | 2 | 436 | 4931 |
| Gold Projects | Human Developers | - | - | 72.3 / 87.2 | 17 | 2793 | 22297 |
| **ZeroRepo** | o3-mini | **100.0** | 2.05 (7/433) | 64.0 / 72.0 | 109 | 6192 | 61922 |
| | Qwen3-Coder | 95.5 | 0.3 (2/854) | 54.0 / 64.0 | 245 | 15559 | 165051 |

Table 20: Performance on the **PyWebEngine** repo. "Nov." denotes the novelty rate; the number in parentheses is Novel/Total, where Novel is the number of novel functionalities and Total is the total number of planned functionalities.

| Agent | Model | Cov. (%) ↑ | Nov. (%) (Novel/Total) ↑ | Pass. / Vot. (%) ↑ | Files ↑ | LOC ↑ | Tokens ↑ |
|---|---|---|---|---|---|---|---|
| MetaGPT | o3-mini | 27.1 | 0.0 (0/52) | 0.0 / 13.5 | 2 | 421 | 3733 |
| | Qwen3-Coder | 18.8 | 0.0 (0/52) | 0.0 / 9.2 | 9 | 238 | 1928 |
| ChatDev | o3-mini | 25.0 | 0.0 (0/40) | 0.0 / 14.2 | 8 | 372 | 3185 |
| | Qwen3-Coder | 27.1 | 0.0 (0/49) | 0.0 / 12.1 | 11 | 679 | 5950 |
| OpenHands | o3-mini | 31.3 | 2.0 (1/55) | 0.0 / 14.2 | 18 | 304 | 2628 |
| | Qwen3-Coder | 25.0 | 0.0 (0/52) | 0.0 / 19.1 | 13 | 427 | 3996 |
| Paper2Code | o3-mini | 27.1 | 0.0 (0/46) | 0.0 / 15.6 | 11 | 619 | 6342 |
| | Qwen3-Coder | 43.8 | 0.0 (0/103) | 0.0 / 19.9 | 10 | 1761 | 16076 |
| Codex CLI | o3 pro | 39.6 | 0.0 (0/88) | 12.1 / 26.7 | 2 | 769 | 7751 |
| Gemini CLI | gemini 2.5 pro | 45.8 | 0.3 (1/318) | 7.6 / 48.1 | 45 | 2975 | 27655 |
| Claude Code CLI | claude 4 sonnet | 64.6 | 38.1 (669/2165) | 33.9 / 66.1 | 80 | 34302 | 317883 |
| Gold Projects | Human Developers | - | - | 81.6 / 86.5 | 681 | 109457 | 917622 |
| **ZeroRepo** | o3-mini | **79.2** | 38.2 (566/1680) | 74.1 / 84.4 | 430 | 27647 | 275782 |
| | Qwen3-Coder | 68.8 | 18.1 (244/1561) | 56.4 / 64.8 | 521 | 48058 | 539052 |

## F.3 EXAMPLES OF COVERAGE CALCULATION AND NOVELTY ASSESSMENT

In this subsection, we provide examples of how coverage and novelty are computed from the constructed RPG, illustrating category alignment for coverage and out-of-distribution detection for novelty.

**Analysis of Coverage Examples.** These examples demonstrate that our coverage metric provides a reasonable allocation of generated functionalities to reference categories. Core areas such as regression, classification, clustering, and preprocessing are consistently captured, while supporting utilities (e.g., normalization, imputation) are distributed into their respective modules without overlap or misplacement. This validates the soundness of our metric design for assessing functional completeness. Moreover, the RPG ensures that functionalities are not only well aligned with reference categories but also diversified across them, highlighting its effectiveness as a planning substrate for repository-level generation.

```
{
  "SVR": [
    "NuSVR"
  ],
  "Gaussian mixture models": [
```

Table 21: Performance on the **TableKit** repo. "Nov." denotes the novelty rate; the number in parentheses is Novel/Total, where Novel is the number of novel functionalities and Total is the total number of planned functionalities.

| Agent | Model | Cov. (%) ↑ | Nov. (%) (Novel/Total) ↑ | Pass. / Vot. (%) ↑ | Files ↑ | LOC ↑ | Tokens ↑ |
|---|---|---|---|---|---|---|---|
| MetaGPT | o3-mini | 13.2 | 0.0 (0/21) | 0.0 / 11.5 | 1 | 186 | 1814 |
|  | Qwen3-Coder | 6.6 | 0.0 (0/17) | 0.0 / 6.4 | 3 | 133 | 1453 |
| ChatDev | o3-mini | 21.1 | 0.0 (0/36) | 0.0 / 15.0 | 2 | 332 | 3517 |
|  | Qwen3-Coder | 19.7 | 0.0 (0/54) | 0.0 / 0.0 | 6 | 918 | 9168 |
| OpenHands | o3-mini | 11.8 | 0.0 (0/26) | 0.0 / 18.1 | 6 | 193 | 1753 |
|  | Qwen3-Coder | 11.8 | 0.0 (0/23) | 0.0 / 12.1 | 2 | 174 | 1914 |
| Paper2Code | o3-mini | 17.1 | 9.4 (5/53) | 0.0 / 23.5 | 7 | 529 | 5325 |
|  | Qwen3-Coder | 17.1 | 0.0 (0/61) | 6.2 / 20.4 | 9 | 1886 | 19337 |
| Codex CLI | o3 pro | 11.8 | 0.0 (0/23) | 21.1 / 30.9 | 2 | 552 | 6299 |
| Gemini CLI | gemini 2.5 pro | 44.7 | 0.0 (0/117) | 38.6 / 48.5 | 15 | 1249 | 12242 |
| Claude Code CLI | claude 4 sonnet | 52.6 | 0.0 (0/191) | 53.1 / 77.7 | 11 | 8509 | 83834 |
| Gold Projects | Human Developers | - | - | 90.6 / 94.0 | 217 | 106447 | 943873 |
| **ZeroRepo** | o3-mini | **72.4** | 21.1 (306/1701) | 81.4 / 88.3 | 477 | 37331 | 395536 |
|  | Qwen3-Coder | 65.8 | 13.9 (178/1500) | 48.0 / 64.8 | 347 | 32387 | 389886 |

Table 22: Performance on the **StatModeler** repo. "Nov." denotes the novelty rate; the number in parentheses is Novel/Total, where Novel is the number of novel functionalities and Total is the total number of planned functionalities.

| Agent | Model | Cov. (%) ↑ | Nov. (%) (Novel/Total) ↑ | Pass. / Vot. (%) ↑ | Files ↑ | LOC ↑ | Tokens ↑ |
|---|---|---|---|---|---|---|---|
| MetaGPT | o3-mini | 11.4 | 0.0 (0/19) | 5.6 / 6.1 | 6 | 228 | 2330 |
|  | Qwen3-Coder | 5.7 | 0.0 (0/10) | 0.0 / 2.8 | 13 | 437 | 5435 |
| ChatDev | o3-mini | 10.2 | 21.1 (8/38) | 1.1 / 9.5 | 9 | 726 | 9644 |
|  | Qwen3-Coder | 11.4 | 0.0 (0/18) | 3.2 / 7.7 | 6 | 320 | 3797 |
| OpenHands | o3-mini | 13.6 | 0.0 (0/32) | 7.9 / 9.0 | 9 | 335 | 3338 |
|  | Qwen3-Coder | 14.8 | 0.0 (0/27) | 9.5 / 12.8 | 5 | 670 | 8476 |
| Paper2Code | o3-mini | 12.5 | 21.6 (8/29) | 0.0 / 10.7 | 9 | 813 | 9793 |
|  | Qwen3-Coder | 13.6 | 30.0 (12/50) | 3.2 / 14.0 | 8 | 1179 | 13519 |
| Codex CLI | o3 pro | 20.5 | 0.0 (0/23) | 8.2 / 9.9 | 9 | 709 | 8473 |
| Gemini CLI | gemini 2.5 pro | 23.7 | 0.0 (0/55) | 13.5 / 23.2 | 6 | 736 | 8063 |
| Claude Code CLI | claude 4 sonnet | 34.1 | 0.0 (0/191) | 18.4 / 27.8 | 28 | 4043 | 46182 |
| Gold Projects | Human Developers | - | - | 87.2 / 96.2 | 271 | 83325 | 893824 |
| **ZeroRepo** | o3-mini | **77.3** | 15.6 (143/1021) | 76.4 / 81.1 | 220 | 24141 | 294292 |
|  | Qwen3-Coder | 77.3 | 8.2 (83/1113) | 66.2 / 73.9 | 436 | 47370 | 598058 |

```
    "gmm expectation maximization",
    "dp gaussian mixture"
  ],
  "Scaling and Normalization": [
    "quantile scaling",
    "scale to [0, 1]",
    "z-score scaling",
    "IQR scaling"
  ],
  "Missing Value Imputation": [
    "mean imputation",
    "matrix completion imputation",
    "impute using K-nearest neighbors",
    "impute with global median",
    "impute missing data",
  ],
  "Ensembles": [
    "light gradient boosting",
    "HistGradientBoosting",
    "bagging classification trees",
    "random forest",
    "LightGBM",
```

```
    "CatBoost",
    "XGBoost"
  ],
  "Clustering Metrics": [
    "density peak clustering",
    "gap statistic",
    "silhouette score calculation",
    "inertia calculation"
  ],
  "Naive Bayes": [
    "multinomial naive bayes",
    "bernoulli naive bayes",
    "gaussian naive bayes"
  ],
  "Linear Models": [
    "ridge regression",
    "lasso regression",
    "huber regression",
    "ransac regression"
  ],
  "SVC": [
    "soft margin SVM",
    "hard margin SVM",
    "SVM with precomputed kernel"
  ]
}
```

```
{
  "Proxy Support": [
    "rotate proxy list",
    "auto detect system proxy",
    "custom dns resolver integration"
  ],
  "HTTP Method Support": [
    "send POST request",
    "GET request with cookies",
    "send DELETE request",
    "PUT with JSON payload"
  ],
  "URL and Query Handling": [
    "encode path segments",
    "parse query string",
    "normalize request url"
  ],
  "Redirect Control": [
    "auto follow redirects",
    "limit redirect chain"
  ],
  "Authentication Support": [
    "send basic auth",
    "include oauth2 bearer token",
    "refresh auth token"
  ],
  "Timeouts and Retries": [
    "set request timeout",
    "apply exponential backoff",
    "custom retry hook"
  ],
  "JSON Processing": [
    "auto deserialize json",
    "validate json schema",
```

```
    "serialize dict to JSON"
  ],
  "SSL Verification": [
    "ssl hostname verification",
    "load custom certificates"
  ],
  "Streaming and Chunking": [
    "process chunked response",
    "resume file download support"
  ]
}
```

**Analysis of Novelty Examples**   The novelty cases illustrate two key observations. First, novelty captures meaningful extensions rather than random noise: in `MLKit-Py`, we see coherent additions such as *Prophet forecasting*, *STL decomposition*, and *genetic programming feature synthesis*, while in `StatModeler` new capabilities include *vector autoregression* and *Cox proportional hazards models*. Second, the new functionalities proposed by the RPG remain reasonable within the target domain: they extend statistical modeling, optimization, or robustness analysis in ways that align with real-world software evolution. Together, these examples confirm that the RPG supports not only stable replication of reference repositories but also the introduction of coherent and domain-consistent innovations.

```
{
  "new_features": [
    "vector autoregression model",
    "forecasting with Prophet",
    "genetic programming feature synthesis",
    "multi-objective bayesian optimization",
    "online learning",
    "apriori association mining",
    "Cox proportional hazards model",
    "STL decomposition",
    "temporal drift detection",
    "fuzz testing",
    "interactive dashboards",
    "NoSQL queries",
    "pareto optimization",
    "demographic parity test",
    "secure argument parsing",
    ...
  ]
}
```

```
{
  "new_features": [
    "vector autoregression model",
    "forecasting with Prophet",
    "genetic programming feature synthesis",
    "multi-objective bayesian optimization",
    "online learning",
    "apriori association mining",
    "Cox proportional hazards model",
    "STL decomposition",
    "temporal drift detection",
    "fuzz testing",
    "interactive dashboards",
    "NoSQL queries",
```

```
      "pareto optimization",
      "demographic parity test",
      "secure argument parsing",
      ...
   ]
}
```

## F.4 EXAMPLES OF LOCALIZATION BEHAVIOR

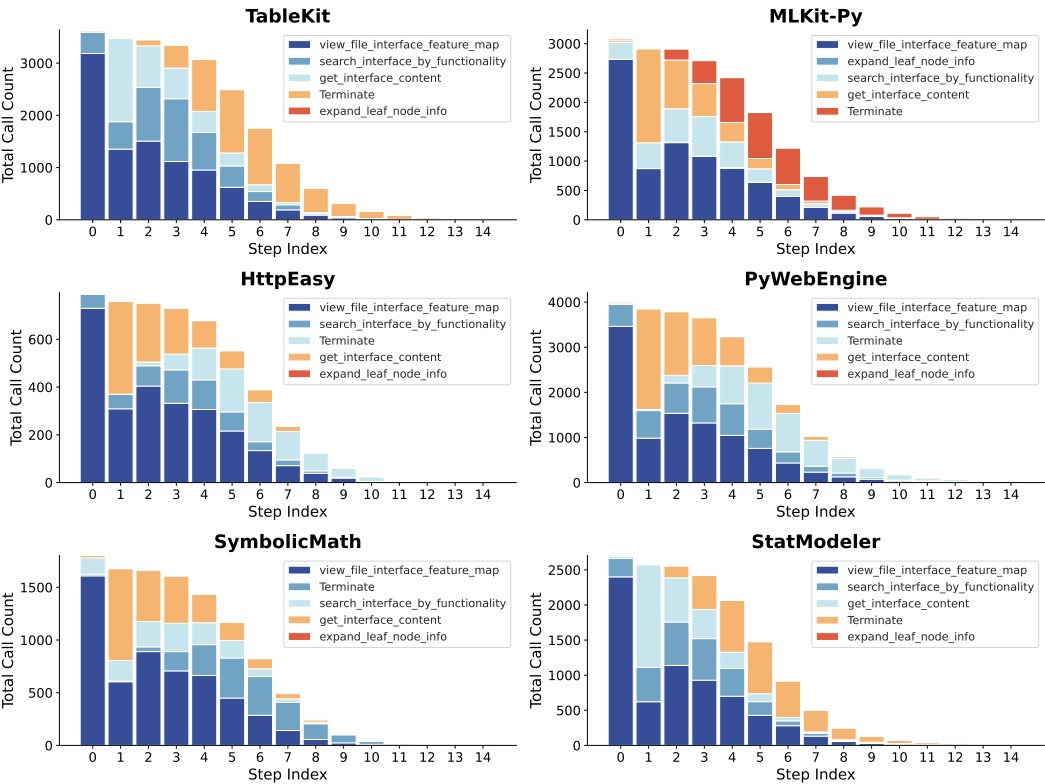

Figure 15: Aggregated function call frequency distribution across localization steps in all repositories using o3-mini.

**Graph guidance structures localization into systematic search.** Figure 15 shows that with graph guidance, localization behavior follows a structured **CCG** pattern (Coarse Search → Content Inspection → Global Graph Exploration). The agent begins by traversing the RPG at a coarse level to identify high-level candidates, then inspects content-rich nodes for detailed signals, and finally explores semantically related structures across the graph. Termination calls rise as the search converges. This progression indicates that the RPG reshapes the agent's behavior into a systematic, relation-aware search process, replacing ad hoc or repetitive probing.

