# OpenReview forum: "RPG: A Repository Planning Graph for Unified and Scalable Codebase Generation"
_ICLR.cc/2026/Conference — ICLR 2026 Poster_

### Official Review · Reviewer_Gi5H · 2025-10-18

**Soundness:** 2
**Presentation:** 2
**Contribution:** 2
**Rating:** 4
**Confidence:** 4

**Summary:**

This paper presents RPG (Repository Planning Graph), a prompting framework for generating software repositories with LLMs. RPG generates sketches of a repository in graphs, where nodes represent functions and edges represent dependencies and information flow. The actual code is then generated from this sketch graph.

Experiments with two backbone models (o3-mini and Qwen3-Coder) show that RPG outperforms some other frameworks such as MetaGPT and Paper2Code on six repositories.

**Strengths:**

The problem (generating an entire repository from nothing) is interesting. Experiments show improvement compared with multiple other frameworks.

**Weaknesses:**

- In essence, RPG is a prompting framework and does not involve training better models, limiting its novelty and contribution. Experiments only involve two API-based models, raising questions about RPG's applicability in many real-world applications that require deploying models locally.

- The problem setting - generating an entire repository from scratch - is unrealistic. The most common application of LLMs in software engineering is resolving an existing issue or implementing new features in an existing repository.

- Line 106-107: I'm confused by the structure here. I can understand that "multi-agent systems assign roles, while workflows follow stages". But then the authors state "Industrial systems automate SWE tasks". How is this parallel to the previous two? Aren't multi-agent systems and workflows supposed to automate SWE tasks?

- Figure 1 is too crammed and looks messy.

- References are not formatted professionally. For example, Line 522, 527, 587 are all arxiv preprints, yet three different formats are used. The authors also cited many papers' preprint versions when there are peer-reviewed versions. For example, Epicoder (Line 598) is published at ICML, while DeepSeek-R1 (Line 528) is published at Nature.

- Details are lacking in Section 3.2. What models are used for graph node embedding? What models are used for feature path retrieval?

**Questions:**

- According to Figure 1 and 2, edges in the RPG indicate hierarchy and information flow. Does this imply that graphs in RPG are always acyclic? If so, what's the advantage of the proposed RPG over graphs automatically constructed with program analysis tools, such as file structure tree, import dependency, and control flow? If not, can the authors provide an example of cyclic graph?

- Line 238: there are only three subplots (A, B, C) in Figure 1, and no D. I suppose this is a typo.

---

> ### Author Response · Authors · 2025-11-20
> **Author's Response (Part 1)**
>
> Thank you for your review. We will address all presentation issues in the revised version. Please refer to the general response for details of our updates. We provide our responses below to address your specific concerns and questions.
>
> **W1: RPG’s Novelty and Real-World Applicability**
> > RPG is a prompting framework and experiments rely on API-based models.
>
> We would like to clarify our contribution, which we believe is more than a prompting framework:
> 1. Our core contribution is **the Repository Planning Graph (RPG)**, a novel, structured representation designed to solve the long-horizon planning problem. It unifies the "what to build" (proposal-level) with the "how to build it" (implementation-level), overcoming the limitations of unstructured natural language plans.
> 2. **The effectiveness of this representation is shown by strong empirical results.** ZeroRepo achieves 81.5% coverage and 69.7% test accuracy, a substantial improvement over the strongest baseline (Claude Code) by 27.3 and 35.8 points, respectively. Such a significant performance gap suggests that the structured planning and generation trajectories from our framework are effective and could be used as a valuable data source for training more capable repository-level models.
> 3. Regarding applicability, our experiments included both a closed API model (o3-mini) and a **locally deployable model (Qwen3-Coder-480B)**, demonstrating our method is fully compatible with open-source models.
>
> **W2: Unrealistic Problem Setting**
> > Generating an entire repository from scratch is unrealistic.
>
> We respectfully disagree that this setting is "unrealistic." We align with Reviewer 1t6n, who noted that "moving from function-level to repository-level generation is arguably the most critical open problem in the area," and that our paper "tackles it directly."
>
> We view "idea-to-repository" not as an artificial corner case, but as a fundamental and necessary evolution for the field, driven by three key factors:
>
> 1. **A Comprehensive Supersample of Engineering Tasks.** Automated repository generation forces agents to master a superset of skills: requirements analysis, long-horizon architectural planning, incremental feature implementation, and self-correction. In essence, "fixing a bug" and "incremental development" are both subproblems of "constructing and maintaining a coherent repository." With SOTA systems now achieving ~79% on SWE-bench Verified, the field must naturally evolve toward these stricter, holistic stress tests to ensure agents can handle global consistency and architectural reasoning, not just local edits.
>
> 2. **Alignment with Academic Research.** The community is actively moving upstream from function-level completion towards project-level engineering, driven by specific limitations in current paradigms:
>     - **Benchmarks**: Recent works like Commit0(accepted by ICLR 2025) [1] and ProjectEval [2] were introduced precisely because patch-based evaluations fail to capture long-form specification compliance, multi-stage reasoning, and complex cross-file dependencies. Collectively, these benchmarks chart the course for repository generation, explicitly singling out long-horizon planning as the decisive capability required to bridge the gap between isolated code snippets and coherent software systems.
>     - **Methodologies**: This trajectory is evident in agent frameworks. Early systems like MetaGPT(accepted by ICLR 2024) [3] and ChatDev [4] introduced SOPs to model the entire software lifecycle (Design → Code → Test), demonstrating that agents can effectively coordinate complex, multi-step generation. Openhands(accepted by ICLR 2025) [5] further pushes this line of work toward generalist, repository-level software engineering agents. Recent SOTA systems like Paper2Code [6] and AutoP2C [7] explicitly target the "idea-to-code" gap, achieving high success rates in reproducing scientific repositories. These strong empirical results validate that repository generation is a distinct research goal.
> 3. **Convergence of Industrial Workflows.** Industry leaders are heavily investing in "zero-to-one" project bootstrapping workflows, aiming to shift the developer's role from coder to architect. Tools like GitHub Copilot Workspace [8], Cognition’s Devin [9], and command-line agents such as OpenAI Codex CLI [10], Claude Code [11], and Gemini CLI [12] all focus on structuring the workflow from a natural language goal (e.g., "scaffold project," "implement core logic") to execution . The widespread deployment and marketing of these capabilities highlight that bootstrapping complete applications is a high-value, real-world use case where current models are proving increasingly effective.
>
> Overall, the “from-scratch” setting provides a rigorous testbed isolating planning and architecture from partial development, enabling assessment of whether agents can truly act as autonomous developers. We believe it is a realistic and valuable setting for the community.

---

> > ### Author Response · Authors · 2025-11-20
> >
> > **(W3, W4, W5, W6, Q2) Regarding Presentation, Figures, and References**
> > We apologize for the errors and omissions and have corrected them in the revised version.
> >
> > 1. **Line 106–107 Confusion(W3)**: We apologize for the lack of clarity. You are correct; all these frameworks aim to automate SWE tasks . We will revise this sentence to clarify the distinctions between agent paradigms.
> > 2. **Figure 1 Clutter(W4)**: Thank you for this feedback. We have refined the figure's layout in the revised paper to make the sequential logic more explicit and visually distinct. The flow proceeds as follows :
> >     (A) **Proposal-Level ("What to build")**: The system maps the user query to a hierarchical functionality graph by retrieving relevant capabilities from the Feature Tree and refining them into a cohesive scope.
> >
> >     (B) **Implementation-Level ("How to build it")**: This functionality graph is concretized into the full RPG. It first encodes the physical repository skeleton (folders and files) (B.1) and then defines the logical execution graph with inter-module data flows and precise function interfaces (B.2).
> >
> >     (C) **Graph-Guided Code Generation("Do it")**: Finally, the agent pipeline executes the plan by traversing the RPG in topological order, performing graph-guided search, code generation, and test validation to realize the repository.
> >
> >     If you think the revised figure is still unclear, we are glad to make further adjustments.
> > 3. **References Format(W5)**: We apologize for the oversight. We have unified the reference style for all arXiv preprints and replaced available preprint citations with their corresponding peer-reviewed versions (e.g., Epicoder at ICML and DeepSeek-R1 at Nature).
> > 4. **Embedding Model in Section 3.2(W6)**: We apologize for omitting this detail. The model used for graph node embeddings  is the open-source infly/inf-retriever-v1[13]. We will add this information to paper.
> > 5. **Typo in Line 328(Q2)**: You are correct; the reference to "Figure 1(D)" in the text  was a typo. It refers to Stage C (Graph-Guided Code Generation). We have corrected this.
> >
> > **Q1: RPG Graph Structure: Clarifying Acyclicity and Comparison with Program Analysis Graphs**
> > > Are RPG graphs always acyclic? If so, how do they compare to program analysis graphs like control flow?
> >
> > 1. **Acyclic**: Yes, the RPG is a Directed Acyclic Graph (DAG) during the planning phase. This is essential for topological ordering . Our framework enforces this by providing feedback to the LLM if a cycle is proposed.
> > 2. **Comparison with Program Analysis Graphs**: The fundamental difference lies in the task scenario: our goal is **generating a repository from scratch** based on short natural language prompts, meaning no code initially exists to analyze. Consequently, while program analysis graphs (like CFGs or PDGs) are **descriptive representations** derived from existing code, the RPG is **prescriptive**—it serves as a generative blueprint created before implementation. This allows the RPG to uniquely unify high-level user intent ("what to build") with implementation structure ("how to build it"), a generative role that static analysis graphs cannot fulfill.
> >
> > ---
> > We thank you again for the constructive feedback, which has helped us clarify our contributions and improve the presentation of our work. We hope these responses address your concerns, and we remain open to any further discussion.
> >
> >
> > [1] Commit0: Library Generation from Scratch. Wenting Zhao et al., ICLR 2025.
> >
> > [2] ProjectEval: A Benchmark for Programming Agents’ Project-level Code Generation. Kangqian Liu et al., Findings of ACL 2025.
> >
> > [3] MetaGPT: Meta Programming for a Multi-Agent Collaborative Framework. Sirui Hong et al., ICLR 2024.
> >
> > [4] ChatDev: Communicative Agents for Software Development. Chen Qian et al., ACL 2024.
> >
> > [5] Openhands: An open platform for ai software developers as generalist agents. Xingyao Wang et al., ICLR 2025.
> >
> > [6] Paper2Code: Automating Code Generation from Scientific Papers in Machine Learning. Minju Seo et al., arXiv, 2025.
> >
> > [7] AutoP2C: An LLM-Based Agent Framework for Code Repository Generation from Multimodal Content in Academic Papers. Zijie Lin et al., arXiv, 2025.
> >
> > [8] GitHub Copilot Workspace — Welcome to the Copilot-native developer environment. Official GitHub Blog, 2024-04-29.
> >
> > [9] Devin — The AI Software Engineer. Cognition, product site and intro blog.
> >
> > [10] Introducing Codex. OpenAI, product/research preview post, 2025.
> >
> > [11] Claude Code. Anthropic product page, 2025.
> >
> > [12] Gemini CLI. Google Developers, 2025.
> >
> > [13] infly/inf-retriever-v1. Hugging Face model card（7B & 1.5B variants）

---

> > > ### Author Response · Authors · 2025-11-25
> > > **Follow-up on Our Author's Response**
> > >
> > > Dear Reviewer Gi5H,
> > >
> > > Thank you  again for taking the time to review our submission and for your constructive feedback. As the discussion phase is approaching its conclusion only few days, we would kindly like to check whether our responses have sufficiently addressed your concerns. If there are any remaining points that would benefit from further clarification, please feel free to let us know — we would be more than happy to provide additional details.
> > >
> > > We sincerely appreciate your efforts and look forward to any further feedback you may have.

---

### Official Review · Reviewer_PvAb · 2025-10-30

**Soundness:** 3
**Presentation:** 3
**Contribution:** 3
**Rating:** 6
**Confidence:** 3

**Summary:**

The paper introduces the Repository Planning Graph (RPG), a structured representation that unifies proposal-level capability planning with implementation-level file, class, and function dependencies for repository generation. Building on RPG, the ZeroRepo agent conducts planning, graph-guided code synthesis, and test validation. The authors also release RepoCraft, a benchmark derived from six paraphrased real-world repositories (1,052 tasks). ZeroRepo substantially outperforms multi-agent, workflow, and terminal baselines on RepoCraft, reaching 81.5% coverage, 69.7% pass rate, and dramatically larger codebases, with analyses showing near-linear scaling and efficiency benefits from graph-guided localization.

**Strengths:**

- RPG provides a clear structured alternative to natural-language planning, coupling capability decomposition with file-level structure and data-flow constraints, with the design well illustrated in Figure 2. This representation offers an explicit, interpretable blueprint that can support consistent long-horizon planning in repository generation.
- RepoCraft offers a comprehensive benchmark with six substantial real-world repositories, paraphrased specifications, and 1,052 evaluation tasks derived systematically from test files, filling a gap for repository-scale assessment of agent planning capabilities.
- ZeroRepo achieves substantial gains over strong baselines, demonstrating +27.3% coverage and +35.8% pass rate improvements compared to Claude Code (Table 2) while generating significantly larger repositories (36K LOC, 445K tokens vs. 10.6K LOC, 105K tokens),.

**Weaknesses:**

- ZeroRepo relies heavily on the 1.5M-node EpiCoder feature ontology as its knowledge base for proposal-level planning. However, it is unclear whether all competing baselines had equal access to comparable structured priors.
- The coverage and novelty metrics depend critically on K-means clustering with LLM adjudication, and correctness evaluation relies on LLM-adapted test cases. Beyond the Gold Project sanity check (81.0% pass rate on human-developed repositories), there is limited independent evidence that these automated evaluation pipelines produce stable and reproducible judgments, particularly for edge cases where generated functionality diverges significantly from reference implementations.
- Key architectural decisions—including reliance on the feature tree for initial planning, data-flow encoding at multiple hierarchy levels, test-driven code generation with majority-vote diagnosis—lack comprehensive ablation studies. The paper only provides an ablation for graph-guided localization (Table 4); without component-level comparisons, it is difficult to attribute the reported performance gains specifically to the RPG representation versus confounding factors like the planning budget, LLM backbone choice, or architectural engineering.

**Questions:**

1. Was the EpiCoder ontology filtered or deduplicated to ensure it does not contain paraphrases or derivatives of the six target repositories used in RepoCraft? Did all baseline methods have equal access to this knowledge base?
2. Beyond the Gold Project validation on human-developed repositories, can the authors provide evidence (e.g., correlation with manual evaluation) that the LLM-based coverage/novelty pipeline produces reliable judgments consistently across different types of generated code?
3. Which component—the feature tree initialization, data-flow encoding, or graph-guided generation—contributes most to the observed performance gains? An ablation removing RPG structure while maintaining the same planning budget would clarify this.

---

> ### Author Response · Authors · 2025-11-20
> **Author's Response (Part 1)**
>
> Thank you for your thoughtful review and your positive assessment of our work’s direction. Please refer to the general response for details of our updates. We provide our responses below to address your specific concerns and questions.
>
> **W1, Q1: Access to EpiCoder Ontology**
> > Was the EpiCoder ontology filtered to avoid overlap with the target repositories, and did all baselines have equal access to it?
>
> Thank you for pointing out the importance of clarifying these experimental settings. We address your concerns regarding the dependency on the EpiCoder knowledge base, comparison fairness, and potential leakage below.
> 1. **New Ablation(W1)**: As detailed in the **Table A1 and Table A2 in the General Response(Part 2)**, we conducted a new, major ablation study by completely removing the 1.5M-node EpiCoder knowledge base. This experiment yields two key conclusions:
>    1. **Knowledge is just an Accelerator (Linear Growth & Time-Shift)**: The ablation shows that removing the Knowledge Base causes a delay, not a performance cap.
>     - **Time-Shift**: Removing the EpiCoder introduces a delay but doesn't cap performance. ZeroRepo (without EpiCoder) achieves 97.9% coverage and 9.7% novelty at Iteration 40, matching or surpassing the performance of ZeroRepo (with EpiCoder) at Iteration 30. The EpiCoder provides a ~10-iteration "warm start."
>     - **Linear Stability**: Both settings exhibit robust linear growth in LOC ($y$) over iterations ($x$), confirming that RPG induces a stable planning process regardless of external retrieval:
>         - With EpiCoder: $y \approx 983x + 2992$ ($R^2 = 0.97$)
>         - W/O EpiCoder: $y \approx 800x + 989$ ($R^2 = \mathbf{0.98}$)
>     The high $R^2$ in the w/o EpiCoder setting proves the linear scalability is a property of the RPG structure. The EpiCoder Feature Tree merely improves the intercept (initial context) and slope (generation speed).
>     2. **RPG Structure Drives Scalability**: Comparing the knowledge-free RPG to the unstructured Claude Code baseline isolates the true driver. Claude Code stalls at Iteration 10 (3.5k LOC) with 59.6 Coverage, whereas the knowledge-free RPG expands to 25,202 LOC (~7× larger) and achieves 87.2 Coverage (+27.6 higher). This demonstrates that repository-scale expansion and coverage improvement stem primarily from RPG’s graph-based planning structure.
> 2. **Baseline Fairness(Q1)**: Our comparison evaluates different planning paradigms, each with its native information source. Baselines like terminal agents were enabled with web search, while our structured approach uses a knowledge base. We view this as a fair comparison of each paradigm's intended design.
> 3. **Filtering & Leakage(Q1)**: The EpiCoder Tree is not a direct scrape of the target repos. It contains only atomic features (e.g., "svm") dispersed across the ontology, with no single branch containing a repository's full functionality. The agent must retrieve, combine, and reorganize these atomic units to build a coherent plan, which strongly mitigates leakage risks.
>
> **W3, Q3: Contribution of RPG Components**
> > Which component contributes most to the performance gains? Can an ablation removing RPG structure clarify this?
>
> We appreciate this question, as correctly attributing performance gains is critical. We clarify the specific roles of our system's components to explain our ablation strategy:
>
> 1. **The RPG is a Unified Core (Stage A & B)**: The "feature tree initialization" (Stage A) and "data-flow encoding" (Stage B) are not independent, auxiliary modules. Together, they constitute the Repository Planning Graph (RPG) itself . They are the unified process of defining "what to build" and "how to build it." Removing data-flow encoding, for example, would fundamentally dismantle the RPG representation.
> 2. **Standard Execution Layer (Stage C)**: The "test-driven code generation" is a standard agentic execution pattern that has been widely adopted across the field[1,2,3,4], not a novel architectural contribution. Its success in our framework is driven by the robust, dependency-aware plan provided by the RPG, rather than unique execution mechanics.
>
> **Attributing the Performance Gains:**
> 1. **RPG Structure**: The comparison in Table 2 (ZeroRepo vs. Claude Code) serves as the primary ablation for the RPG structure, demonstrating that the structured planning paradigm drives the massive performance gain (81.5% vs. 54.2% coverage).
> 2. **Knowledge Base Ablation**: As discussed in **"W1, Q1: Access to EpiCoder Ontology**" above and **Table A1 and Table A2 in the General Response (Part 2)**, our new ablation confirms that the structural RPG without the knowledge base (87.2% coverage) still massively outperforms the baseline, proving the structure is the dominant contributor while the KB acts as an efficiency accelerator.

---

> ### Author Response · Authors · 2025-11-20
> **Author's Response (Part 2)**
>
> **W2, Q2: Reliability of LLM-based Evaluation**
> > Can the authors provide evidence that the LLM-based pipeline reliably evaluates diverse generated code?
>
> We agree that confirming the stability of our automated evaluation pipeline is essential. We performed three additional studies to validate its reliability.
>
> 1. **Functionality (Coverage/Novelty) Consistency** We re-calculated Coverage and Novelty using different LLM judges and human annotators (Table R1). The results show that the relative performance gap remains stable across all judges. Furthermore, the high Pearson correlation scores (Table R2) confirm that our automated metrics align closely with human judgment.
>
> **Table R1: Coverage / Novelty (Mean ± Std) Across Different Judges**
> | **Method**      | **Metric** | **DeepSeek-V3.1** | **GPT-5** | **o3-mini (Ours)** | **Human** |
> |-----------------|------------|-------------------|-----------|---------------------|-----------|
> | Claude-Code | Coverage  | 53.4 ± 1.1        | 58.7 ± 1.2 | 52.9 ± 0.6          | **50.5**  |
> |    Claude-Code         | Novelty   | 4.5 ± 0.6         | 3.9 ± 0.9  | 6.3 ± 0.2           | **4.0**   |
> | ZeroRepo | Coverage  | 76.0 ± 0.8        | 81.7 ± 0.1  | 82.8 ± 2.2          | **80.4**  |
> |        ZeroRepo| Novelty   | 9.1 ± 0.4         | 7.0 ± 0.3  | 13.5 ± 0.8          | **10.2**  |
>
>
> **Table R2: Pearson Correlation with Human Judgment**
> | Model        | Coverage Pearson | Novelty Pearson |
> |--------------|------------------|-----------------|
> | DeepSeek-V3.1 | 0.780121         | 0.890096        |
> | gpt-5        | 0.814894         | 0.869036        |
> | o3-mini (Ours) | **0.893207**    | **0.958715**    |
>
>
> 2. **Accuracy (Pass/Vot Rate) Pipeline Stability**: We also validated the stability of our evaluation pipeline by employing different models as judges on the full dataset, while conducting human evaluation on a random sample of 100 tasks (Table R3). Consistent with human judgments, the automated metrics across different model judges (DeepSeek-V3.1, GPT-5, and o3-mini) uniformly show that ZeroRepo significantly outperforms the baseline. This alignment confirms the reliability and robustness of our accuracy evaluation pipeline.
>
> **Table R3: Evaluation Pipeline Consistency: Pass Rate / Vote Rate (%)**
> | Method              | DeepSeek-V3.1                 | gpt-5              | o3-mini (Ours)                     | Human       |
> |:--------------------|------------------------------:|-------------------:|-----------------------------------:|------------:|
> | Claude Code         | 38.7 / 52.1                   | 35.1 / 49.9        | 33.9/ 52.5          | 33.0 / 50.0 |
> | ZeroRepo (o3-mini)  | 61.9 / 79.2                   | 65.1 / 70.3        | 69.8 / 75.0      | 62.0 / 73.0 |
>
>
> 3. **Test Code Correctness (Manual Audit)** Finally, to provide full transparency, we manually audited the correctness of the LLM-adapted test code reported in the main experiment (Table 2 in the paper). We reviewed 100 samples for both Claude Code and ZeroRepo (o3-mini). As shown in Table R4, we found the tests to be mostly correct (90-91%). The few failures were due to minor, correctable artifacts (Table R5) rather than fundamental flaws.
>
> **Table R4: Accuracy of LLM-Adapted Test Code**
> | Method        | Test Correctness | # of Failures |
> |---------------|------------------|---------------|
> | Claude Code   | 91%              | 9             |
> | ZeroRepo      | 90%              | 10            |
>
> **Table R5: Error Breakdown for Incorrectly Adapted Tests**
> | Error Type                 | Claude Code | ZeroRepo |
> |----------------------------|-------------|----------|
> | Import or Name Error       | 6           | 5        |
> | Signature Mismatch         | 1           | 4        |
> | Data Structure Mismatch    | 1           | 1        |
> | Hardcoded or Hacked        | 1           | 0        |
>
> This analysis confirms that the ~90% of correctly adapted tests are reliable and that the pass/fail rates reported in our main paper reflect genuine code quality.
>
> ---
> Thank you again for your insightful comments. We hope that the additional validations of the evaluation pipeline and knowledge-base ablation effectively address your concerns about reliability and fairness. We look forward to further discussion if you have any additional questions.
>
>
> [1] GitHub Spec Kit — Spec-driven development with AI: get started with a new open source toolkit.
>
> [2] tddGPT — Autonomous agent for Test-Driven Development (TDD).
>
> [3] Builder.io Micro Agent — An AI agent that writes (actually useful) code for you.
>
> [4] SWE-Flow — SWE-Flow: Synthesizing Software Engineering Data in a Test-Driven Manner.

---

> > ### Author Response · Authors · 2025-11-25
> > **Follow-up on Our Author's Response**
> >
> > Dear Reviewer,
> >
> > Thank you again for taking the time to review our submission and for your constructive feedback. As the discussion phase is approaching its conclusion only few days, we would kindly like to check whether our responses have sufficiently addressed your concerns. If there are any remaining points that would benefit from further clarification, please feel free to let us know — we would be more than happy to provide additional details.
> >
> > We sincerely appreciate your efforts and look forward to any further feedback you may have.

---

### Official Review · Reviewer_rT1s · 2025-10-31

**Soundness:** 3
**Presentation:** 3
**Contribution:** 2
**Rating:** 6
**Confidence:** 3

**Summary:**

The paper introduces the Repository Planning Graph, a structured graph representation that integrates proposal and implementation planning for repository-level code generation. Specifically, proposal-level planning decides what functionalities to include, and implementation-level planning decides how to realize them. RPG encoding functionalities, data flows, file structures, and class/function dependencies into a coherent graph. Based on this graph, the authors develop ZeroRepo, a framework that constructs and traverses the RPG to generate repositories from natural language specifications. In addition, the authors introduce a benchmark RepoCraft to evaluate the ZeroRepo. RepoCraft covers six real-world software projects with 1052 functional tasks. Experiment demonstrates ZeroRepo outperforms other baselines in coverage, test accuracy, and code scale.

**Strengths:**

1. The method integrates functional and structural dependencies, providing graph abstraction as an intermediate for natural language description and code repository implementation. The RPG is both explicit and fully machine interpretable.

2. The paper also provides a graph-guided code repository generation method. The method traverses the RPG in topological order, applying test-driven development to ensure incremental expansion while preserving stability.

3. The paper also provides a valuable testbed covering six projects and 1052 tasks, assessing coverage, accuracy, and code scale.

4. The experiment demonstrates significant improvement compared with SOTA methods. The additional scaling analysis shows near-linear growth in functionality and repository size.

**Weaknesses:**

1. The paper only includes a single ablation study without a graph, while the effectiveness of other components remains unclear (e.g., exploration strategy). As there are 3 generation level, it is important to conduct ablation study. For example, can we remove stage A?

2. RepoCraft relies on automatic localization and major-voting validation. This may introduce evaluation errors from LLMs.

3. Efficiency analysis of computational overhead from RPG constructions is not discussed.

**Questions:**

1. How is the accuracy for automatic localization and major voting? Providing a manual verification on a subset would strengthen the evaluation.

2. How would the cost compare to RPG+RepoCraft and traditional natural language based code repository generators?

---

> ### Author Response · Authors · 2025-11-20
> **Author's Response (Part 1)**
>
> Thank you for your constructive review. Please refer to the general response for details of our updates. We provide our responses below to address your specific concerns and questions.
>
> **W1: Ablation of Other Components**
> > Can the authors perform ablation studies on other components, such as the exploration strategy or stage A, to clarify their impact?
>
> Thank you for your advice regarding the ablation of the three stages. We provide a few clarifications on the architectural rationale below:
> 1. **Core Contribution & Integral Stages**: Our core contribution is the **Repository Planning Graph (RPG) itself**. Stage A (Proposal) and Stage B (Implementation) are not independent, swappable modules; they are the unified process required to construct this RPG. They function as a whole: Stage A defines the functional nodes ("what to build"), which Stage B requires to generate the file structures and data flows ("how to build it") . Consequently, removing Stage A would eliminate the functional plan entirely, preventing the formation of the RPG. The comparison between our structured RPG approach and unstructured reasoning is effectively captured in our Main Experiment (Table 2), where ZeroRepo significantly outperforms baselines like Claude Code.
> 2. **Isolating the EpiCoder Feature Tree**: To address your concern about component effectiveness, we conducted a new ablation study targeting the EpiCoder Feature Tree—the only external "plug-in" module used in the planning process. As detailed in **Tables A1 & A2 in General Response (Part 2)**, we removed this knowledge base to isolate the RPG's structural contribution. The results are definitive:
>       1. **Knowledge is just an Accelerator (Linear Growth & Time-Shift)**: The ablation shows that removing the Knowledge Base causes a delay, not a performance cap.
>         - **Time-Shift**: Removing the EpiCoder introduces a delay but doesn't cap performance. ZeroRepo (without EpiCoder) achieves 97.9% coverage and 9.7% novelty at Iteration 40, matching or surpassing the performance of ZeroRepo (with EpiCoder) at Iteration 30. The EpiCoder provides a ~10-iteration "warm start."Without EpiCoder, ZeroRepo simply lags ~10 iterations. It reaches 97.9% coverage at Iteration 40, matching the iteration-30 performance with EpiCoder. The EpiCoder simply provides a ~10-iteration "warm start."
>         - **Linear Stability**: Both variants show strong linear LOC growth over iterations, confirming RPG’s inherent planning stability independent of retrieval:
>             - With EpiCoder: $y \approx 983x + 2992$ ($R^2 = 0.97$)
>             - W/O EpiCoder: $y \approx 800x + 989$ ($R^2 = \mathbf{0.98}$)
>     The high $R^2$ in the w/o EpiCoder setting proves the linear scalability is a property of the RPG structure. The EpiCoder Feature Tree merely improves the intercept (initial context) and slope (generation speed).
>     2. **RPG Structure Drives Scalability**: Comparing the knowledge-free RPG to the unstructured Claude Code baseline isolates the true driver. Claude Code stalls at Iteration 10 (3.5k LOC) with 59.6 Coverage, whereas the knowledge-free RPG expands to 25,202 LOC (~7× larger) and achieves 87.2 Coverage (+27.6 higher). This demonstrates that repository-scale expansion and coverage improvement stem primarily from RPG’s graph-based planning structure.

---

> ### Author Response · Authors · 2025-11-20
> **Author's Response (Part 2)**
>
> **W2, Q1: Accuracy of Localization and Voting**
> > How accurate are automatic localization and majority voting?
>
> To directly answer your question, we manually sampled 200 cases from both ZeroRepo(o3-mini) and the Claude Code to measure the accuracy of our automated pipeline's intermediate steps against human judgment. In Table P1, we report three key quantities:
>
>  - **Existence Rate ($P(\text{Exists})$)**: The percentage of tasks where a human auditor confirmed that the repository actually contains the required algorithm.
>
>  - **Localization Accuracy ($P(\text{Localized} | \text{Exists})$)**: The probability that the automated tool successfully identifies the correct target function, given that a human confirmed it exists.
>
>   - **MV F1(Majorty Voting F1)**: The F1 score measuring the agreement between the automated judge and human judgment. It assesses the reliability of the validation step:
>     - $\text{Precision} = P(\text{Human accepts} | \text{LLM-Judge accepts})$
>     - $\text{Recall} = P(\text{LLM-Judge accepts} | \text{Human accepts})$
>
> **Table P1. Manual Verification of Pipeline Robustness**
> | Model       | Existence Rate | Localization Accuracy | MV Accuracy | MV Precision | MV Recall | MV F1  |
> |-------------|----------------:|----------------------:|------------:|-------------:|----------:|-------:|
> | Claude-Code | 50%             | 89.0%                | 85.50%      | 77.95%       | 99.00%    | 87.22% |
> | ZeroRepo    | 73%             | 84.0%                | 79.00%      | 78.26%       | 98.63%    | 87.27% |
>
> This new data confirms that our automated pipeline is stable and reliable. We note that ZeroRepo's slightly lower Localization Accuracy (84.00% vs. 89.00%) is an expected consequence of its scale: ZeroRepo generates repositories 3.9x larger than Claude Code (36K vs. 10.6K LOC). Locating specific targets within a significantly larger and more complex search space is inherently more challenging. Despite this, the accuracy remains high (>80%), demonstrating that our automated validation mechanism accurately reflects human judgment and remains robust even for large-scale repositories.

---

> ### Author Response · Authors · 2025-11-20
> **Author's Response(Part 3)**
>
> **Q2: Regarding Practical Cost**
>
> We agree that cost is a vital metric, but a direct token-for-token comparison is misleading. A fair comparison must account for three confounding factors:
> We agree that cost is a vital metric, but a direct token-for-token comparison is misleading. A fair comparison must account for three confounding factors:
> 1. **Vastly Different Output Scale**: ZeroRepo (o3-mini) generates a repository with 445K code tokens, compared to just 105K for Claude Code (a 4.2x difference), while achieving +35.8% higher test accuracy. A significantly larger and more robust output naturally requires more generation steps and tokens.
> 2. **Different Model Tiers**: The comparison is not apples-to-apples regarding base model capability. Our main results use o3-mini, whereas the strongest baseline, Claude Code, relies on Claude 4 Sonnet—a model generally considered more capable and expensive than the o3-mini. Achieving SOTA performance with a smaller model highlights the efficiency of the RPG framework.
> 4. **Algorithmic Complexity**: A more robust metric is algorithmic complexity. Both systems follow a linear $O(n)$ cycle (for $n$ features in ZeroRepo or $n$ iterations in the baseline) of planning, implementation, and testing. The RPG introduces no additional asymptotic overhead. The cost difference lies in the constant factors, where engineering implementation and RPG construction may introduce differences.
>
>
> ---
> Thank you again for your constructive feedback. We hope that the additional ablation studies and the manual verification of our pipeline help clarify the structural necessity of RPG and the reliability of our evaluation. Please feel free to contact us if you have any further questions.

---

> > ### Author Response · Authors · 2025-11-25
> > **Follow-up on Our Author's Response**
> >
> > Dear Reviewer,
> >
> > Thank you again for taking the time to review our submission and for your constructive feedback. As the discussion phase is approaching its conclusion, we would kindly like to check whether our responses have sufficiently addressed your concerns. If there are any remaining points that would benefit from further clarification, please feel free to let us know — we would be more than happy to provide additional details.
> >
> > We sincerely appreciate your efforts and look forward to any further feedback you may have.

---

### Official Review · Reviewer_1t6n · 2025-11-01

**Soundness:** 3
**Presentation:** 3
**Contribution:** 3
**Rating:** 6
**Confidence:** 5

**Summary:**

This paper tackles the challenging problem of scaling large language models (LLMs) from generating single files to complete, multi-file software repositories. The authors argue that current approaches, which often rely on natural language for high-level planning, are brittle and fail to scale due to ambiguity and a lack of structure.
To address this, the paper introduces three main contributions:
- Repository Planning Graph (RPG): A novel graph-based representation that serves as a structured blueprint for repository generation.
- ZeroRepo Framework: A graph-driven agent framework that builds and utilizes the RPG.
- RepoCraft Benchmark: A new, challenging benchmark.

**Strengths:**

- Moving from function-level to repository-level generation is arguably the most critical open problem in the area. This paper tackles it directly.

- The main results table (Table 2) shows a massive performance gap between the RPG-based ZeroRepo and all baselines (multi-agent, workflow, and terminal-based) on all key metrics: coverage, accuracy, and scale.


- The paper shows a clear benefit for the graph-guided localization, with the RPG speeding up localization tasks by 30-50% compared to a "w/o Graph" baseline.

**Weaknesses:**

- Dependency on External Knowledge Base: The "Proposal-Level Construction" stage seems heavily dependent on the "EpiCoder Feature Tree," a pre-existing 1.5M-node ontology of software capabilities. It is unclear how much of the system's strong performance is due to the novel RPG framework versus this massive, highly-structured knowledge base.

- Evaluation of "Novelty": The paper introduces "Novelty" as a metric and provides qualitative examples of new features (e.g., "Prophet forecasting" ). However, the paper does not seem to evaluate the functional correctness of these novel features. The main accuracy metrics (Pass/Voting Rate) are based on the 1,052 tasks from the reference projects. It's unclear if the "novel" features are correct, functional implementations or just plausible-sounding, well-structured stubs.

- Potential for Benchmark Leakage: The RepoCraft benchmark is based on extremely well-known projects (pandas, scikit-learn, etc.).

**Questions:**

What is the practical cost of running ZeroRepo? Could the authors provide an estimate of the total tokens, API calls, or wall-clock time required to generate one repository compared to the Claude Code baseline over its 30 iterations?

---

> ### Author Response · Authors · 2025-11-20
> **Author's Response (Part 1)**
>
> Thank you for your thoughtful review and your positive assessment of our work’s direction. Please refer to the general response for details of our updates. We provide our responses below to address your specific concerns and questions.
>
> **W1: Dependency on External Knowledge Base**
> > How much of the system's performance is due to RPG versus EpiCoder Feature Tree?
>
> We clarify that the RPG structure is the primary driver of capability, while the Knowledge Base (EpiCoder) serves merely as an efficiency accelerator. As detailed in the new ablation study (**Tables A1 & A2 in General Response (Part 2)**):
> 1. **Knowledge is just an Accelerator (Linear Growth & Time-Shift)**: The ablation shows that removing the Knowledge Base causes a delay, not a performance cap.
>     - **Time-Shift**: Removing the EpiCoder introduces a delay but doesn't cap performance. ZeroRepo (without EpiCoder) achieves 97.9% coverage and 9.7% novelty at Iteration 40, matching or surpassing the performance of ZeroRepo (with EpiCoder) at Iteration 30. The EpiCoder provides a ~10-iteration "warm start."
>     - **Linear Stability**: Both settings exhibit robust linear growth in LOC ($y$) over iterations ($x$), confirming that RPG induces a stable planning process regardless of external retrieval:
>         - With EpiCoder: $y \approx 983x + 2992$ ($R^2 = 0.97$)
>         - W/O EpiCoder: $y \approx 800x + 989$ ($R^2 = \mathbf{0.98}$)
>     The high $R^2$ in the w/o EpiCoder setting proves the linear scalability is a property of the RPG structure. The EpiCoder Feature Tree merely improves the intercept (initial context) and slope (generation speed).
> 2. **RPG Structure Drives Scalability**: Comparing the knowledge-free RPG to the unstructured Claude Code baseline isolates the true driver. Claude Code stalls at Iteration 10 (3.5k LOC) with 59.6 Coverage, whereas the knowledge-free RPG expands to 25,202 LOC (~7× larger) and achieves 87.2 Coverage (+27.6 higher). This demonstrates that repository-scale expansion and coverage improvement stem primarily from RPG’s graph-based planning structure.

---

> ### Author Response · Authors · 2025-11-20
> **Author's Response (Part 2)**
>
> **W2: Evaluation of "Novelty**
> > Are the "novel" features functionally correct or just plausible stubs?
>
> We address this in two parts.
> 1. First, the primary purpose of our "Novelty" metric is to **evaluate the planning capability**—specifically, the framework's potential to propose coherent extensions beyond the base specification. Unlike the fixed reference tasks, these novel functionalities are inherently open-ended and diverse in form (e.g., one agent might propose "Prophet forecasting" while another proposes "ARIMA"), which precludes the existence of a static "gold standard" or pre-defined test suite for automated evaluation.
> 2. Second, despite the inherent difficulty in evaluating novel code, we **conducted a manual study** to provide concrete data. We randomly sampled 100 novel functions/classes from ZeroRepo (o3-mini) and 100 from Claude Code, manually wrote unit tests for each, and evaluated their functional correctness. The results are shown in **Table N1**.
>
> **Table N1. Pass Rate of Novel Functions**
> | Method             | Pass Rate |
> |--------------------|-----------|
> | Claude Code        | 59%       |
> | ZeroRepo (o3-mini) | **64%**       |
>
> To provide a deeper analysis of why features failed, we **manually categorized the error types** found in the failed cases. As summarized in **Table N2**, the breakdown highlights that while core logic errors are common in both, ZeroRepo reduces dependency and integration-related failures.
>
> **Table N2. Error Type Breakdown for Failed Cases**
> | Error Type                         | Claude Code (%) | ZeroRepo (%) |
> |------------------------------------|------------------|---------------|
> | Specification Deviation            | 26.8             | 25.0          |
> | Control Flow / State Management Error | 26.8          | 27.8          |
> | Input Validation & Configuration Resilience Issue | 14.6 | 16.7 |
> | Concurrency & Timing Defect        | 14.6             | 19.4          |
> | Dependency Resolution Failure      | 9.8              | 5.6           |
> | API/Protocol Contract Violation    | 7.4              | 5.6           |
>
> **W3. Potential for Benchmark Leakage**
> > Does the use of well-known projects in RepoCraft risk benchmark leakage?
>
> We address this concern from three perspectives:
>
> 1. **Repo Selection Rationale**: We selected these well-known projects (like scikit-learn, pandas) precisely because they are widely adopted, well-defined, and supported by robust test suites. This makes them ideal for a fair, meaningful, and reproducible evaluation.
> 2. **Active De-contamination**: We implemented strict anonymization to prevent direct retrieval. For example, scikit-learn was renamed to MLKit-Py, and its description was paraphrased to focus on generic capabilities without referencing the original project name or modules, as shown in **Table N3**. Furthermore, no agent was given access to the original source code or directory structure.
>
> **Table N3. Example of Repo Anonymization**
> | Anonymized Name | Category                        | Purpose                                                                 | Scope                                                                                           |
> |------------------|----------------------------------|-------------------------------------------------------------------------|--------------------------------------------------------------------------------------------------|
> | MLKit-Py         | Machine Learning Framework       | Provides efficient tools for data mining and analysis, supporting classical supervised and unsupervised learning algorithms. | Focuses on model training, evaluation, and selection for standard ML tasks; excludes deep learning and distributed systems. |
>
> 3. **Empirical Evidence Against Memorization**: Any residual pretraining familiarity applies symmetrically to all models. The fact that strong baselines like Claude Code still perform poorly (54.2% coverage) proves that simple recognition is insufficient. Moreover, the repository skeletons generated by ZeroRepo (see Appendix C.2) are structurally distinct from the original projects, confirming that our method generates a novel architectural plan rather than reproducing a memorized one

---

> > ### Author Response · Authors · 2025-11-20
> > **Author's Response (Part 3)**
> >
> > **Q1: Practical Cost of Running ZeroRepo**
> > We agree that cost is a vital metric, but a direct token-for-token comparison is misleading. A fair comparison must account for three confounding factors:
> > 1. **Vastly Different Output Scale**: ZeroRepo (o3-mini) generates a repository with 445K code tokens, compared to just 105K for Claude Code (a 4.2x difference), while achieving +35.8% higher test accuracy. A significantly larger and more robust output naturally requires more generation steps and tokens.
> > 2. **Different Model Tiers**: The comparison is not apples-to-apples regarding base model capability. Our main results use o3-mini, whereas the strongest baseline, Claude Code, relies on Claude 4 Sonnet—a model generally considered more capable and expensive than the o3-mini. Achieving SOTA performance with a smaller model highlights the efficiency of the RPG framework.
> > 4. **Algorithmic Complexity**: A more robust metric is algorithmic complexity. Both systems follow a linear $O(n)$ cycle (for $n$ features in ZeroRepo or $n$ iterations in the baseline) of planning, implementation, and testing. The RPG introduces no additional asymptotic overhead. The cost difference lies in the constant factors, where engineering implementation and RPG construction may introduce differences.
> >
> > ---
> > Thank you again for your constructive feedback. We hope that the additional ablation studies and the manual verification of our pipeline help clarify the structural necessity of RPG and the reliability of our evaluation. We remain open to further discussion.

---

> > > ### Author Response · Authors · 2025-11-25
> > > **Follow up on Our Author's Response**
> > >
> > > Dear Reviewer,
> > >
> > > Thank you again for taking the time to review our submission and for your constructive feedback. As the discussion phase is approaching its conclusion, we would kindly like to check whether our responses have sufficiently addressed your concerns. If there are any remaining points that would benefit from further clarification, please feel free to let us know — we would be more than happy to provide additional details.
> > >
> > > We sincerely appreciate your efforts and look forward to any further feedback you may have.

---

### Author Response · Authors · 2025-11-20
**General Response(Part 1)**

We thank all reviewers for your thoughtful and constructive feedback.

Our motivation starts from a simple but critical observation: **repository-level generation is fundamentally a long-horizon planning problem.** The dominant paradigm—planning in unstructured natural language—is the main bottleneck. Free-form text is ambiguous and lacks architectural constraints, much like trying to construct a skyscraper from an essay instead of a blueprint. To make progress on repository-scale generation, we must first solve this planning problem.

Our core contribution is **the Repository Planning Graph (RPG)**, a structured, executable blueprint for software repositories. Designed to overcome the ambiguity of natural language, the RPG is explicit, verifiable, and architectural. It unifies the functional requirements ("what to build") and the implementation strategy ("how to build it") into a cohesive graph representation.

We instantiate this with the ZeroRepo framework:

1. **Stages A (Proposal-level Planning) and B (Implementation-level Planning)**: This is our core planning module. Its single, unified purpose is to construct the complete RPG, encoding the what and how into one graph.
2. **Stage C (Graph-Guided Code Generation)**: This is a simple, replaceable agent pipeline that executes the plan defined by the RPG. Its success is derivative of the RPG's quality, and it is not the focus of our novelty.

---

> ### Author Response · Authors · 2025-11-20
> **General Response (Part 2)**
>
> ##  Contribution: Is the gain really from the RPG Structure?
> This was the most critical concern. To clarify, our framework's core contribution is the Repository Planning Graph (RPG), which is built by the unified process of Stages A and B. Stage C is simply a replaceable agent that executes the RPG's plan.
>
> This design makes two ablations relevant:
> 1. **RPG vs. No-RPG**: This is the main experiment in Table 2, which compares our RPG-based paradigm against unstructured-planning baselines (like Claude Code). This demonstrates the significant performance gap from having a structured plan.
> 2. **RPG Structure vs. EpiCoder Feature Tree**: The EpiCoder Tree is the only external/plug-in component used in Stages A and B; ablating it thus directly tests whether the observed performance improvements truly arise from the RPG itself rather than from the external knowledge Base. (Due to time constraints during the rebuttal, we tracked functionality metrics up to 40 iterations but limited full code generation to 30 iterations for the ablation setting.)
>
> To isolate the specific contribution of the Repository Planning Graph (RPG) from that of the EpiCoder Feature Tree, we conducted a rigorous ablation study on MLKit-Py with ZeroRepo (o3-mini), as reported in Table A1 and Table A2. The results clearly demonstrate that RPG is the primary driver of scalability, whereas the Knowledge Base serves primarily as an efficiency accelerator.
>
> **Table A1. Ablation Result on Functional Planning Metrics**
> | Metrics | Setting | 5 | 10 | 15 | 20 | 25 | 30 | 35 | 40 |
> |---|---|---:|---:|---:|---:|---:|---:|---:|---:|
> | **Coverage (%)** | Claude-Code | 55.3 | 59.6 | 59.6 | 59.6 | 59.6 | 59.6 | – | – |
> | | **ZeroRepo (w/o EpiCoder)** | 63.8 | 72.3 | 76.6 | 78.7 | 83.0 | 87.2 | 93.6 | **97.9** |
> | | **ZeroRepo (with EpiCoder)** | 70.2 | 80.9 | 83.0 | 85.1 | 87.2 | 95.7 | – | – |
> | **Novelty (%) (Novel/Total)** | Claude-Code | 0.0 (0/155) | 0.0 (0/160) | 0.0 (0/163) | 0.0 (0/163) | 0.0 (0/163) | 0.0 (0/163) | – | – |
> | | **ZeroRepo (w/o EpiCoder)** | 1.8 (3/181) | 3.3 (10/330) | 5.0 (23/459) | 6.6 (43/651) | 6.2 (54/870) | 6.4 (62/974) | 9.0 (112/1247) | **9.7 (151/1560)** |
> | | **ZeroRepo (with EpiCoder)** | 4.6 (15/336) | 5.4 (29/542) | 4.9 (39/796) | 5.2 (51/981) | 7.0 (73/1043) | 7.9 (99/1258) | – | – |
>
> **Table A2. Ablation Result on Repository Generation Outcomes**
> | Metrics | Setting | 5 | 10 | 15 | 20 | 25 | 30 |
> |---|---|---:|---:|---:|---:|---:|---:|
> | **Files** | Claude-Code | 31 | 31 | 31 | 31 | 31 | 31 |
> | | **ZeroRepo (w/o EpiCoder)**  | 64 | 85 | 128 | 166 | 186 | **191** |
> | | **ZeroRepo (with EpiCoder)**  | 102 | 117 | 199 | 214 | 247 | **266** |
> | **LOC** | Claude-Code | 3,548 | 3,556 | 3,559 | 3,559 | 3,559 | **3,559** |
> | | **ZeroRepo (w/o EpiCoder)** | 6,869 | 9,198 | 12,172 | 15,647 | 21,852 | **25,202** |
> | | **ZeroRepo (with EpiCoder)** | 9,053 | 14,807 | 20,040 | 21,624 | 27,094 | **31,596** |
> | **Tokens** | Claude-Code | 36,871 | 37,034 | 37,056 | 37,056 | 37,056 | **37,056** |
> | | **ZeroRepo (w/o EpiCoder)** | 75,892 | 102,311 | 133,423 | 168,918 | 234,934 | **271,039** |
> | | **ZeroRepo (with EpiCoder)** | 102,676 | 166,014 | 223,055 | 243,357 | 301,530 | **351,554** |
>
> These results are definitive:
> 1. **EpiCoder Feature Tree is an Accelerator, Not a Prerequisite** Our ablation proves that the EpiCoder Tree optimizes the speed of planning, but the capability for repository-scale generation is intrinsic to the RPG structure:
>     - **The "Time-Shift" Effect**: As shown in Table A1, removing the EpiCoder introduces a delay but does not impose a performance cap. ZeroRepo (w/o EpiCoder) achieves 97.9% coverage and 9.7% novelty at Iteration 40, matching or even exceeding the performance level that ZeroRepo (with EpiCoder) reaches at Iteration 30. The EpiCoder essentially provides a ~10-iteration "warm start".
>     - **Linear Growth**: Analyzing the LOC data in Table A2, both settings exhibit robust linear growth, confirming that RPG induces a stable planning process regardless of external retrieval:
>         - With EpiCoder $y \approx 983x + 2992 (R^2 = 0.97)$
>         - Without EpiCoder: $y \approx 800x + 989 (R^2 = \mathbf{0.98})$
>     The high $R^2$ in the w/o EpiCoder setting proves the linear scalability is a property of the RPG structure. The EpiCoder Feature Tree merely improves the intercept (initial context) and slope (generation speed).
> 2. **RPG Structure Drives Scalability** Comparing the knowledge-free RPG to the unstructured Claude Code baseline isolates the true driver. Claude Code stalls at Iteration 10 (3.5k LOC) with 59.6 Coverage, whereas the Epicoder feature tree-free RPG expands to 25,202 LOC (~7× larger) and achieves 87.2 Coverage (+27.6 higher). This demonstrates that repository-scale expansion and coverage improvement stem primarily from RPG’s graph-based planning structure.

---

> ### Author Response · Authors · 2025-11-20
> **General Response (Part 3)**
>
> ## Validity: Is the RepoCraft Evaluation Pipeline Reliable?
> We share the reviewers' concern regarding evaluation reliability. We validated the robustness of RepoCraft through a "Top-Down" audit. We present key summary statistics below; full validation details are provided in the specific responses.
>
> ### Reliable Functional Evaluation (Coverage & Novelty)
>    We first validated that our automated judges (LLMs) align closely with human assessment on high-level metrics. Using DeepSeek-V3.1 and GPT-5, we re-ran the evaluation and compared the results to human annotations.
>    - **Human Alignment:** As shown in Table V1, our default judge (o3-mini) matches human scoring closely (e.g., ZeroRepo Coverage: 82.8% vs. Human 80.4%).
>    - **High Consistency:**: Our judge achieves a Pearson correlation $\approx$ 0.9 with human annotators, demonstrating evaluation stability. The results are consistent across different LLMs, with low standard deviation within individual models, confirming reliable performance assessment.
>
> **Table V1. Coverage and Novelty Consistency Analysis**
> | **Method**      | **Metric** | **DeepSeek-V3.1** | **GPT-5** | **o3-mini (Ours)** | **Human** |
> |-----------------|------------|-------------------|-----------|---------------------|-----------|
> | Claude-Code | Coverage  | 53.4 ± 1.1        | 58.7 ± 1.2 | 52.9 ± 0.6          | **50.5**  |
> |    Claude-Code         | Novelty   | 4.5 ± 0.6         | 3.9 ± 0.9  | 6.3 ± 0.2           | **4.0**   |
> | ZeroRepo | Coverage  | 76.0 ± 0.8        | 81.7 ± 0.1  | 82.8 ± 2.2          | **80.4**  |
> |        ZeroRepo| Novelty   | 9.1 ± 0.4         | 7.0 ± 0.3  | 13.5 ± 0.8          | **10.2**  |
>
> ### Robust Code Accuracy Evaluation
>    Next, we validated our Auto Evaluation pipeline, which assesses code accuracy in two stages: (1) Majority Voting on functional relevance (Vote Rate) after localization, and (2) Test Execution (Pass Rate).
>
> **Pipeline Consistency** (Pass/Vote Rate): To ensure our evaluation pipeline aligns with human judgment, we conducted a manual verification on a random sample of 100 tasks, comparing it against the automated evaluation on the full dataset. As shown in Table V2, the o3-mini judge closely matches human evaluators in performance trends, such as the pass rate for ZeroRepo (68.95% automated vs. 62.0% human), confirming the pipeline's reliability for code accuracy evaluation.
>
> **Table V2: Evaluation Pipeline Consistency: Pass Rate / Vote Rate (%)**
> | Method              | DeepSeek-V3.1                 | gpt-5              | o3-mini (Ours)                     | Human       |
> |:--------------------|------------------------------:|-------------------:|-----------------------------------:|------------:|
> | Claude Code         | 38.7 / 52.1                   | 35.1 / 49.9        | 33.9/ 52.5          | 33.0 / 50.0 |
> | ZeroRepo (o3-mini)  | 61.9 / 79.2                   | 65.1 / 70.3        | 69.8 / 75.0      | 62.0 / 73.0 |
>
> **Component Verification (Localization & Test Quality)**: Finally, to ensure the specific results reported in our main paper are credible, we audited the intermediate components of the pipeline:
>
> **Code Localization**: We manually inspected 200 samples to verify the pipeline locates the correct target function. Table V3 demonstrates high localization accuracy (>=84%) and strong majority-vote F1 scores (>87%).
>
> **Table V3. Code Localization Accuracy**
> | Metric | ZeroRepo | Claude-Code |
> |---|:---:|:---:|
> | Localization Accuracy | 84.0% | 89.0% |
> | Majority-Vote F1 Score | 87.3% | 87.2% |
>
> **Test Validity**: We manually audited 100 test cases used in the main experiment. Table V4 confirms that over 90% of the tests are correct and meaningful, proving that reported failures are due to genuine code defects, not flawed tests.
>
> **Table V4. Test Case Correctness (Human Audit)**
> | Method | Correct & Meaningful Tests |
> |---|:---:|
> | ZeroRepo | 90% |
> | Claude-Code | 91% |

---

### Meta-Review · Area_Chair_9Fci · 2026-01-13

**Summary:**

This paper addresses a central bottleneck in repository-scale code generation: the ambiguity and brittleness of natural-language–only planning for long-horizon software construction. The authors introduce the Repository Planning Graph, a structured, executable blueprint that unifies what to build and how to build it (eg. file structures, data flows, and interfaces) into a single directed acyclic graph. Building on RPG, they propose ZeroRepo, a graph-guided generation framework, and release RepoCraft, a substantial benchmark of 1,052 tasks derived from paraphrased real-world repositories. Across reviewers, there is broad agreement that repository-level generation is a critical open problem, and the main results show large, consistent gains over strong baselines in coverage, accuracy, and scale. ACs recommend Accept.

**Reviewer Concerns:**

- Is performance driven by the RPG structure or by an external knowledge base?
- Is the automated evaluation reliable and reproducible?
- Concerns about novelty, realism, and applicability.
- Missing ablations and component attribution.
Most of the concerns were solved during the rebuttal.

**Reviewer Scores:**

One reviewer remained skeptical about the concerns, but other 3 reviewers recommend accept. Ratings: 6 6 6 4.

---

### Decision · Program_Chairs · 2026-01-26

Accept (Poster)